# Ultrasound-mediated delivery of doxorubicin to the brain results in immune modulation and improved responses to PD-1 blockade in gliomas

Given the marginal penetration of most drugs across the blood-brain barrier, the efficacy of various agents remains limited for glioblastoma (GBM). Here we employ low-intensity pulsed ultrasound (LIPU) and intravenously administered microbubbles (MB) to open the blood-brain barrier and increase the concentration of liposomal doxorubicin and PD-1 blocking antibodies (aPD-1). We report results on a cohort of 4 GBM patients and preclinical models treated with this approach. LIPU/MB increases the concentration of doxorubicin by 2-fold and 3.9-fold in the human and murine brains two days after sonication, respectively. Similarly, LIPU/MB-mediated blood-brain barrier disruption leads to a 6-fold and a 2-fold increase in aPD-1 concentrations in murine brains and peritumoral brain regions from GBM patients treated with pembrolizumab, respectively. Doxorubicin and aPD-1 delivered with LIPU/MB upregulate major histocompatibility complex (MHC) class I and II in tumor cells. Increased brain concentrations of doxorubicin achieved by LIPU/MB elicit IFN-γ and MHC class I expression in microglia and macrophages. Doxorubicin and aPD-1 delivered with LIPU/MB results in the long-term survival of most glioma-bearing mice, which rely on myeloid cells and lymphocytes for their efficacy. Overall, this translational study supports the utility of LIPU/MB to potentiate the anti-tumoral activities of doxorubicin and aPD-1 for GBM.

The prognosis for patients suffering from glioblastoma (GBM) remains dismal despite extensive molecular characterization. The failure of several drug-based therapeutic approaches may be in part explained by the blood-brain barrier (BBB) that prevents sufficient brain penetration for most agents. For instance, modern antibody-based treatments that have improved the outcomes of many solid tumors do not cross the BBB[1]. Indeed, GBM cells are known to migrate and infiltrate brain regions well beyond what is revealed on magnetic resonance imaging (MRI) by contrast uptake where the BBB is impermeable to several systemically delivered agents[2,3]. Even when the enhancing tumor region is completely resected, infiltrating residual tumor cells lead to recurrence, with patients almost invariably succumbing to their disease[4,5].

Penetration of different drugs and biologicals in the brain can be achieved through the opening of the BBB with low-intensity pulsed ultrasound (LIPU) in combination with intravenous injection of microbubbles (MB), i.e., LIPU/MB[6–16]. This technology works by using a skull-implantable device or MRI-guided transcranial focused ultrasound (FUS) that sends ultrasound waves that induce the vibration of MB to open the BBB[10–13,16,17]. This technique has resulted in increased brain concentrations of therapeutic agents in preclinical glioma models and patients with either GBM or brain

✉e-mail: catalina.leechang@northwestern.edu; adam.sonabend@northwestern.edu

metastases[6,7,10,12,13,18]. Clinical studies have shown that this technique is safe and effective[10,16,17,19], with ongoing studies further exploring its therapeutic applications.

The failures of recent large randomized clinical trials that evaluated anti-PD-1 immunotherapy (aPD-1) to improve the outcome of patients with newly diagnosed or recurrent GBM[20–22] highlight the importance of developing treatment combinations that reach the tumor cells and elicit effective anti-tumoral immunity. Doxorubicin (and liposomal doxorubicin, DOX), a cytotoxic anthracycline that intercalates into the DNA and inhibits topoisomerase type II[23], has displayed immunogenic effects in several cancers[24,25]. This chemotherapy activates the pathway associated with cyclic GMP-AMP synthase (cGAS) and its downstream signaling effector stimulator of interferon genes (STING) that senses cytosolic DNA[26–28], promoting the expression of type I interferon (IFN) signature[29]. In addition, DOX induces immunogenic cell death in tumor cells[25]. This anthracycline also led to an increase in tumor-infiltrating IFN-γ-expressing CD8+ and CD4+ T cells to sustain anticancer activities in preclinical models of sarcoma, lymphoma, breast, and colon cancer[30,31]. The immunological qualities described for DOX have led to its exploration as an immune-modulating agent to enhance the efficacy of immune checkpoint inhibitors for cancer[24,32]. This effect has been also demonstrated clinically. A multi-cohort clinical trial evaluated treatment responses when different chemotherapy agents preceded aPD-1 therapy in patients with advanced breast cancer. Induction therapy with DOX showed a doubling of objective response rates compared to treatment with aPD-1 alone (35 vs 17%, respectively)[33]. Along with these clinical responses, bulk RNA-seq of tumors exposed to short-term treatment with DOX showed increased tumor expression of several inflammatory gene signatures, including tumor necrosis factor α (TNF-α) signaling through NF-κB and those related to IFN-α and IFN-γ response[33].

Although DOX has been characterized by its immunogenic attributes, the inability to penetrate the CNS limits tumor and immune cells residing in the brain from being affected by these anthracycline-specific immune properties. Additionally, the limited inflow of antibodies into the brain precludes their binding to immune cells in the tumor microenvironment (TME) and peritumoral regions[34–36]. In this context, we investigated the use of LIPU/MB to increase DOX and aPD-1 concentrations in the brain to promote an antitumoral immune response for GBM.

In this work, we report the immune response and pharmacokinetics related to the use of LIPU/MB to enhance the penetration and therapeutic effect of both DOX and aPD-1 in mouse GBM models, as well as in a cohort of 4 recurrent GBM patients. These patients had a skull-implantable ultrasound device (SonoCloud-9; SC9, Carthera, Lyon, France) and received aPD-1 and DOX under single-patient expanded access programs (EAP). They had also previously participated in another clinical trial (NCT04528680)[17]. Additionally, we present preclinical and clinical evidence that supports the use of the LIPU/MB technology in combination with DOX and aPD-1 to induce the upregulation of MHC class I and II in GBM cells and generate IFN-γ responses mediated by myeloid cells and T cells.

## Results

### LIPU/MB increases the penetration of DOX into the human and murine brain

Four patients with GBM, who failed two lines of therapy including standard of care (radiation and temozolomide) and LIPU/MB-based opening of the BBB with concomitant albumin-bound paclitaxel[17], were treated with LIPU/MB with intravenous administration of DOX and aPD-1 at recurrence (Fig. 1a). Treatment included an induction cycle of low-dose liposomal DOX (30 mg) alone, followed by a 2nd dose of DOX and aPD-1 10–14 days later, followed by subsequent cycles of DOX and aPD-1. All treatment cycles that included DOX and aPD-1 were delivered in conjunction with the activation of the SonoCloud-9 device. DOX was administered immediately after sonication, over 30 minutes in all cases. aPD-1 was administered before sonication. All 4 patients underwent surgery after 2–8 cycles of DOX/aPD-1 as clinically indicated tumor debulking (growing mass effect) or biopsy. In this context, our tissue analysis included paired specimens prior to treatment with DOX and aPD-1 (pre-treatment GBM samples) and tumor tissue resected after DOX and aPD-1 delivered with LIPU/MB (on-treatment GBM samples) (Fig. 1a) (Supplementary Table 1). In two of these patients, as per standard neurosurgical technique, we resected and were able to sample peritumoral brain regions that were subjected to LIPU/MB (sonicated) with concomitant administration of DOX and aPD-1 and peritumoral brain regions that were outside the sonication field (non-sonicated). The determination of whether the peritumoral brain was sonicated or non-sonicated was based on the location relative to the ultrasound emitters, as illustrated in Fig. 1b. All surgical samples were acquired 2 days after treatment with liposomal DOX and aPD-1 with concomitant LIPU/MB. The DOX concentrations quantified across all multiple samples from each patient in non-sonicated peritumoral brain regions ranged from 0.0278 μg/g (0.0479 μmol/kg) to 0.112 μg/g (0.194 μmol/kg). On the other hand, DOX concentrations ranged from 0.05 μg/g (0.0872 μmol/kg) to 0.405 μg/g (0.698 μmol/kg) in multiple samples from sonicated peritumoral brain regions (Supplementary Fig. 1). Overall, LIPU/MB led to a 2-fold increase (95% CI of mean:1.406-2.659) in DOX concentration in sonicated peritumoral brain samples compared to non-sonicated peritumoral brain samples two days post-sonication ($P = 0.012$, chi-squared; Fig. 1c).

To further determine the effect of LIPU/MB-based BBB disruption on DOX penetration within the brain, we quantified the concentration of DOX in the brains of mice obtained two days after treatment with DOX delivered with and without LIPU/MB (Fig. 1d). We found increased DOX concentrations in mouse brains after treatment with DOX delivered with LIPU/MB compared to brains obtained from mice treated with DOX without LIPU/MB ($P = 0.003$, t-test; Fig. 1e). Consistently, LIPU/MB led to a 3.92-fold increase (95% CI of mean: 2.015-5.826) in DOX concentration at 48 h with LIPU/MB compared to without LIPU/MB ($P = 0.003$, t-test; Fig. 1f). These results demonstrate the ability of LIPU/MB to increase systemically delivered DOX in the brains of mice and humans.

### DOX upregulates antigen-presenting molecules in GBM cells

DOX has previously been shown to enhance the expression of MHC class I (MHC I) in preclinical models of ovarian and colorectal carcinoma[37,38]. Thus, we investigated whether human GBM samples treated with DOX and LIPU/MB exhibit upregulation of antigen-presenting molecules by GBM cells. We analyzed pre-treatment and on-treatment tumor samples in the cohort of 4 recurrent GBM patients who had treatment with DOX and aPD-1 delivered with LIPU/MB (Fig. 1a). For this purpose, we employed multiplex immunofluorescence to evaluate the abundance of SOX2+ cells (tumor cell marker[39]) expressing HLA-ABC and HLA-DR (Fig. 2a). This analysis was performed in tumor regions delineated by a neuropathologist (Supplementary Fig. 2a). We found increased cell density of SOX2+ HLA-ABC+ cells ($P = 0.0079$, chi-squared; Fig. 2b) and SOX2+ HLA-DR+ cells ($P = 0.024$, chi-squared; Fig. 2c) in GBM samples treated with DOX compared to pre-treatment GBM samples. To evaluate whether the upregulation of HLA-ABC and HLA-DR was a result of tumoral progression irrespective of the therapy with LIPU/MB with DOX and aPD-1, we analyzed paired tumor samples obtained at the first and second recurrence of a cohort of 8 GBM patients that did not receive this immunotherapy (Supplementary Fig. 2b). In this control cohort, we did not find differences in the cell density of SOX2+ HLA-ABC+ cells ($P = 0.2582$, chi-squared; Supplementary Fig. 2c) and SOX2+ HLA-DR+ cells ($P = 0.7491$, chi-squared; Supplementary Fig. 2d) between the first and second recurrent tumors.

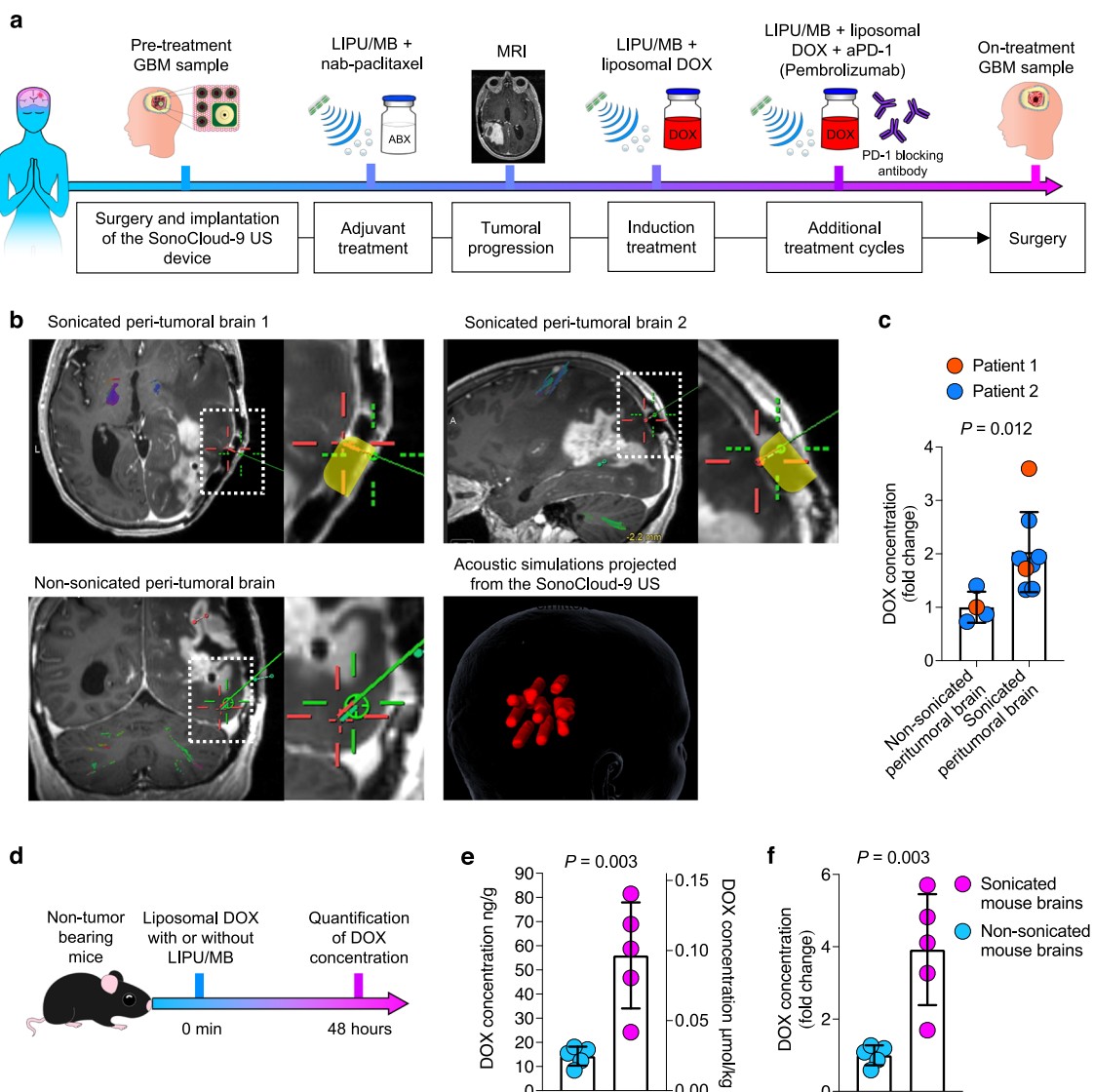

**Fig. 1 | LIPU/MB increases DOX concentrations into the human and murine brain. a** Clinical course of recurrent GBM patients analyzed in this study. Patients underwent surgery for tumor resection (pre-treatment GBM sample) and skull implantation of the SonoCloud-9 ultrasound device for treatment with previous chemotherapy. Induction treatment with liposomal DOX delivered with LIPU/MB was initiated to treat the recurrent tumor followed by additional treatment cycles with both liposomal DOX and aPD-1 delivered by LIPU/MB. The tumor exposed to these therapies was resected (on-treatment GBM sample) and analyzed. **b** Representative MRI scans showing the biopsy sites in the sonicated (top) and non-sonicated (bottom left) peri-tumoral brain determined by whether these sites were immediately deep to an ultrasound emitter, as well as acoustic simulation of the ultrasound beams for this case (bottom right). A magnification of each MRI scan is provided to show the position of the ultrasound emitter implanted in the skull colored in yellow. The biopsy sites are indicated in green by the neuronavigator for both sonicated and non-sonicated peritumoral brain samples. **c** Bar plot representing the fold change in DOX concentration in non-sonicated ($n = 4$ brain samples) and sonicated peritumoral brain tissues ($n = 8$ brain samples) from 2 GBM patients. A mixed effects model was constructed considering sonication as a fixed effect and patients as a random effect influencing the fold change in DOX concentration. $P$ value was obtained by a chi-squared test of the likelihood ratio test of the full model with sonication as a fixed effect against the model without the effect. **d** Therapeutic scheme employed to determine the DOX concentrations in the brains of non-tumor-bearing mice after LIPU/MB obtained 48 hrs post-treatment. **e, f** Bar plots showing the concentration values (**e**) and fold change (**f**) of DOX in mouse brains sonicated and non-sonicated obtained 48 hrs post-treatment. $n = 5$ mice per group. Data are presented as mean ± SEM in **c**, **e** and **f**. Source data are provided as a Source Data file. Each dot represents one mouse in **e** and **f**. $P$ values in **e** and **f** derived from two-sided unpaired $t$-tests.

Next, to investigate whether the concentrations of DOX achieved in the peritumoral brain would modulate MHC I expression in GBM cells, we treated the patient-derived xenograft (PDX) cells, GBM6 and GBM63, with a range of DOX concentrations (0 to 9.6 µM). We included treatments with human IFN-γ at 1 and 10 ng/ml concentrations as positive controls. We also evaluated temozolomide for comparison to determine whether the induction of MHC I expression was a DOX-associated effect. GBM6 and GBM63 cells were exposed to DOX, temozolomide, or IFN-γ for 5 h, washed extensively, incubated for additional 72 h in drug-free media followed by surface marker analysis

using flow cytometry (Fig. 2d). We found that the DOX concentration range of 0.15-1.2 µM upregulated HLA-ABC and HLA-DR expression relative to the no treatment group in GBM6 cells (adjusted $P < 0.05$, one-way ANOVA; Fig. 2e). In GBM63, we determined that the DOX concentration range of 0.15-0.6 µM upregulated the expression of HLA-ABC and HLA-DR (adjusted $P < 0.05$, one-way ANOVA; Fig. 2f). Notably, these DOX immunogenic concentration ranges overlapped with the concentrations reached in the sonicated peritumoral brain regions (Fig. 1c). In contrast, temozolomide did not upregulate these antigen-presenting molecules in PDX cells, underscoring a specific

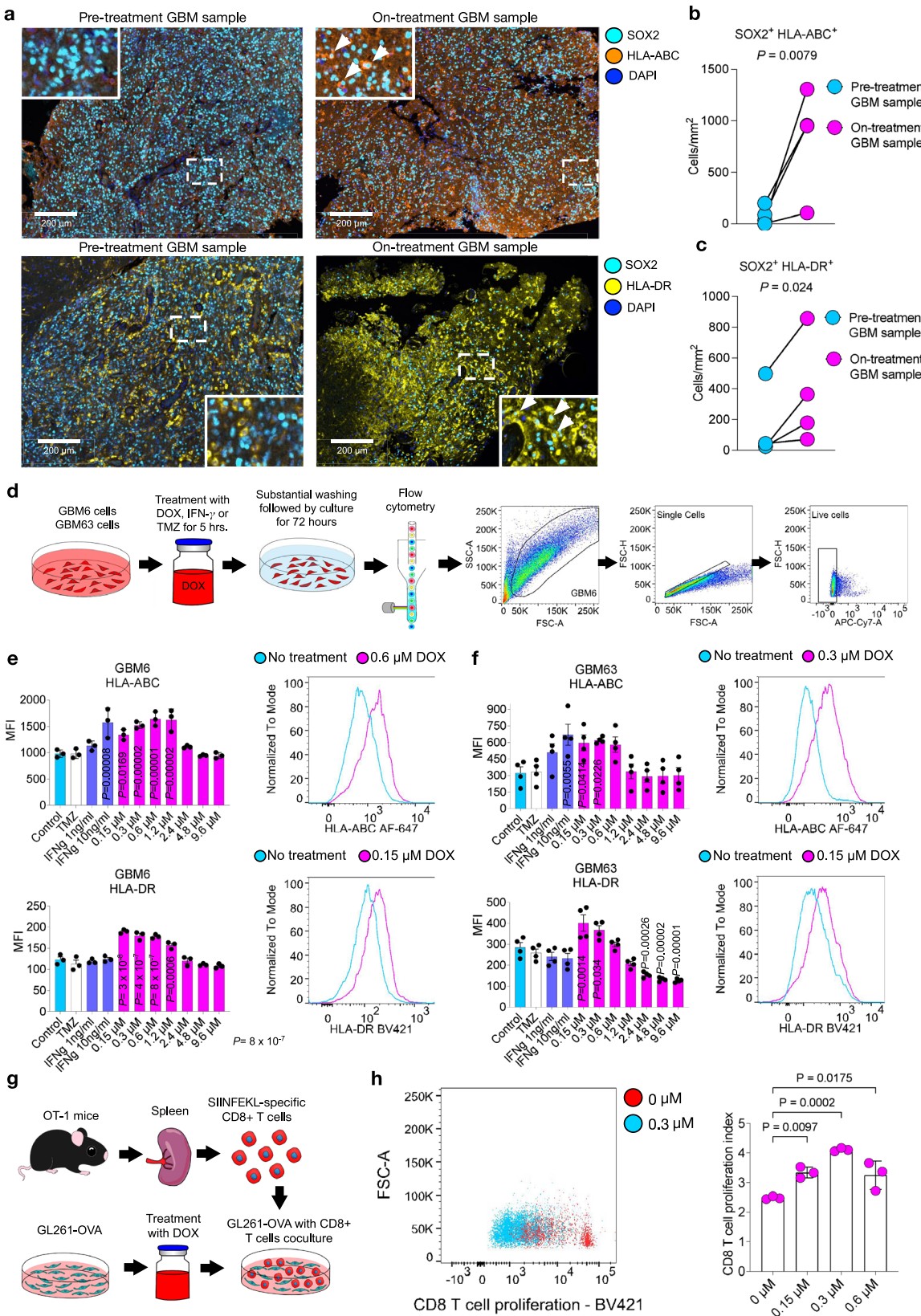

immunogenic effect of DOX (Fig. 2e and f). In the murine GBM models GL261 and CT-2A, a similar DOX concentration range upregulated H-2K$^b$, the murine ortholog of HLA I, (comparison to the no treatment group $P < 0.05$, one-way ANOVA; Supplementary Fig. 2e).

Next, we conducted a T cell proliferation assay using CD8$^+$ T cells from OT-1 mice (Fig. 2g), which are known to recognize the OVA peptide presented by MHC I. Upon co-culture with GL261-OVA cells treated with DOX at various concentrations, we observed an increase in T cell proliferation ($P < 0.05$, one-way ANOVA; Fig. 2h).

Overall, these results show that DOX increases the expression of antigen-presenting molecules in tumor cells of GBM patients, GBM PDX lines, and murine GBM models. Furthermore, the upregulation of

**Fig. 2 | DOX upregulates antigen-presenting molecules in human GBM cells.** **a** Representative multiplex immunofluorescence images illustrating SOX2⁺, HLA-ABC⁺, and HLA-DR⁺ cells in pre-treatment and on-treatment GBM samples. **b, c** Dot plot showing the cell density of SOX2⁺ HLA-ABC⁺ (**b**) and SOX2⁺ HLA-DR⁺ (**c**) cells in pre-treatment and on-treatment GBM samples. $n = 4$ paired GBM samples. A mixed effects model was constructed considering DOX+aPD-1 treatment as a fixed effect and patients as a random effect influencing SOX2⁺ HLA-ABC⁺ and SOX2⁺ HLA-DR⁺ cell densities. $P$ values were obtained by a chi-squared test of the likelihood ratio test of the full model including DOX+aPD-1 treatment as a fixed effect against the model without the fixed effect. **d** Schematic of the flow cytometry experiment and gating strategy performed to assess the effect of different DOX concentrations on the expression of antigen-presenting molecules in the PDX cell lines, GBM6 and GBM63. **e, f** (left) Bar plots showing the expression of HLA-ABC and HLA-DR assessed as MFI values in GBM6 (**e**) and GBM63 (**f**). $n = 3$ biological replicates per

condition. (right) Representative histograms showing the expression of HLA-ABC and HLA-DR in control and 0.15-0.6 μM DOX-treated cells. Histograms are representative data from three biological replicates. **g** Experimental setup of CD8⁺ T cells isolated from OT-1 mice co-cultured with GL261-OVA cells treated with varying concentrations of DOX to assess T cell activation and proliferation. **h** (left) Scatter plot displaying CD8⁺ T cell proliferation after co-culture with GL261-OVA cells, untreated (red dots) or treated with 0.3 μM DOX (blue dots). (right) Bar graph showing the CD8⁺ T cell Proliferation Index for untreated and DOX-treated GL261-OVA cells at different concentrations (0, 0.1, 0.3, and 0.6 μM). The Proliferation Index is calculated based on the fluorescence intensity decay of BV421, indicating the division of T cells. $n = 3$ biological replicates per condition. $P$ values were obtained by one-way ANOVA with post hoc Dunnett's multiple comparisons test in **e**, **f** and **h**. Source data are provided as a Source Data file. Data are presented as mean ± SEM in **e**, **f** and **h**.

MHC I on tumor cells by DOX facilitates more efficient antigen presentation to T cells, leading to their activation and proliferation as suggested by the OVA system.

## DOX plus LIPU/MB modulates the phenotype of GBM-associated myeloid cells

Tumor-associated macrophages and microglia are the most abundant immune components of the tumor microenvironment and constitute approximately 30-50% of the mass of GBM[40]. These cells have been shown to display a wide variety of phenotypes in gliomas[41]. In this context, we investigated the effect of DOX and concomitant LIPU/MB on the phenotype of these cells. Using the GL261 intracranial murine GBM model, we evaluated different doses of liposomal DOX (1, 2, and 5 mg/kg) delivered with and without LIPU/MB. Following treatment, microglia, macrophages, and T cells residing in the brain and immune cells from the blood were analyzed using flow cytometry (Fig. 3a and Supplementary Fig. 3). Though we did not notice differences in the percentages of microglia and macrophages across treatment conditions (Supplementary Fig. 4a), we observed that 5 mg/kg of DOX with concomitant LIPU/MB led to increased production of IFN-γ by microglia (gated on CD45⁺ᵈⁱᵐ CD11b⁺) ($P < 0.0001$, one-way ANOVA, Fig. 3b and Supplementary Fig. 4b), as well as in monocyte-derived macrophages ($P < 0.05$, one-way ANOVA, Fig. 3c and Supplementary Fig. 4c). No other doses of DOX nor the 5 mg/kg dose delivered in the absence of LIPU/MB elicited an increased production of IFN-γ by GBM-associated microglia and macrophages (Fig. 3d). Additionally, the visualization of cells in tSNE plots revealed that regardless of whether they were CD45⁺ or CD45⁺ᵈⁱᵐ, CD11b⁺ cells (myeloid cells) were the predominant source of IFN-γ expression relative to other cells in the group of 5 mg/kg of liposomal DOX + LIPU/MB (Fig. 3e). We also evaluated the production of TNF-α and IL-1β by GBM-associated microglia and macrophages. However, we did not observe differences in the production of TNF-α (Supplementary Fig. 4c) and IL-1β (Supplementary Fig. 4d) by microglia or macrophages between groups receiving different DOX doses (Supplementary Fig. 4e, f).

We also investigated the immunomodulatory effect of different doses of liposomal DOX in the lymphocyte compartment. Twelve days after DOX treatment initiation, we did not find significant differences in the percentages of total CD4⁺ T cells (Supplementary Fig. 5a) nor in the proportion of CD4⁺ T cells expressing IFN-γ, TNF-α, and IL-1β (Supplementary Fig. 5b). Similarly, the percentages of CD8⁺ T cells (Supplementary Fig. 5c) and those expressing IFN-γ, TNF-α, IL-1β, and granzyme b (GZMb) (Supplementary Fig. 5d) were similar between groups. When analyzing the peripheral immune cells among groups, the percentages of monocytes (Supplementary Fig. 6a), CD4⁺ T cells (Supplementary Fig. 6b), and CD8⁺ T cells (Supplementary Fig. 6c) producing IFN-γ, TNF-α, IL-1β, and GZMb were similar between groups. These results indicate that LIPU/MB, with concomitant administration of DOX, can modulate the phenotype of tumor-associated macrophages and microglia in murine GBM.

Considering the robust IFN-γ phenotype exhibited by microglia and monocyte-derived macrophages induced by liposomal DOX plus LIPU/MB, we investigated the expression of H-2Kᵇ (MHC I) and PD-L1 as these proteins are upregulated in response to IFN-γ[42,43]. We found that H-2Kᵇ (MHC I) was upregulated by microglia and monocyte-derived macrophages in the mouse groups that were treated with 5 mg/kg of liposomal DOX regardless of whether the LIPU/MB was used to deliver the chemotherapy ($P < 0.05$, one-way ANOVA; Supplementary Fig. 7a). Likewise, PD-L1 was upregulated in the groups treated with 5 mg/kg of liposomal DOX with and without LIPU/MB compared to the control groups and those treated with lower doses of the anthracycline ($P < 0.05$, one-way ANOVA; Supplementary Fig. 7b).

To validate these results, we employed the human microglia cell line, HMC3, and investigated the production of IFN-γ, and expression of HLA-ABC and PD-L1 using a range of DOX concentrations (Supplementary Fig. 8a). By gating on single live cells, we corroborated that at DOX concentrations ranging from 0.15 to 2.4 μM, this anthracycline promoted the production of IFN-γ ($P < 0.05$, one-way ANOVA; Supplementary Fig. 8b), increased expression of HLA-ABC ($P < 0.0001$, one-way ANOVA; Supplementary Fig. 8c) and increased expression of PD-L1 ($P < 0.0001$, one-way ANOVA; Supplementary Fig. 8d) compared to the cells that did not receive any treatment. In light of these results, we evaluated whether DOX would upregulate the expression of other interferons. Thus, we evaluated the expression of *IFNG*, *IFNA1*, and *IFNB1* by treating our microglia cell line with DOX for 5 h, cultured with drug-free media followed by gene expression analysis (Supplementary Fig. 8e). We found a significant induction of *IFNG* in response to DOX treatment (Supplementary Fig. 8f) which is consistent with the flow cytometry data in glioma-associated microglia and HMC3 cells. In addition, we found variability in the expression of *IFNA1* with certain DOX concentrations prompting a modest increase, though not reaching statistical significance (Supplementary Fig. 8g). Interestingly, we found that DOX treatment upregulated *IFNB1* in our microglial cell line (Supplementary Fig. 8h). This finding complements the initial observation regarding type II IFN secretion by myeloid cells.

Next, we investigated whether these preclinical observations were reproducible in GBM patients treated with liposomal DOX delivered with LIPU/MB. To this end, we analyzed pre-treatment and on-treatment tumor samples from our GBM patient cohort to assess the expression of IFN-γ⁺ and HLA-ABC⁺ by TMEM119⁺ (microglial marker) and CD163⁺ (myeloid cell marker) cells using multiplex immunofluorescence. We observed that TMEM119⁺ IFN-γ⁺ cells were more abundant in on-treatment GBM samples compared to pre-treatment GBM samples (Fig. 3f). Cell density quantification of TMEM119⁺ IFN-γ⁺ cells between pre-treatment and on-treatment GBM samples confirmed these findings ($P = 0.0021$, chi-squared; Fig. 3g). We isolated myeloid cells (TMEM119⁺ and CD163⁺ cells) from both pre-treatment and on-treatment GBM samples and projected these immune cells into a PCA plot (Fig. 3h). We observed that the expression of IFN-γ was more prevalent in TMEM119⁺ cells from tumors treated with DOX plus LIPU/

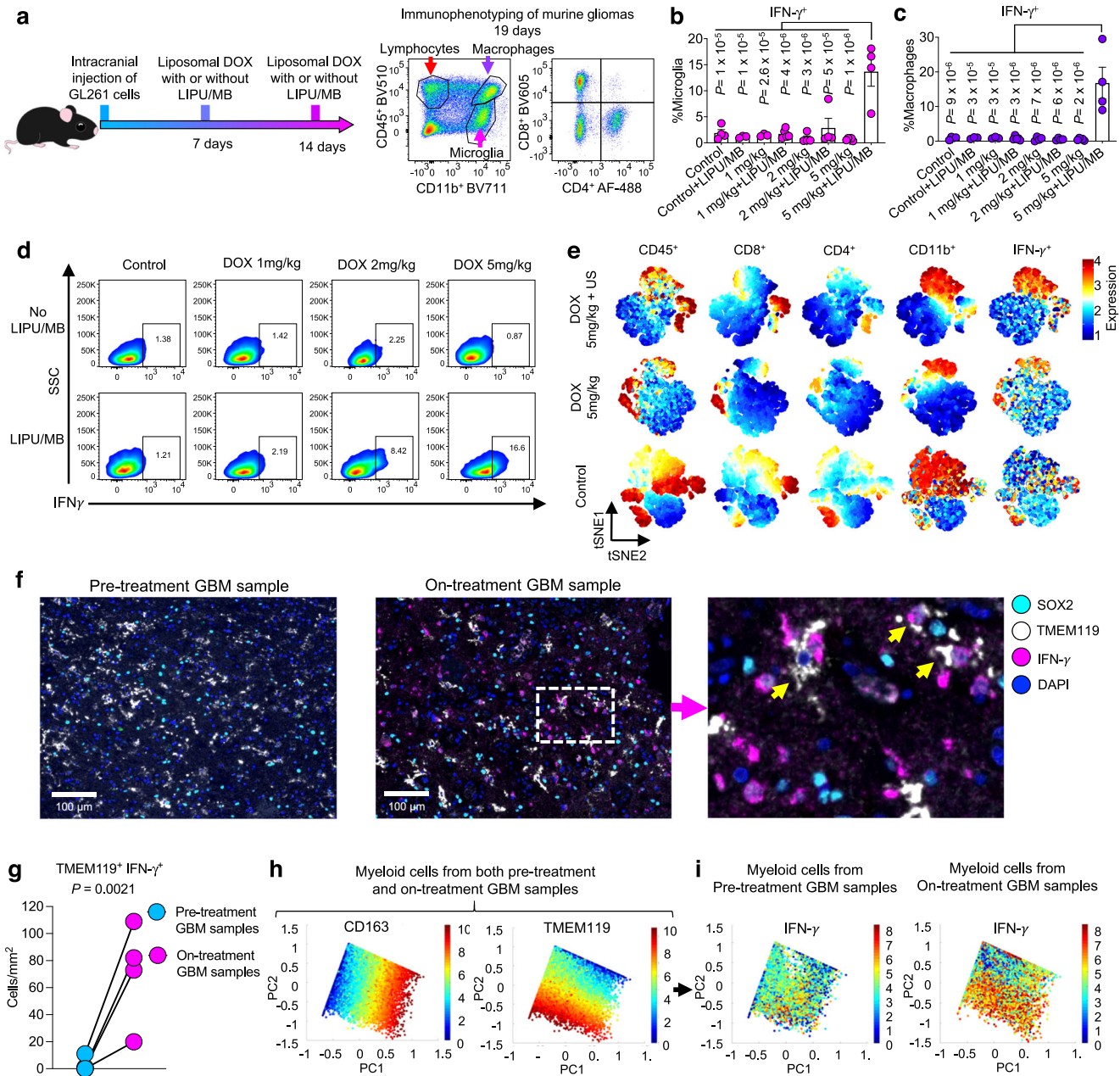

**Fig. 3 | Delivery of liposomal DOX via LIPU/MB induces an IFN-γ phenotype in GBM-infiltrating myeloid cells. a** (left) Therapeutic scheme employed to treat GL261-bearing mice with liposomal DOX with and without LIPU/MB. (right) Flow cytometry plots showing the immune cell populations analyzed in murine GBM. **b**, **c** Bar plots showing the percentage of microglia (**b**) and macrophages (**c**) producing IFN-γ+ from groups treated with different doses of liposomal DOX (1, 2, and 5 mg/kg) with or without LIPU/MB. *n* = 3 mice for control+LIPU/MB and 1 mg/kg; *n* = 4 mice for control, 2 mg/kg, 2 mg/kg+LIPU/MB, and 5 mg/kg+LIPU/MB; and *n* = 5 mice for 1 mg/kg+LIPU/MB and 5 mg/kg. Samples were derived from biologically independent mice from 1 experiment. **d** Representative scatter plots from each treatment group in **b** and **c** showing the gating strategy to determine IFN-γ+ myeloid cells. **e** tSNE plots showing cell distributions and marker expressions (CD45, CD4, CD8, CD11b, and IFN-γ) with color indicating expression levels. Each tSNE plot shows the aggregate of live single cells from all mice from a treatment group in **b** and **c**. **f** Multiplex immunofluorescence images of pre-treatment and on-treatment GBM samples showing SOX2+, TMEM119+, and IFN-γ+ cells. **g** Dot plots

showing the TMEM119+ IFN-γ+ cell density in pre-treatment and on-treatment GBM samples. *n* = 4 paired tumors. One dot represents one tumor. A mixed effects model was constructed considering treatment as a fixed effect and patients as a random effect influencing TMEM119+ IFN-γ+ cell density. *P* value was obtained by a chi-squared test of the likelihood ratio test of the full model against the model without the fixed effect. **h** PCA plots showing CD163 and TMEM119 cells from multiplex immunofluorescence data from both pre-treatment and on-treatment GBM samples. **i** PCA plots showing the production of IFN-γ+ by TMEM119 and CD163 cells in pre-treatment and on-treatment GBM samples. *n* = 116,717 myeloid cells in pre-treatment GBM samples, *n* = 95,719 myeloid cells in on-treatment GBM samples. The color key indicates the expression levels. Each dot represents one mouse in **b** and **c**. *P* values in **b** and **c** were derived from one way-ANOVA with post hoc Tukey's multiple comparisons test. Source data are provided as a Source Data file. Data shown as mean ± SEM in **b** and **c**. The gating strategy is detailed in Supplementary Fig. 3.

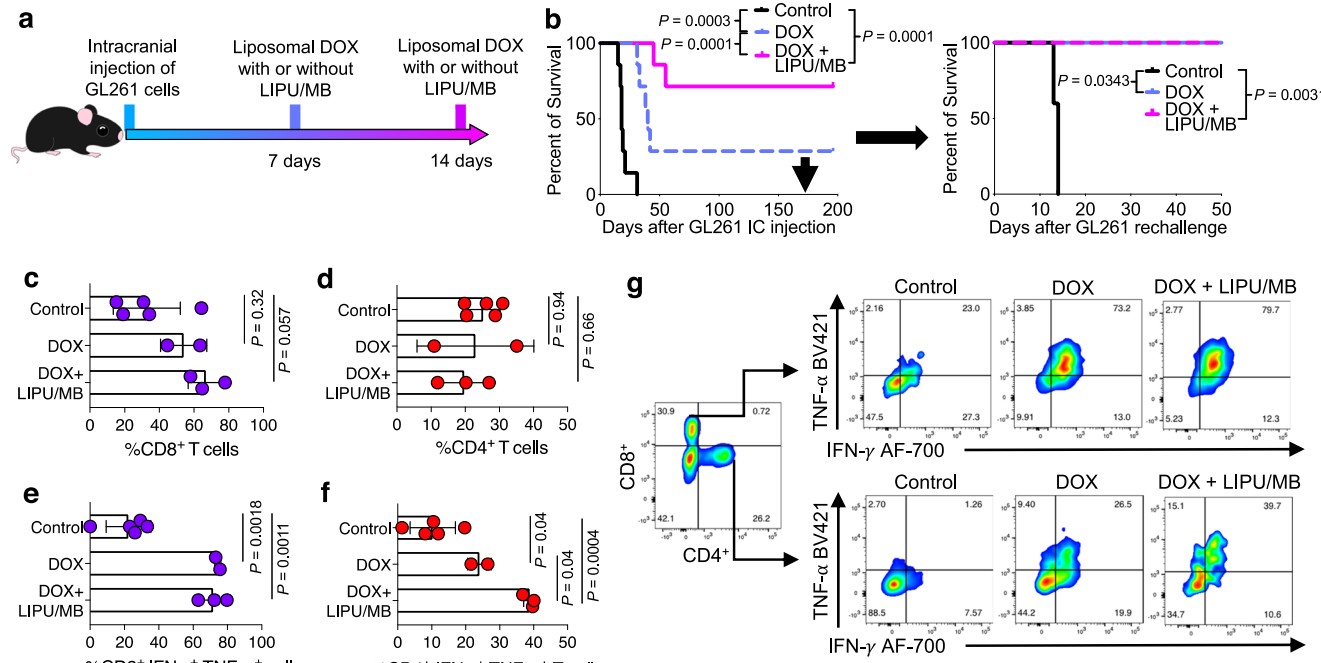

**Fig. 4 | LIPU/MB-based BBB opening enhances the efficacy of liposomal DOX in murine GBM models. a** Therapeutic scheme employed to treat GL261-bearing mice with liposomal DOX with LIPU/MB. **b** Kaplan-Meier curves showing survival of GBM-bearing mice treated with liposomal DOX with and without LIPU/MB (*n* = 7 mice per group) as well as the tumor rechallenge experiment performed in long-term survivors (control *n* = 7 mice, DOX *n* = 2 mice, DOX + LIPU/MB *n* = 5 mice). *P* values by log-rank test. **c**, **d** Percentage of CD4[+] (**c**) and CD8[+] (**d**) T cells in long-term survivors and control groups. **e**, **f** Percentage of CD4[+] (**e**) and CD8[+] (**f**) T cells expressing IFN-γ and TNF-α in the brain of long-term survivor mice after tumor re-challenge in the contralateral brain hemisphere. *n* = 5 mice for the control group, *n* = 2 mice for the DOX group, *n* = 3 mice for the DOX + LIPU/MB group. Dots represent biologically independent mice from 1 experiment. **g** Representative flow cytometry plots of CD4[+] and CD8[+] IFN-γ[+] TNF-α[+] cells in different treatment groups. Source data are provided as a Source Data file. Data are presented as mean ± s.d. (**c**–**f**). *P* values by one way-ANOVA with post hoc Tukey's multiple comparisons test in **c**–**f**.

MB compared to those TMEM119[+] cells from pre-treatment GBM samples (Fig. 3i). With regards to IFN-γ[+] CD163[+] cells, we observed a non-significant trend towards increased cell density post-treatment (*P* = 0.081; chi-squared, Supplementary Fig. 9a). Analysis of TMEM119[+] and CD163[+] IFN-γ[+] cell densities in the GBM control cohort revealed no significant differences between the first and second recurrences (Supplementary Fig. 9b).

We also found an increased number of TMEM119[+] HLA-ABC[+] (*P* = 0.03229, chi-squared; Supplementary Fig. 9c) but not in CD163[+] HLA-ABC[+] cells (*P* = 0.066, chi-squared; Supplementary Fig. 9d) in on-treatment GBM samples relative to pre-treatment GBM samples. Our control cohort analysis revealed no significant differences in the cell densities of TMEM119[+] HLA-ABC[+] and CD163[+] HLA-ABC[+] (Supplementary Fig. 9e). With regards to HLA-DR[+] expression by TMEM119[+] and CD163[+] cells, we did not find differences between groups (*P* = 0.071 for TMEM119[+] and *P* = 0.16 for CD163[+], chi-squared, Supplementary Fig. 9f).

### DOX-related immune modulation contributes to its anti-tumoral response in murine gliomas

Considering the immune effects of DOX after delivery with LIPU/MB on GBM-associated myeloid cells, we evaluated whether this drug delivery strategy can enhance the efficacy of liposomal DOX in preclinical GBM models. Before testing DOX, we evaluated whether LIPU/MB alone had any effect on survival in the murine GBM models, and consistent with previous literature[7], we did not observe differences in survival related to sonication in GL261 (*P* = 0.27, log-rank test; Supplementary Fig. 10a) or CT-2A (*P* = 0.14, log-rank test; Supplementary Fig. 10b). We then injected GL261 cells intracranially in C57BL/6 mice and treated them with DOX delivered with and without LIPU/MB (Fig. 4a). Two treatments of DOX delivered with LIPU/MB led to long-term survival (mice

surviving more than 100 days following intracranial injection of tumor cells) of 75% of mice bearing intracranial tumors compared to 28.5% of mice treated with DOX without LIPU/MB (*P* = 0.001, log-rank test, Fig. 4b). To evaluate whether DOX treatment leads to long-lasting anti-tumoral immune surveillance, we re-challenged the long-term survivor mice by re-injecting GL261 cells in the contralateral hemisphere of the brain, as well as to age-matched control mice. Only mice previously treated with DOX regardless of LIPU/MB survived upon intracranial tumor rechallenge (*P* = 0.0343 for DOX vs control, *P* = 0.0031 for DOX + LIPU/MB vs control; Fig. 4b), suggesting a long-standing anti-tumoral immunity elicited by treatment with DOX.

Considering the long-lasting anti-tumoral immunity we observed, we performed immunophenotyping of T cells from long-term survivors. To provide additional stimulation for immune response in the brain prior to the analysis, we injected GBM cells into long-term survivors (2[nd] rechallenge) and age-matched controls. No treatment was provided after the tumor re-challenge. We observed a trend for an increased percentage of CD8[+] T cells in the brain of long-term survivors that were initially treated with liposomal DOX and LIPU/MB compared to control mice (*P* = 0.057, one-way ANOVA; Fig. 4c) and no significant changes in the percentage of CD4[+] T cells among groups (*P* = 0.94 for DOX vs. control, *P* = 0.66 for LIPU/MB + DOX vs. control, one-way ANOVA; Fig. 4d). However, CD8[+] T cells from the long-term survivor mice treated with liposomal DOX with and without LIPU/MB exhibited an IFN-γ[+] TNF-α[+] phenotype (*P* = 0.0018 for DOX vs. control, *P* = 0.0011 for LIPU/MB + DOX vs. control, one-way ANOVA; Fig. 4e). Although there was a higher percentage of CD4[+] IFN-γ[+] TNF-α[+] T cells from mice treated with DOX alone compared to the control group (*P* = 0.04, one-way ANOVA; Fig. 4f), the greatest percentage of T cells characteristic of a Th1 phenotype was exhibited by long-term survivors treated with DOX delivered by LIPU/MB (*P* = 0.04 for LIPU/MB + DOX

vs DOX alone, $P = 0.0004$ for LIPU/MB + DOX vs DOX alone, one-way ANOVA; Fig. 4f, g).

Lastly, we evaluated whether a tumor-specific response by peripheral blood mononuclear cells (PBMCs) could be elicited with DOX treatment. For this, we treated GL261-bearing mice with different doses of DOX followed by isolation of PBMCs. Then, we exposed PBMCs isolated from the blood of tumor-bearing mice to GL261 tumor lysate. By employing an ELISpot assay, we found that PBMCs exposed to the highest dose of liposomal DOX (5 mg/kg) showed increased secretion of IFN-γ ($P < 0.04$, one-way ANOVA; Supplementary Fig. 11) suggesting a stronger activation of PBMCs against tumoral antigens in the context of exposure to the highest dose of DOX.

We expanded our investigation to include the CT-2A murine glioma model. This treatment approach did not yield a significant survival advantage for LIPU/MB with DOX, compared to DOX alone in CT-2A-bearing mice (Supplementary Fig. 10c). This indicated that while LIPU/MB significantly enhances DOX efficacy in the GL261 model, it may require a combination with other modalities to achieve similar results in CT-2A models.

## LIPU/MB increases the penetration and efficacy of aPD-1
We then investigated whether LIPU/MB could enhance the concentration of immune checkpoint blockade antibodies in the brain and their related efficacy. We treated non-tumor bearing mice with the human aPD-1, nivolumab, followed by LIPU/MB and fluorescein IV injections for visualization of areas of BBB disruption that were further dissected and analyzed with a nivolumab-specific ELISA (Fig. 5a). As control groups, we included a group that received aPD-1 and fluorescein but did not receive LIPU/MB, and another group that received LIPU/MB followed by fluorescein but without aPD-1 (Fig. 5b). Whereas aPD-1 concentrations in plasma were similar over time (Fig. 5c), we found that the groups that received nivolumab and LIPU/MB had increased concentrations of the human aPD-1 in sonicated areas compared to the group that received aPD-1 without LIPU/MB (1 hour: 6.3-fold increase, 4 h: 6.6-fold increase relative to the aPD-1 without LIPU/MB group, $P < 0.01$, one-way ANOVA; Fig. 5d). Of note, we did not find differences in aPD-1 concentration between murine brains that were extracted 1 and 4 h after LIPU/MB suggesting that the penetration of this immune checkpoint antibody occurs within the first hour of BBB opening.

Next, we investigated the antitumoral effects of enhanced delivery of aPD-1 by LIPU/MB in the CT-2A murine GBM model and observed a modest increase in survival that included a subset of long-term survivor mice treated with aPD-1 plus LIPU/MB compared to mice treated with the isotype IgG2a antibody ($P = 0.047$, log-rank test) and a trend that did not reach significance, when compared to the group treated with only aPD-1 ($P = 0.051$, log-rank test; Fig. 5e).

To determine the added value of LIPU/MB in increasing aPD-1 concentrations in the brain of GBM patients, we measured pembrolizumab levels in plasma, tumor, and peritumoral brain in two GBM patients that underwent sonication with the SC9 ultrasound device concomitantly with pembrolizumab administration for which we had available sonicated and non-sonicated peritumoral brain samples. By employing a pembrolizumab-specific ELISA, we determined that the aPD-1 concentration in the serum of these GBM patients 2 days after treatment was 39.45 and 45 μg/mL (Fig. 5f), which was in the range reported in previous clinical studies[34,44]. Additionally, we evaluated the concentrations of aPD-1 delivered with LIPU/MB in different areas of resected tumors. As negative controls, we also assessed aPD-1 concentration in pre-treatment tumor samples from the same GBM patients. We found a mean aPD-1 concentration of 7.5 μg/g (95% CI of mean: 3.53-11.64 μg/g) in the treated tumor specimens (Fig. 5g). To further assess the ability of LIPU/MB to increase the concentration of aPD-1 in the brain, we measured aPD-1 levels in peritumoral regions that were subjected to sonication and in those peritumoral regions not subjected to sonication. In the two patients for whom we were able to

acquire peritumoral brain samples that were sonicated as well as non-sonicated during surgery 2 days after LIPU/MB with concomitant DOX and aPD-1 (Fig. 1b). In these samples, we observed an increase of aPD-1 concentration in the sonicated peritumoral brain regions compared to the non-sonicated peritumoral brain (Fig. 5g). Specifically, there was a 2-fold increase in aPD-1 concentration in peritumoral brain regions that were covered by the sonication field of the SC9 ultrasound device relative to those samples obtained from regions outside the sonication field (Fig. 5h).

In this exploratory analysis, we observed an increase in aPD-1 concentration in the sonicated peritumoral brain regions compared to the non-sonicated peritumoral brain (Fig. 5g). The data suggest a 2-fold increase in aPD-1 concentration in peritumoral brain regions influenced by the sonication field of the SC9 ultrasound device, relative to those samples collected from regions outside the sonication field (Fig. 5h).

In sum, the integration of these preclinical and clinical results shows the ability of LIPU/MB to enhance the penetration of therapeutic antibodies into the human and murine brains.

## LIPU/MB delivery of DOX enhances efficacy of aPD-1 in GBM
To investigate whether LIPU/MB can enhance the efficacy of DOX and aPD-1, we treated mice bearing intracranial GL261 cells with DOX in combination with aPD-1 delivered with and without LIPU/MB (Fig. 6a). We investigated the tolerance of this therapeutic combination through monitoring the weight of the treated healthy mice, as well as histological analysis of their brains. In these mice, we did not observe any signs of toxicity (Supplementary Fig. 12a, b). Next, we investigated the therapeutic effect of combining aPD-1 with DOX without LIPU/MB in GBM-bearing mice. Whereas we did not find any effect on survival with aPD-1, the combination of liposomal DOX and aPD-1 extended survival compared to the control group ($P = 0.0001$, log-rank test; Fig. 6b). Long-term survivors treated with DOX with and without aPD-1 remained alive after rechallenging with an intracranial injection of GL261 cells compared to age-matched controls that succumbed due to tumor growth.

We then evaluated whether LIPU/MB increases the therapeutic efficacy of DOX and aPD-1 in murine gliomas. DOX delivered with LIPU/MB resulted in the cure of 75% of mice compared to the control group ($P < 0.0001$, log-rank test) and the group treated with DOX without LIPU/MB ($P = 0.0271$, log-rank test; Fig. 6c). The addition of aPD-1 to DOX plus LIPU/MB showed similar results in generating the same percentage of long-term survivors suggesting that there was limited room for increased efficacy when adding aPD-1 (Fig. 6c). Thus, we evaluated the proposed combination in a different GBM model. We treated CT-2A-bearing mice with the same therapeutic scheme used previously for GL261 cells (Fig. 6a). In this model, we found that the combination of DOX + aPD-1 without LIPU/MB extended survival compared to the DOX group ($P = 0.0018$, log-rank test) and the aPD-1 group ($P = 0.0378$, log-rank test; Fig. 6d). The addition of aPD-1 to DOX plus LIPU/MB led to therapeutic enhancement denoted as long-term survival percentage of 80% relative to mice treated with DOX plus LIPU/MB without immunotherapy ($P < 0.0001$, log-rank test; Fig. 6d). We also observed a trend for improved efficacy of DOX and aPD-1 when combined with LIPU/MB relative to the group treated with DOX + aPD-1 without LIPU/MB ($P = 0.0681$, log-rank test; Fig. 6d).

Considering the induction of an IFN-γ⁺ phenotype in GBM-infiltrating myeloid cells following delivery of DOX with LIPU/MB (Figs. 3b and 3c), we evaluated whether the depletion of microglia and bone-marrow-derived macrophages using a CSF1 inhibitor (PLX3397) would impair the therapeutic efficacy of the proposed treatment combination. After confirming that PLX3397 depleted brain myeloid cells (Supplementary Fig. 13a), we evaluated whether PLX3397 alone would have any effect on survival in our murine GBM model. In CT-2A-bearing mice, we did not observe differences in survival related to

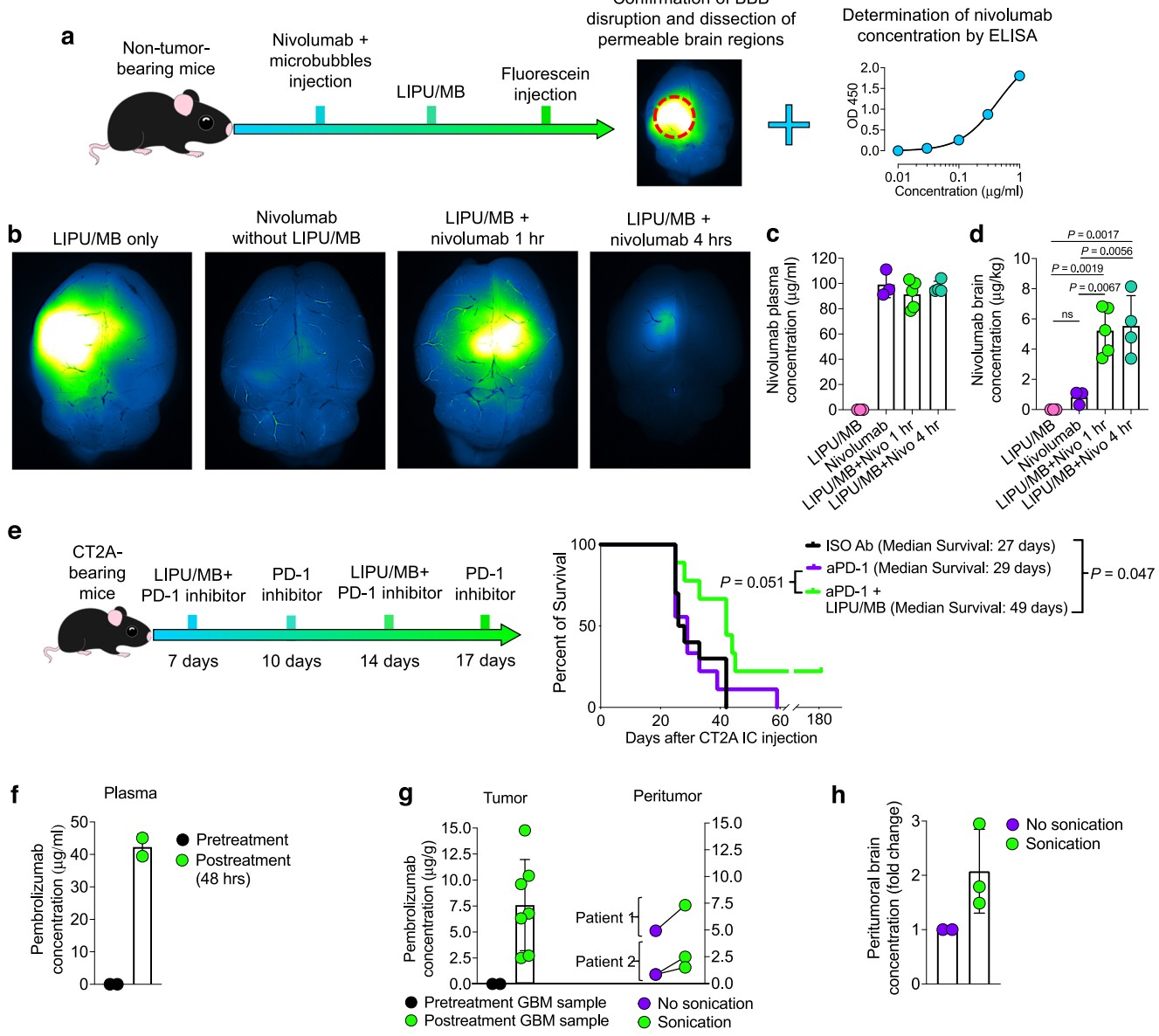

**Fig. 5 | LIPU/MB increases PD-1 antibody brain concentration. a** Scheme showing the experimental procedure for nivolumab concentrations measurement via ELISA in BBB-intact brains, comparing sonicated to non-sonicated mice. **b** Fluorescent images of mouse brains from 4 groups that received fluorescein, nivolumab, with and without LIPU/MB obtained after 1 and 4 h. **c, d** Bar plots showing the nivolumab concentration in plasma (**c**) and the brain (**d**) in the following groups: LIPU/MB, nivolumab without LIPU/MB, nivolumab with LIPU/MB after 1 hour, and nivolumab with LIPU/MB after 4 h. *n* = 3 mice for LIPU/MB and nivolumab without LIPU/MB, *n* = 5 mice for LIPU/MB+nivo 1 h, and *n* = 4 mice for LIPU/MB+nivo 4 hr. All samples were derived from biologically independent mice from 1 experiment. **e** (left) Therapeutic scheme and (right) Kaplan-Meier curve of CT-2A-bearing mice treated with aPD-1 delivered with and without LIPU/MB. (*n* = 10 mice treated with ISO Ab, *n* = 9 mice treated with aPD-1 group, *n* = 9 mice treated

with LIPU/MB + aPD-1). *P* values by Log-rank test. **f** Bar plot showing the pembrolizumab levels in plasma before and 48 h after immunotherapy. *n* = 2 pretreatment and 2 post-treatment plasma samples from 2 GBM patients. **g** (left) Bar plot showing the pembrolizumab concentration in pre-treatment (*n* = 2) and on-treatment (*n* = 7) GBM samples obtained 48 h after immunotherapy administration. (right) Dot plot representing the concentration of pembrolizumab in sonicated and non-sonicated peritumoral brain regions obtained after 48 h of immunotherapy administration. *n* = 2 non-sonicated and 3 sonicated peritumoral brain samples from 2 GBM patients. **h** Bar plot representing pembrolizumab concentration fold change in peritumoral regions. *n* = 3 sonicated and 2 non-sonicated peritumoral brain samples from 2 GBM patients. *P* values in **c** and **d** were derived from one-way ANOVA with post hoc Tukey's multiple comparisons test. Source data are provided as a Source Data file. Data are presented as mean ± s.d. in **c**, **d**, **f**, **g** and **h**.

PLX3397 (*P* = 0.7363, log-rank test; Supplementary Fig. 13b). Next, we treated a group of CT-2A-bearing mice with PLX3397 and another group of CT-2A-bearing mice with control diet and employed the same therapeutic strategy involving DOX and aPD-1 delivered with LIPU/MB (Fig. 6e). Notably, the depletion of GBM-infiltrating myeloid cells by PLX3397 decreased the survival of CT-2A-bearing mice compared to the mouse group fed with control diet (*P* = 0.0422, log-rank test; Fig. 6f). Considering previous evidence showing the ability of inflammatory stimuli to induce memory by brain resident myeloid cells[45], we

evaluated whether the depletion of myeloid cells would impair the survival of long-term survivors following rechallenge with CT-2A. Thus, we partitioned the group of long-term survivors previously treated with the control diet, DOX, and aPD-1 delivered with LIPU/MB and started treating them with either PLX3397 or a control diet. Interestingly, we found that mice fed with the control diet survived the tumor rechallenge, whereas mice treated with PLX3397 exhibited worse survival following the tumor rechallenge (*P* = 0.0169, log-rank test; Fig. 6f).

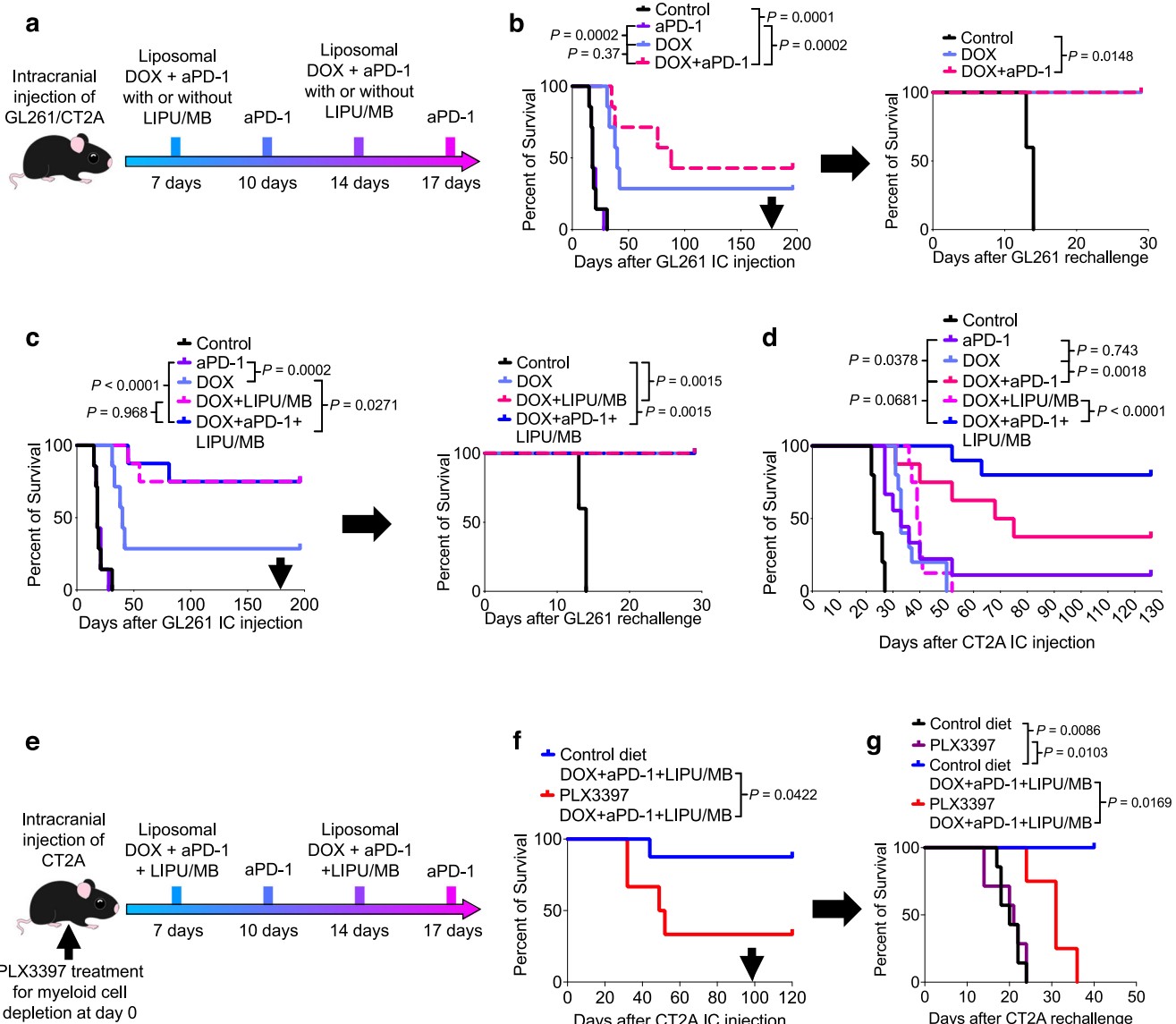

**Fig. 6 | Enhanced delivery of liposomal DOX and aPD-1 by LIPU/MB increases survival in GBM-bearing mice. a** Therapeutic scheme employed to treat GBM-bearing mice with the combination of liposomal DOX and aPD-1 delivered with or without LIPU/MB using GL261 and CT-2A cells. **b** Kaplan-Meier curves showing the survival of GL261-bearing mice treated with aPD-1, liposomal DOX ($n$ = 7 mice per group), and the tumor rechallenge experiment performed in long-term survivors ($n$ = 5 mice for the control group, $n$ = 2 mice for DOX group, $n$ = 3 mice for DOX +aPD-1 group). **c** Kaplan-Meier curves showing the survival of GL261-bearing mice treated with aPD-1, liposomal DOX with LIPU/MB ($n$ = 7 mice for control, aPD-1, and DOX groups, $n$ = 8 mice for DOX + LIPU/MB and DOX+aPD-1 + LIPU/MB groups), and the tumor rechallenge experiment performed in long-term survivors ($n$ = 5 mice for the control group, $n$ = 2 mice for DOX group, $n$ = 6 mice for both DOX + LIPU/MB and DOX+aPD-1 + LIPU/MB groups). **d** Kaplan-Meier curve showing survival of CT-2A-bearing mice treated with aPD-1, the combinatorial therapy with and without LIPU/MB ($n$ = 10 mice for the control group, $n$ = 9 for aPD-1 group, $n$ = 8 mice for DOX+aPD-1 group, $n$ = 10 mice for DOX group, $n$ = 8 mice for DOX + LIPU/MB group, and $n$ = 10 mice for DOX+aPD-1 + LIPU/MB group). **e** Therapeutic scheme employed following myeloid cell depletion using PLX3397 in GBM-bearing mice with the combination of liposomal DOX and aPD-1 delivered with LIPU/MB using CT-2A cells. **f** Kaplan-Meier curve showing the survival of CT-2A-bearing mice treated with either control diet or PLX3397 followed by liposomal DOX, aPD-1, and LIPU/MB ($n$ = 8 mice for the control diet group, $n$ = 6 mice for the PLX3397 group). **g** Kaplan-Meier curve showing the survival of newly injected mice with CT-2A, and long-term survivors treated with either the control diet or PLX3387 ($n$ = 7 mice for the control diet group, $n$ = 7 mice for the PLX3397 group, $n$ = 3 mice for the control diet DOX+aPD-1 + LIPU/MB group, and $n$ = 4 mice for the PLX3397 DOX+aPD-1 + LIPU/MB group). $P$ values in **b**–**d**, **f** and **g** were derived from the log-rank test. Source data are provided as a Source Data file.

## DOX plus aPD-1 with concomitant LIPU/MB promotes an IFN-γ⁺ phenotype in GBM-infiltrating T cells

Previous evidence has demonstrated the critical role of IFN-γ produced by T cells for DOX antitumoral immunity[31]. In addition, considering the abundance of CD8⁺ and CD4⁺ T cells exhibiting an IFN-γ⁺ TNF-α⁺ phenotype in long-term survivors treated with DOX (Fig. 4e, f), we interrogated the phenotype of tumor-infiltrating T cells in GBM patients who received DOX plus aPD-1 delivered with LIPU/MB. We performed flow cytometry analysis of CD45⁺ cells that were

further gated on CD8⁺ and CD4⁺ T cells (Supplementary Fig. 14). We found that CD8⁺ and CD4⁺ T cells from GBM patients treated with DOX and aPD-1 delivered with LIPU/MB exhibited a higher expression and percentage of IFN-γ compared to T cells derived from GBM patients that did not receive any treatment before surgery ($P$ = 0.0086 for CD4⁺ T cells and $P$ = 0.02 for CD8⁺ T cells, unpaired $t$-test; Fig. 7a).

To determine whether the efficacy of DOX plus aPD-1 delivered with LIPU/MB is dependent on CD8⁺ T cells, we performed a survival

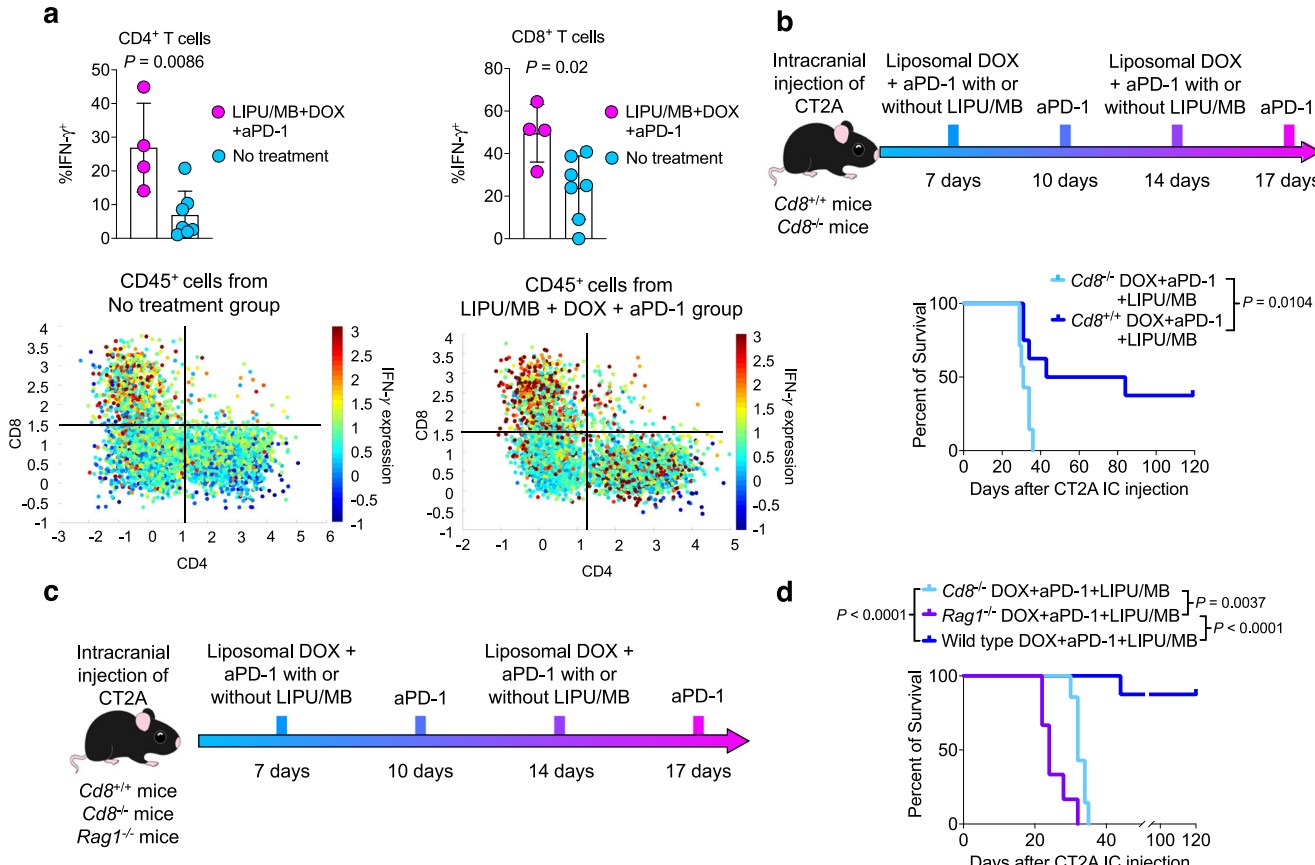

**Fig. 7 | LIPU/MB-mediated delivery of liposomal DOX and aPD-1 influences and requires tumor-infiltrating lymphocytes. a** Top: Bar plots showing the percentage of CD8+ and CD4+ T cells expressing IFN-γ+ from non-treated and DOX-treated GBMs. Bottom: Scatter plots showing the expression of IFN-γ+ by CD8+ and CD4+ T cells from non-treated and DOX-treated GBMs. The color key indicates normalized expression values of the indicated markers. $n = 4$ GBM samples treated with liposomal DOX + pembrolizumab + LIPU/MB, $n = 7$ GBM samples that did not receive treatment. $P$ values derived from unpaired one-sided $t$-test. **b** Therapeutic scheme and Kaplan-Meier curve representing the survival experiment and outcomes of CT-2A-bearing mice in the context of $Cd8^{+/+}$ and $Cd8^{-/-}$ backgrounds

treated with liposomal DOX and aPD-1 with LIPU/MB ($n = 7$ mice for the $Cd8^{-/-}$ DOX +aPD-1 + LIPU/MB group, $n = 8$ mice for $Cd8^{+/+}$ DOX+aPD-1+LIPU/MB group). **c, d** Therapeutic scheme, and Kaplan Meier curve showing the survival of GBM-bearing mice treated with the combination of liposomal DOX and aPD-1 delivered with LIPU/MB using CT-2A cells in wild type, $Cd8^{-/-}$ and $Rag1^{-/-}$ genetic backgrounds ($n = 7$ mice for the $Cd8^{-/-}$ DOX+aPD-1 + LIPU/MB group, $n = 6$ mice for $Rag1^{-/-}$ DOX +aPD-1 + LIPU/MB group, $n = 8$ mice for wild-type DOX+aPD-1+LIPU/MB group). $P$ values in **b** and **d** were derived from the log-rank test. Source data are provided as a Source Data file. Data are presented as mean ± s.d. in (**a**). Gating strategy detailed in Supplementary Fig. 3.

experiment using CT-2A cells to test DOX and aPD-1 in the context of LIPU/MB in $Cd8^{-/-}$ and $Cd8^{+/+}$ C57BL/6 mice. We observed a survival benefit only in mice that had intact immunity treated with DOX plus aPD-1 delivered with LIPU/MB, whereas none of the mice absent of CD8+ T cells survived despite treatment ($P = 0.0104$, log-rank test; Fig. 7b). Using the same murine glioma model, we evaluated whether other players of the lymphocyte population affected the survival of GBM-bearing mice following our proposed combinatorial therapy. Thus, we tested DOX plus aPD-1 delivered with LIPU/MB in wild-type mice, $Cd8^{-/-}$ mice, and $Rag1^{-/-}$ mice that are characterized by the absence of B cells as well as both CD8+ and CD4+ T cells (Fig. 7c). Consistent with the previous result, we noticed that wild type mice led to a high percentage of long-term survivors, whereas $Cd8^{-/-}$ and $Rag1^{-/-}$ mice did not ($P < 0.0001$, log-rank test; Fig. 7d). Further analysis showed that $Cd8^{-/-}$ CT-2A-bearing mice exhibited increased survival compared to $Rag1^{-/-}$ mice ($P = 0.0037$, log-rank test; Fig. 7d), suggesting that CD4+ T cells and B cells have an effect on survival in GBM-bearing mice treated with the proposed combinatorial therapy.

Together, these results indicate that DOX plus aPD-1 delivered with the SC9 ultrasound device induces an IFN-γ+ phenotype in T cells from GBM patients. In preclinical models, the efficacy of DOX plus aPD-1, which can be enhanced by LIPU/MB, relies on the antitumoral activity of lymphocytes.

## Discussion

Although there has been significant progress in understanding key aspects of GBM tumor biology[46], the efficacy of many systemically delivered drugs for this disease is limited by the BBB. LIPU/MB-based BBB disruption is an emerging approach to overcome this challenge. In this translational study, patients with recurrent GBM who had progressed after prior chemotherapy delivered with the skull-implantable SonoCloud-9 device were treated with DOX as an immunomodulator, and pembrolizumab as an immune checkpoint inhibitor administered concomitantly with BBB opening by LIPU/MB. In this context, we demonstrated that LIPU/MB can enhance the penetration of both DOX and aPD-1 in the human and murine brain.

DOX is commonly used in chemotherapy regimens for its cancer cell-killing properties[47]. However, in recent years, its immunomodulatory effects have become increasingly recognized[24,25,33]. In this context, the results of our study provide pharmacokinetic and immunological evidence that DOX delivered with LIPU/MB exhibits immunogenic properties at specific doses influencing both cancer and immune cells[30]. This is crucial to understand the optimal doses and conditions for DOX in combination with immunotherapy[32].

Like our study, previous preclinical reports relied on FUS and DOX for brain tumor treatment[48,49]. Similarly, previous clinical studies have employed ultrasound-mediated disruption of the BBB to facilitate the

delivery of chemotherapeutic agents like DOX and antibodies for the treatment of brain tumors[10,19,50]. However, these studies, including the sole clinical investigation that evaluated ultrasound-based BBB disruption for delivering DOX in a GBM patient[50], primarily utilized MR-guided transcranial FUS. The technology we employed in this study relies on an implantable ultrasound device that, in contrast to MR-guided transcranial FUS, does not require the sound waves to penetrate the skull. Therefore, ultrasound waves at 1 MHz can be used in this implantable approach to target a large volume at once. This differs from transcranial FUS where a much lower frequency (typically 220 kHz) is used to more efficiently penetrate the bone, and a small focal volume is scanned across the targeted region[11].

We observed that our chemoimmunotherapy regimen modulated the phenotype of the tumor microenvironment. We discovered that the treatment not only stimulated a response from GBM-infiltrating myeloid cells through IFN-γ production but also led to infiltration by CD4$^+$ IFN-γ$^+$ TNF-α$^+$ T cells in long-term survivor mice treated with DOX and LIPU/MB. This finding aligns with previous reports that demonstrate an increase in Th1 cells after DOX treatment[30]. Furthermore, in GBM patients receiving DOX and pembrolizumab delivered with LIPU/MB, we observed IFN-γ$^+$ expression by tumor-infiltrating CD8$^+$ and CD4$^+$ T cells. This is noteworthy as the T cells infiltrating GBM tumors have a hypofunctional phenotype with low production of IFN-γ, IL-2, and TNF-α compared to healthy donor PBMCs[51]. The suggested mechanism for this modulation involves DOX-mediated epigenetic remodeling including nucleosome turnover and histone eviction around gene promoters[52,53], promoting the expression of IFN-γ transcription factors including *Eomes* and *Tbx21* (T-bet), resulting in increased intratumor IFN-γ protein levels[30]. Our results suggest that the successful introduction of DOX into the brain through LIPU/MB leads to upregulation of the *IFNG* gene, resulting in increased IFN-γ production in T cells and GBM-associated microglia and macrophages. Our findings highlight the role played by CD8$^+$ T cells and myeloid cells in the efficacy of DOX and aPD-1 delivered with LIPU/MB, as evidenced by the loss of survival benefit in the absence of these immune cells. Therefore, the induction of an IFN-γ-specific phenotype on both innate and adaptive immune cells by DOX promotes a more favorable TME which could improve outcomes for immunotherapy for GBM.

Phase III clinical trials have consistently demonstrated the limited efficacy of anti-PD-1 immune checkpoint inhibitors in treating unselected GBM populations[20–22]. However, there is evidence reporting sustained benefits in a subset of GBM patients[54–56]. Our study and several other lines of evidence suggest that the lack of efficacy observed in these trials can be partially attributed to poor penetration of antibodies to the brain, the relative weakness of aPD-1 as monotherapy in the context of an immunosuppressive microenvironment in GBM, and biological differences across tumors that render some cases more susceptible to others[54,57,58]. In this study, we focused on the modulation of the immunosuppressive microenvironment through the use of DOX, and the delivery challenge posed by the BBB, through the use of LIPU/MB. Indeed, despite a recent clinical study reporting sufficient concentration of pembrolizumab in the CSF to block PD-1 on endogenous T cells and CAR T cells[34], it remains unclear whether these antibodies effectively penetrate brain tissue. We measured the concentration of aPD-1 within tumors and peritumoral tissues in GBM patients. Given previous findings of immune checkpoint blockade inducing a clonal remodeling of peripheral T cells[59], our strategy to employ LIPU/MB for brain-directed aPD-1 delivery was designed to maintain T cell activity once they infiltrate the brain by preventing interactions between PD-1 and their ligands PD-L1/2 expressed by both tumor and myeloid cells. We found higher concentrations of aPD-1 in sonicated brain peritumoral regions compared to those that were not sonicated, demonstrating the therapeutic advantage of LIPU/MB in delivering antibodies directly into the brain of GBM patients. Given the timing of sampling two days after aPD-1 administration and LIPU/MB, it

is possible that the concentration of these antibodies was higher initially, as suggested by our pharmacokinetic studies in mice.

Our study underpins the potential benefits of combining DOX with immune checkpoint inhibitors in the treatment of GBM. An approximately 2-fold increase in the concentration of both DOX and pembrolizumab was observed, which corresponds to the concentration range that facilitates MHC molecule upregulation and immune modulation in both tumor and myeloid cells in controlled experiments that were performed in vitro. Whereas the limited number of GBM patients prevents us from investigating whether the drug concentration increases translate into clinical efficacy, such an increase is nonetheless suggestive of enhanced drug delivery and potential therapeutic advantage. The efficacy of DOX, when administered with LIPU/MB, extends beyond simple drug delivery; it notably modulates the TME. This procedure and treatment led to the upregulation of IFN-γ in T cells and GBM-associated myeloid cells and enhanced expression of antigen-presenting molecules, including HLA-ABC and HLA-DR. Such DOX-mediated upregulation potentially improves the presentation of tumor antigens to T cells, thereby amplifying the effectiveness of T cell-based immunotherapies, including PD-1 blockade.

The upregulation of HLA-ABC and HLA-DR we observed with DOX might improve antigen presentation, and ultimately efficacy of immunotherapy. High expression of MHC I and MHC II was associated with favorable outcomes in melanoma patients treated with immune checkpoint inhibitors[60,61]. Furthermore, transcriptional suppression of MHC I genes contributes to resistance to immune checkpoint inhibitors in patients with Merkel cell carcinoma[62]. On the other hand, treatment with aPD-1 targeting CD8$^+$ PD-1$^+$ T-cells lacking proper antigenic stimulation to their TCRs results in apoptotic and dysfunctional T cells and, thus, aPD-1 therapeutic resistance[63]. Previous clinical trials exploring the immunological synergy between DOX and immune checkpoint blockade in other cancers[33,64–67] further support the promise of this combinatorial therapy in GBM. Though we have tested aPD-1, second-generation immune checkpoint inhibitors are under development, and investigation of how these will perform combined with DOX will provide additional insights.

Our study has limitations to keep in mind. The number of clinical samples was small as only selected GBM patients qualified for a 3$^{rd}$ line of therapy. In addition, the absence of a group of patients treated with DOX plus pembrolizumab without LIPU/MB prevents us from comparing the added therapeutic value of LIPU/MB to modulate the TME and tumor immunogenicity. These patients had previously participated in a clinical trial that included a SonoCloud-9 implant and delivery of paclitaxel that could have influenced the immune landscape of these tumors. Nonetheless, the results derived from these human tumors were reconciled and validated with preclinical GBM models not treated with previous therapies where DOX and aPD-1 were assessed with and without LIPU/MB. It is possible that an intraoperative pharmacokinetic study, similar to our previous approach[17] but adapted to DOX and aPD-1, might allow a more accurate delineation of the sonicated brain, and thus, lead to more robust differences in the concentration of DOX and antibodies between the sonicated and non-sonicated brain. From an immunological standpoint, the upregulation of PD-L1 potentially induced by the secretion of IFN-γ by GBM-associated myeloid cells treated with DOX represents an immunosuppressive event that suppresses T cell cytotoxic activity[68]. The addition of aPD-1 therapy aims at overcoming this immunosuppressive effect by preventing the interaction between the PD-1 receptor and their ligands, PD-L1 and PD-L2. Moreover, as suggested by multiple studies[54–58,69–71], the efficacy of this combinatorial immunotherapy in GBM might likely be restricted to subsets of patients that exhibit tumor-intrinsic and microenvironmental features that are conducive to mounting a robust antitumoral immune response. Studies evaluating these therapies should consider these variables across patients.

In conclusion, this study describes a chemoimmunotherapy approach in conjunction with BBB opening for GBM. These results are the basis for a clinical trial we initiated to prospectively evaluate the safety and efficacy of immune checkpoint blockade with DOX and LIPU/MB in the adjuvant setting for GBM (NCT05864534). Ultimately, patient selection, combinatorial therapeutic regimens, and enhanced drug delivery to the brain might increase the prospect of an efficacy signal for GBM.

## Methods

This translational study was conducted in accordance with the institutional ethical regulations and the Declaration of Helsinki principles. Informed consent was obtained from all GBM patients. Institutional review board (IRB) approval was acquired from Northwestern University. This study sought to determine the ability of the LIPU/MB technology to improve the penetration of DOX and aPD-1 for GBM. Murine GBM models and GBM patient tumor samples treated with liposomal DOX and aPD-1 delivered with LIPU/MB were analyzed to evaluate immunological responses by T cells and GBM-associated microglia and macrophages. The efficacy of the combinatorial treatment was evaluated in GBM-bearing mice.

### GBM patient cohort treated with LIPU/MB-based delivery of DOX and aPD-1 under expanded access protocol

The human GBM tissue analysis presented derives from patients who had a skull-implantable ultrasound at the time of GBM progression, following enrollment into a phase 1 clinical trial where this device was used for LIPU/MB-based delivery of albumin-bound paclitaxel for patients with recurrent GBM (NCT04528680). At the time of tumor progression following treatment on this trial, these patients consented and enrolled in an expanded-access single-patient protocol where LIPU/MB was repurposed to deliver DOX and aPD-1 (pembrolizumab). For all patients presented, a subsequent resection or biopsy was performed during treatment using LIPU/MB to deliver DOX and aPD-1.

### LIPU/MB-mediated BBB opening in GBM patients and treatment with liposomal DOX and pembrolizumab

As part of a phase I clinical trial evaluating nab-paclitaxel (NCT04528680), the SonoCloud-9 ultrasound device (Carthera, Lyon, France) was implanted during neurosurgery targeting the tumor and peritumoral brain in recurrent GBM patients. Patients received additional treatment cycles in an outpatient setting in which the ultrasound device was activated by percutaneous access with a single-use transdermal needle concomitantly with the injection of 10 μL/kg of IV microbubbles (DEFINITY, Lantheus, Billerica, USA). 200 mg of pembrolizumab was administered before sonication, and 30 mg of liposomal DOX was administered after sonication. Steroids and mannitol were avoided to perform pharmacokinetic studies. 48 h after treatment with liposomal DOX and pembrolizumab, GBM patients underwent surgery for tumor resection. The acquisition of non-eloquent peritumoral brain samples either sonicated or not sonicated was performed when justified as per standard neurosurgical technique as reported previously[72,73].

### Determination of the DOX concentration in peritumoral brain tissue and mouse brains

Four GBM patients undergoing surgery for resection of tumors with implantation of the SonoCloud-9 device were studied for this analysis. Brain parenchyma that did not display enhancement in contrast MRI was considered peritumoral brain. Sonicated and non-sonicated peritumoral brain samples were identified taking into account their location relative to the sonication field of the ultrasound device. Non-sonicated peritumoral brain samples were acquired first with the use of a new set of surgical instruments for each sample. Every peritumoral brain tissue was washed with saline solution to remove blood. For DOX

quantification in mouse brains, C57BL/6 mice were treated with 5 mg/kg of liposomal DOX and Evans blue delivered with LIPU/MB. 48 h after treatment mice were euthanized. For sonicated brains, we dissected the regions where Evans blue was visible. For non-sonicated brains, we obtained one hemisphere for further analysis. These samples were sectioned into 30 mg pieces and flash-frozen for further quantifying DOX levels. DOX was quantified in peritumoral brain samples and mouse brains using LC-MS/MS (5500 Triple Quad equipped with an ExionLC™ AC20, SCIEX, Framingham, MA). A 50 μL aliquot of sample was mixed with 250 μL of acetonitrile containing 25 ng of a DOX analog (IS) in a 96-well deep well plate. After shaking for 5 minutes, the sample was centrifuged at 4000 rpm for 10 mins at 4 °C. Chromatographic separation was achieved with a Kinetex Biphenyl, $50 \times 2.1$ mm, 5 μm (Phenomenex, Torrance, CA, USA) column. The mobile phase was A: 5 mM ammonium acetate in 10% methanol in water (v/v) and B: 0. 5 mM ammonium acetate in 45% acetonitrile and 45% methanol in water (v/v). After injection, initial conditions with A at 30% were held for 5 min. The flow rate was 0.3 ml/min at 25 °C. Retention times for DOX and IS were 1.2 and 1.4 min, respectively with a total run time of 5 min. A turbo ion spray interface was used as the ion source operating in negative mode. Acquisition was performed in multiple reaction monitoring mode (MRM) using m/z $542.2 \rightarrow 395.2$ and $513.2 \rightarrow 365.0$ ion transitions at low resolution for DOX and IS, respectively.

### Animal studies

All animal experiments were approved by Northwestern University's Institutional Animal Care and Usage Committee under protocol no. IS000017464. Six- to twelve-week-old male and female C57BL/6 mice were purchased from Charles River Laboratories for these experiments. Sex was not considered in the study design because this variable was not relevant to the study. All animals were housed in a pathogen-free animal facility at Northwestern University at a relatively constant temperature of 24 °C and humidity of 30%–50%. The $Cd8^{-/-}$ mice breeders were purchased from the Jackson Laboratory (B6.129S2-$Cd8a^{tm1Mak}$/J, stock #002665). The genotyping protocol was performed following the recommendation of the Jackson Laboratory and separated by gel electrophoresis on a 1.5% agarose gel. Equal ratio of male and female mice were used for all the experiments.

### LIPU/MB-mediated BBB opening for DOX and aPD-1 delivery in mouse models

All sonication procedures were performed as previously described[6]. In brief, a preclinical ultrasound device (Sonocloud Technology) manufactured by Carthera (France) along with IV injection of 100 μL of MB (Lumason, Bracco) reconstituted following the manufacturer's protocol. C57BL/6 mice were employed for sonication experiments in which they were anesthetized with ketamine/xylazine cocktail intraperitoneally (ketamine 100 mg/kg, xylazine 10 mg/kg). Hair from the mice heads was removed using Nair hair removal lotion followed by washes with warm water. Different doses of liposomal DOX, mouse PD-1 blocking antibody, isotype IgG4 antibody, nivolumab, and/or fluorescein were injected via retro-orbital route followed by MB injection and sonication. For LIPU/MB-mediated BBB opening, mice were placed in a supine position with their heads at 15 mm from a 10 mm diameter flat ultrasound transducer holder touching the degassed water contained in the ultrasound device. The sonication procedure was performed for 60 seconds using a 1-MHz 25,000-cycle burst at a 1 Hz pulse repetition frequency and an acoustic pressure of 0.3 MPa predefined in the ultrasound device. Mice were removed from the ultrasound transducer holder, put in a clean cage, placed upon a heating pad, and monitored until they recovered from anesthesia.

### Flow cytometry of PDX cell lines and HMC3 cells

PDX cell lines were provided by Dr. Jann Sarkaria (Mayo Clinic). HMC3 cells were acquired from ATCC (cat. CRL-3304). GBM6, GBM63, and

HMC3 cells were seeded on 6-well plates for treatment with increasing concentrations of DOX hydrochloride (D1515, Sigma Aldrich), IFN-γ (300-02, Peprotech) or 50 µM of TMZ (Accord). PDX and HMC3 cells were treated for 5 h followed by 3 washes with PBS and cultured with Dulbecco's modified Eagle's medium (Corning) supplemented with 10% FBS (Hyclone) and 1% penicillin/streptomycin (Corning). 72 h after treatment, PDX cells were harvested using Accutase cell detachment solution (Corning) and were stained initially with eBioscience Fixable Viability Dye eFluor 780 (Thermo Fisher). Staining with HLA ABC AF-647 (cat. 311414, dilution 1:100) and HLA-DR BV421 (cat. 307636, dilution 1:100) was done for PDX cells. Staining with IFN-γ AF-700 (cat. 502520, dilution 1:100), HLA ABC FITC (cat. 311404, dilution 1:100), and PD-L1 Pe-Cy7 (cat. 374506, dilution 1:100) was done for HMC3 cells. All antibodies for this experiment were from Biolegend. Data were acquired using BD FACSymphony Flow Cytometer.

## T cell proliferation assay

CD8 + T cells were isolated from spleens of OT-1 or wild-type C57BL/6 mice using the EasySep Mouse CD8 + T Cell Isolation Kit from StemCell Technologies, following the manufacturer's protocol. Mice were euthanized using CO2 asphyxiation; spleens were excised and immediately placed in ice-cold RPMI. The spleens were then homogenized with a tissue grinder and the homogenates were strained through a 70 µm mesh. The strained tissue was washed twice using PBS, followed by centrifugation at 1500 rpm for 10 minutes at 4 °C. The cell pellet was resuspended to achieve a density of $1 \times 10^8$ cells/ml. In a 5 ml sterile tube, cells were incubated with a 50 µl/ml CD8+ isolation cocktail for 10 minutes, and subsequently, 125 µl/ml of magnetic beads were added for an additional 5 minutes. The tube was placed in an EasySep Magnet for 2.5 minutes to separate the cells. The supernatant, enriched in CD8 + T cells, was collected and washed with complete RPMI medium.

For labeling, the CD8 + T cells were stained with eBioscience eFluor 450 proliferation dye from ThermoFisher. T cells were resuspended at a concentration of $2 \times 10^7$ cells/ml in 1× PBS and combined with an equal volume of 10 µM proliferation dye. The mixture was incubated for 10 minutes at 37 °C. The reaction was quenched with 10 ml of RPMI, and the tube was placed on ice for 10 minutes. After two washes with complete RPMI, cells were ready for the proliferation assay. The labeled CD8 + T cells were adjusted to a final concentration of $5 \times 10^5$ cells/ml. Aliquots of 100 µl were dispensed into each well of a 96-well plate containing $5 \times 10^4$ GL261 wild-type or GL261-OVA cells pre-treated with varying concentrations of DOX. The cells were then co-cultured for 72 h. Post-incubation, the CD8 + T cells were harvested, washed twice with 1× PBS, and stained with Fixable Viability Dye eFluor780 (dilution 1:1000, ThermoFisher) and BV605 anti-Mouse CD8α antibody (dilution 1:100, cat. 100743, BioLegend) for subsequent analysis.

The fluorescence intensity of the stained cells was measured using a BD FACSymphony A1 Cell Analyzer (BD Biosciences). The decay in fluorescence intensity of the proliferation dye on the CD8 + T cells was analyzed to assess the proliferative activity during the co-culture with the cancer cells.

## RT-qPCR of HMC3 cells

HMC3 cells were initially seeded in 6-well plates at a density of 600,000 cells per well. After 24 h, DOX was added to the media at concentrations ranging from 0 µM to 4.8 µM. The cells were then incubated for 5 h at 37 °C with 5% $CO_2$. Subsequently, the cells underwent PBS washing, and fresh media was added. After 24 h, cell pellets were collected. For RNA isolation and DNAse treatment, the Direct-zol RNA kit (#R2053, Zymo Research) was employed according to the manufacturer's instructions. RNA concentration and purity were assessed using a Nanodrop (Thermo Fisher), and the samples were stored at −80 °C. cDNA was generated from 1 gr of purified RNA by using a high-capacity cDNA reverse transcription Kit (#4368814,

Applied Biosystems) following manufacturer instructions. The resulting cDNA was diluted and analyzed by quantitative PCR (qPCR) using the CFX Connect Real-Time PCR system (BioRad) with the SsoAdvanced™ Universal SYBR® Green Supermix (#1725271, BioRad).

Primers were designed using the Primer 3 software. Conditions were as follows: one cycle at 95 °C for 10 min, followed by 45 cycles of 10 s at 95 °C, 10 s at the primer hybridization temperature, and 10 s at 72 °C. 2 − ΔΔCt method was adopted to analyze the qPCR results. Primer sequences for *IFNG*, *IFNA1*, and *IFNB1* are provided as Supplementary Data.

## Multiplex immunofluorescence of human GBM samples

5 µm sections were cut from FFPE GBM samples. Slides were loaded onto Leica Bond Rx where they were baked at 60 °C for 30 min followed by a dewaxing process consisting in rinsing the slides three times with 150 µL of preheated Bond dewax solution and three rinses of 150 µL ethanol. Bond ER1 solution was used for antigen retrieval for antibodies incubated with pH6. Bond ER2 solution was used for antigen retrieval for antibodies incubated with pH9. Slides were incubated at 99 °C for 20 min. For the next steps, the Opal 7-color IHC kit (NEL821001KT, Akoya Biosciences) was employed. Primary antibodies were diluted using the Opal Antibody Diluent/Block solution provided with the kit. The following antibodies were used: SOX2 (cat. Ab92494, Abcam, clone EPR3131, 1:5000, pH9) paired with Opal 540, TMEM119 (cat. HPA051870, Sigma-Aldrich, 1:250, pH6) paired with Opal 520, CD163 (cat. ab182422, Abcam, clone EPR19518, 1:600, pH9) paired with Opal 620, IFN-γ (cat. ab231036, Abcam, clone EPR21704, 1:200, pH9) paired with Opal 570, HLA-DR (cat. ab20181, Abcam, clone TAL 1B5, 1:1000, pH6) paired with Opal 650, HLA-ABC (cat. ab70328, Abcam, clone EMR8-5, 0.3 µg/mL, pH6) paired with Opal 690. Multiplex staining was performed with an antigen retrieval step, protein blocking, epitope labeling, and signal amplification between each cycle. At the end of all cycles, Spectral DAPI (Akoya Biosciences) was used to counterstain the slides, which were mounted with a long-lasting aqueous-based mounting medium.

## Imaging and analysis of multiplex immunofluorescence images

Images were acquired using the Vectra 3 Automated Quantitative Pathology Imaging System from Akoya Biosciences. Multispectral images (MSI) were acquired in tumor regions previously delineated by a neuropathologist. Spectral unmixing was performed for all the MSI files using a spectral library for all Opal dyes as a reference in inForm Tissue Finder software 2.6. (Akoya Biosciences). Cell segmentation was performed employing DAPI to delineate nuclei as well as phenotyping of particular cell types including SOX2+ HLA ABC+, SOX2+ HLA DR+, TMEM119+ IFN-γ+, TMEM119+ HLA ABC+, and CD163 HLA ABC+. Next, processed images from all tumor samples were exported to data tables. These exported files were further processed in R employing the R packages Phenoptr and PhenoptrReports to merge and create consolidated files for each tumor sample. Consolidated files were used to quantify the phenotypes of interest.

To evaluate IFN-γ by GBM-associated myeloid cells, CD163+ and TMEM119+ cells were isolated in R from pre-treatment and on-treatment GBM samples. Next, fcs files containing only myeloid cells were created using inForm2fcs using.txt files. Newly created fcs files were uploaded to Matlab v. R2021b. Data were transformed (asinh) using cofactor 5 and plotted in a PCA plot considering CD163 and TMEM119 expression, IFN-γ was then assessed in myeloid cells from pre-treatment and on-treatment GBM samples.

## Processing and flow cytometry analysis of human GBM-infiltrating T cells

Non-treated and DOX-treated GBM samples were acquired by the Nervous System Tumor Bank at Northwestern University. Tumor samples were immediately processed into single-cell suspension using

the Adult Brain Dissociation Kit (cat. 130-107-677, Miltenyi Biotec) following the manufacturer's protocol. Single-cell suspension of GBM samples was cryopreserved using RPMI media (Corning), DMSO, and FBS (Hyclone). Cryopreserved non-treated and DOX-treated single-cell suspensions were thawed at the same time for stimulation and staining. Cells were washed with complete RPMI media and re-stimulated with a cell activation cocktail (cat. 423303, Biolegend) for 4 h. After 4 h of re-stimulation, cells were washed with 1X PBS and stained with Zombie-NIR (cat. 423105, Biolegend) for cell viability. Next, cells were incubated for 5 minutes with human Fc block (BD 564219) on ice and were stained for surface markers with the following fluorescently conjugated antibodies: CD45 BV605 (cat. 368524, dilution 1:100, Biolegend), CD3 PerCP (cat. 300326, dilution 1:100, Biolegend), CD8 PE-Cy7 (cat. 300914, dilution 1:100, Biolegend), CD4 FITC (cat. 317408, dilution 1:100, Biolegend). Cells were then fixed with Fixation/Permeabilization concentrate (cat. 00-5123-43, eBiosciences) and stained intracellularly with IFN-γ Alexa Fluor 700 (cat. 505824, dilution 1:100, Biolegend). Data was acquired using BD FACSymphony Flow Cytometer and analyzed using FlowJo (BD). To visualize the expression of IFN-γ by T cells in scatter plots, the associated Matlab-based tool, cyt3[74], was employed. FCS files including $CD45^+$ live single cells were uploaded into cyt3 for visualization. Subsampling was performed to represent 5000 cells for each group: No treatment and liposomal DOX + pembrolizumab + LIPU/MB. The expression of each marker was normalized equally across the board.

### Cell lines and implantable syngeneic murine GBM models for survival and immunophenotyping studies

Murine GL261 cell line was acquired from the National Institutes of Health. The CT-2A cell line was acquired from Millipore (cat. SCC194). GL261 and CT-2A were cultured in Dulbecco's modified Eagle's medium (Corning) supplemented with 10% FBS (Hyclone) and 1% penicillin/streptomycin (Corning) at 37 °C in incubators with humified atmosphere of 5% $CO_2$ and 95% air. All murine cell lines used for this study were routinely tested for mycoplasma and were confirmed negative before intracranial orthotopic injection. To perform intracranial injection of syngeneic murine GBM cell lines, mice were anesthetized with a ketamine/xylazine cocktail. Artificial tears were used to prevent eye drying and protect the eye. The surgical site was cleaned with a swab of povidone-iodine and 70% ethanol. An incision was made along the sagittal axis of the mouse head to expose the skull underneath. Then, using a sterile handheld drill (Harvard Apparatus) a burr hole was created 3 mm lateral and 2 mm caudal relative to sagittal and bregma sutures. Afterward, the mice were placed in a stereotaxic device (Harvard Apparatus). 200,000 GL261 or 100,000 CT-2A cells were injected into the left hemisphere of the brain at 3 mm depth through the burr hole. After the injection of GBM cells, the incision was closed using 9 mm stainless steel wound clips. On days 7 and 14 after IC implantation of GBM cell lines, mice were treated with liposomal DOX from the pharmacy of Northwestern Memorial Hospital. On days 7, 10, 14, and 17 after IC injection of GBM cells, mice were treated with either 200 μg of anti-mouse PD-1 (CD279) BE0146 (BioXcell) or rat IgG2a isotype control BE0089 (BioXcell). Animals were monitored until they reached the study endpoint. Euthanasia was performed when mice had a loss of body mass greater than 20% of the pre-study value or/and developed neurological deficits including spinning, ataxia, inability to walk, limping, seizing, etc.

### Immunophenotyping and treatment of GBM-bearing mice with increasing doses of liposomal DOX

Tumor and blood samples were processed for immunophenotyping analysis as previously described[75]. Specifically, mice were bled retro-orbitally, and blood samples were collected in heparinized PBS solution (1 mg/mL, Sigma-Aldrich). Red blood cells were lysed using an ACK lysing solution (Gibco, Thermo Fisher). After blood collection, mice

were euthanized in a CO2 chamber and intracardially perfused with chilled PBS. PBMCs were isolated from whole blood using Ficoll-Paque density gradient centrifugation. Whole blood was diluted 1:1 with sterile phosphate-buffered saline (PBS). The diluted blood was carefully layered over an equal volume of Ficoll-Paque in a centrifuge tube, ensuring a clear separation of layers. The samples were then centrifuged at 400 g for 30 minutes at room temperature, with the centrifuge brake deactivated to allow undisturbed layer separation. Post-centrifugation, the sample separated into four distinct layers. The thin, white PBMC layer located above the Ficoll-Paque was carefully aspirated using a pipette and transferred into a new centrifuge tube. The cells were washed by adding PBS and centrifuging at 200-300 g for 10 minutes. The supernatant was discarded, and the cell pellet was resuspended in fresh PBS. This washing step was repeated to ensure the complete removal of any residual Ficoll-Paque and platelets. The final cell pellet was resuspended in an appropriate medium for subsequent viability assessment using the trypan blue exclusion method. The isolated PBMCs were then used immediately for antibody staining.

Brain single-cell suspensions were obtained by mechanical dissociation using a manual tissue homogenizer (Potter-Elvehjem PTFE pestle, Sigma-Aldrich) in HBSS. Myelin and debris were removed by Percoll gradient separation. Brain single-cell suspensions were filtered through a 70-mm cell strainer and a syringe plunger. Cells were washed with complete RPMI media and were used immediately for antibody staining.

### Flow cytometry and immunophenotype analysis

Immunophenotype analysis of brain single-cell suspension and PBMCs was performed from different DOX treatment groups. After collection single-cell suspensions, cells were counted and washed with staining buffer (5% bovine serum albumin, 0.001% sodium azide in PBS). Next, cells were incubated for 5 minutes with human Fc block (BD 564219) on ice and were stained for surface markers for 30 min at 4 °C with the following fluorescently conjugated antibodies: CD45 BV605 (cat. 103138, dilution 1:100, Biolegend), CD3 PerCP (cat. 300326, dilution 1:100, Biolegend), CD8 PE-Cy7 (cat. 300914, dilution 1:100, Biolegend), CD4 FITC (cat. 317408, dilution 1:100, Biolegend), CD11b Pacific Blue (cat. 101224, dilution 1:100, Biolegend), H2-Kb BV421 (cat. 116514, dilution 1:100, Biolegend), and PD-L1 (cat. 124314, dilution 1:100, Biolegend). Cells were washed twice with cold PBS. Cells were stained with Fixable Viability Dye eFluor 780 (eBioscience, Thermo Fisher) for 30 minutes at 4 C. Cells were washed twice with the staining buffer. For the detection of cytokine production, cells were fixed and permeabilized using the eBioscience Foxp3/Transcription Factor Staining Buffer Set (Invitrogen, Thermo Fisher) for 90 min at room temperature. Cells were washed twice with the Permeabilization Buffer (provided in the permeabilization/fixation buffer kit) and incubated with 1 mL Ab for 1 h at 4 °C. Cells were washed twice with the Permeabilization Buffer (provided in the permeabilization/fixation buffer kit) and incubated with 1 mL Ab for 1 hour at 4 °C. Cells were washed twice with staining buffer. To evaluate cytokine expression, cells were stimulated for 5 h at 37 °C with the eBioscience Cell Stimulation Cocktail plus protein transport inhibitors (500x, Thermo Fisher) prior staining. Cells were washed twice with the staining buffer and stained with antibodies against IFN-γ (cat. 505824, Biolegend), GZMb (cat. 515406, Biolegend), TNF-α (cat. 502932, Biolegend), and IL-1β (cat. 25-7114-82, Invitrogen). Data were acquired using BD FACSymphony Flow Cytometer and all analysis employing flow cytometry data used FlowJo v. 10.7.1 and v. 10.10.

Regarding the gating strategy, immune cell populations were first identified using forward and side scatter characteristics to exclude doublets, followed by viability dye exclusion for dead cells using the eBioscience Fixable Viability Dye eFluor780. After gating on the $CD45^+$ and $CD11b^+$ populations to identify myeloid cells, and on $CD45^+$ and

CD11b⁻ for lymphocytes, we further characterized these populations by CD8⁺ and CD4⁺ markers for T cells. To create tSNE plots using flow cytometry data, the gating comprising single live cells for each group of samples was used and exported with all compensated parameters. The newly generated FCS files containing the single live cell data were uploaded for further data processing in the cytofkit2 app[76]. Ceil was the merged method employed along with autoLgcl as a transformation method with a fixed number of 2500 cells. The CD11b, CD4, and CD8 markers were used to perform a supervised clustering analysis. tSNE was the dimensionality reduction method employed with a tSNE perplexity of 30, tSNE Max Iterations of 1000, and seed 42. Rphenograph was used as the clustering method. Files and results were obtained using the previous parameters. Next, the new generated.Rdata file was uploaded to cytofkit2 to explore the data. In the marker panel, expression level plots were visualized using the spectral2 color palette, local scaling range, and centered for CD4, CD8, CD11b, CD45, and IFN-γ.

### IFN-γ ELISpot in PBMCs from mice treated with liposomal DOX

GL261 cells were injected intracranially in C57BL/6 mice. Mice were treated on day 7 and day 14 with liposomal DOX 5 mg/kg through the retroorbital route. On day 19, mice were anesthetized with ketamine/xylazine cocktail intraperitoneally (ketamine 100 mg/kg, xylazine 10 mg/kg). Blood was extracted by introducing a 1 ml TB syringe with a 25 G needle into the heart followed by slowly pulling the blood into the syringe. Next, the extracted blood was transferred to a K2 EDTA (K2E) blood collection tube (BD vacutainer) and placed in a shaker for 15 min at room temperature. Blood was centrifuged for 30 minutes to isolate plasma and cells. Cells were counted to get 100,000 PBMCs for further steps. Mouse IFN-γ Single-Color ELISpot (ImmunoSpot) was used to detect the number of IFN-γ spots in PBMCs of GL261-bearing mice. Following the isolation of PBMCs, these cells were incubated with GL261 tumor lysate. ELISpot assay was performed following the manufacturer's protocol. Data and images of the IFN-γ spots in 96-well plates were acquired using the classic AID EliSpot reader (Autoimmun diagnostika GBMH).

### Determination of the concentration of nivolumab in brain from C57BL/6 mice

Non-tumor-bearing C57BL/6 mice were used for this experiment. Nivolumab (Brystol Myers Squibb) was acquired from the pharmacy of Northwestern Memorial Hospital. Nivolumab ELISA kit (ab237651, Abcam) was employed to determine nivolumab concentrations in the brain and blood. Mice were anesthetized with ketamine/xylazine cocktail and treated with artificial tears once fully anesthetized. Next, mice were treated with nivolumab through the retro-orbital route. The dose used for mice was calculated based on allometric scaling which considers the animal equivalent dose based on body surface area converted from the human dose of 3 mg/kg of nivolumab[77]. After treatment with nivolumab, mice were placed in the US transducer holder for sonication as described above. Intravenous NaFI (Sigma-Aldrich) previously dissolved in PBS was administered immediately after sonication at a dose of 20 mg/kg in 100 μL. 1 and 4 h after sonication, mice were euthanized using a $CO_2$ chamber, and brains were harvested and placed in PBS. Mouse brains were imaged using the Nikon AZ100 epifluorescent microscope. With a clean scalpel, fluorescent areas of the brains representing regions of BBB disruption were dissected, separated, and weighed. Fluorescent brain regions were homogenized in 1 mL of Assay buffer (included in the Nivolumab ELISA kit) using a tissue grinder with PTFE pestle (Kimble, Capitol Scientific). Brain samples were diluted in a 1:10 ratio and blood samples were diluted in a 1:100 ratio using the Assay buffer. The rest of the procedure was done following manufacturer's kit protocol. The absorbance of samples (set to 450 nm) was read in Cytation 5 multi-mode reader (Biotek). A standard curve was generated employing the standards included in the kit to determine the concentration of nivolumab in each sample by interpolating values of the standard curve.

### Determination of the concentration of pembrolizumab in human GBM and peritumoral brain tissue

Two GBM patients undergoing surgery for resection of the tumor with implantation of the Sonocloud-9 US device were studied for this analysis. A Pembrolizumab ELISA kit (ab237652, Abcam) was employed to determine pembrolizumab concentrations. Blood samples were collected concurrently with peritumoral brain tissue during the patients' surgical resection. All brain samples were weighed. Samples were homogenized in 1 mL of Assay buffer (included in the Pembrolizumab ELISA kit) using a tissue grinder with PTFE pestle (Kimble, Capitol Scientific). Brain samples were not further diluted, and blood samples were diluted in a 1:100 ratio. The rest of the experiment was done following the manufacturer's kit protocol. The absorbance of samples (set to 450 nm) was read in Cytation 5 multi-mode reader (Biotek). A standard curve was generated employing the standards included in the kit to determine the concentration of pembrolizumab in each sample.

### Statistical analysis

Data are shown as mean ± SEM or ± SD as indicated in each figure legend. Data following normal distributions were subjected to unpaired or paired Student's t-test. One-way ANOVA was used to compare means between groups and determine which specific groups differed from each other Dunnett's or Tukey's multiple comparison tests were employed. We used the *lme4*[78] to perform a linear mixed effects analysis of the relationship between DOX concentrations in peritumoral regions and sonication. As fixed effects, we entered sonication or DOX+aPD-1 treatment into the model. As a random effect, we had patients as an intercept. $P$ values were obtained by likelihood ratio tests of the full model with the effect in question against the model without the effect in question. Kaplan-Meier curves were used to plot survival results analyzed using log-rank test. α = 0.05 was used to determine statistical significance. After intracranial injection of murine GBM cells, tumor-bearing mice were randomized to each treatment group. For each experiment, the replicate numbers are reported in each figure legend. Measurements were taken from distinct samples in mouse and human experiments. Prism v. 10 (GraphPad, San Diego, CA, USA), Matlab v. R2023b, R v. 4.3.0, and RStudio were used for statistical analysis and generation of figures.

### Reporting summary

Further information on research design is available in the Nature Portfolio Reporting Summary linked to this article.

## Data availability

All data are available within the Article, Supplementary Information, or Source Data file. Source data are provided in this paper. Source data are provided with this paper.

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

## Acknowledgements

This work was supported by the NIH grant 1R01NS110703-01A1 (AMS), NIH/NCI 1U19CA264338-01 (A.M.S. and R.S), NIH/NCI 1R01CA245969-01A1 (A.M.S. and R.S,), P50CA221747 SPORE for Translational Approaches to Brain Cancer. Additionally, this work was supported by generous philanthropic support from the Moceri Family Foundation and the Panattoni family. V.A.A. is financially supported by the Mexican government through the Mexican National Council for Science and Technology (CONACYT) and the Plan of Combined Studies in Medicine (PECEM) of the National Autonomous University of Mexico (UNAM). M. McCord is supported by the F32 award 1F32CA264883. We thank J. Walshon, A. Steffens, and M. Santa Flowers from the Nervous System Tumor Bank supported by the P50CA221747 SPORE for Translational Approaches to Brain Cancer. Multiplex Immunofluorescence was performed at the Immunotherapy assessment core at Northwestern University. Imaging work was performed at the Northwestern University Center for Advanced Microscopy generously supported by NCI CCSG P30 CA060553 awarded to the Robert H Lurie Comprehensive Cancer Center. This work was supported by the Northwestern University—Flow Cytometry Core Facility supported by Cancer Center Support Grant (NCI CA060553). We thank B. Frederick, B. Shmaltsuyeva, and H. Fan at the Northwestern University Pathology Core Facility funded by the Cancer Center Support Grant (no. NCI CA060553). Most importantly, we thank the patients and their families for their contribution to this research.

## Author contributions

V.A.A performed the majority of the experiments and analysis. V.A.A., R.St., C.L.C., and A.M.S. conceptualized and designed the study. V.A.A., A.G., K.S.K., K.J.H., D.Y.Z., B.C., C.L.C., J.M., G.M., G.I.V., S.D. and L.C. performed animal experiments. A.G., G.B., C.De., M.C., A.C., L.C., C.G., C.A., R.V.L., G.B., R.W. and R.Sa. acquired imaging and clinical data. V.A.A., A.G., K.S.K., S.P., C.M.H., J.M., M.Mc., C.Dm., G.B., G.I.V., and C.L.C. analyzed the data. K.M., M.C. and C.M.H. processed and provided tumor samples from the brain tumor bank. JM performed ELISpot. I.P. and K.S.K. performed qRT-PCRs. M.E.C., S.P. and B.Z. performed the multiplex immunofluorescence staining. V.A.A., K.S.K., C.L.C., G.I.V., M.S.L., S.P. and M.E.C. performed flow cytometry experiments and the acquisition of data. M.Mu. measured the concentration of DOX in brain samples. V.A.A. took and analyzed the multiplex immunofluorescence images. V.A.A. determined aPD-1 concentrations in mouse and human tumor samples. A.M.S. performed neurosurgeries, sonications in GBM patients, and obtained tumor and peritumoral brain samples. V.A.A. and A.M.S. wrote the manuscript. R.St., C.L.C. and A.M.S. supervised the study.

## Competing interests

A.M. Sonabend and R. Stupp have received in-kind and or funding support for research from Agenus, BMS, and Carthera. A.M. Sonabend, V.A. Arrieta, KS. Kim, C. Amidei, and R. Stupp are co-authors of an IP filed by Northwestern University related to the content of this manuscript ("CANCER IMMUNOTHERAPIES" No. PCT/US2023/034299). A.M. Sonabend has served as consultant for Carthera and EnClear Therapies. R. Stupp has acted or is acting as a scientific advisor or has served on advisory boards for the following companies: Alpheus Medical, Astra-Zeneca, Boston Scientific, Carthera, Celularity, GT Medical, Insightec, Lockwood (BlackDiamond), Northwest Biotherapeutics, Novocure, Inc., Syneos Health (Boston Biomedical), TriAct Therapeutics, Varian Medical Systems. C. Desseaux, G. Bouchoux, and M. Canney are employees and hold an ownership interest in Carthera. M. Canney, G. Bouchoux, and A. Carpentier have patents related to the ultrasound technology described herein. R. Stupp is an advisory member and consultant for Carthera. A. Carpentier is a consultant for Carthera. All other authors declare no competing interests.

## Additional information

Víctor A. Arrieta[1,2,3], Andrew Gould[1,2], Kwang-Soo Kim[1,2], Karl J. Habashy[1,2], Crismita Dmello [1,2], Gustavo I. Vázquez-Cervantes[1,2], Irina Palacín-Aliana[1,2,4], Graysen McManus [1,2], Christina Amidei[1,2], Cristal Gomez[1,2], Silpol Dhiantravan[1,2], Li Chen[1,2], Daniel Y. Zhang[1,2], Ruth Saganty[1,2], Meghan E. Cholak[5], Surya Pandey[5], Matthew McCord[1,2,6], Kathleen McCortney [1,2], Brandyn Castro[1,2], Rachel Ward[1,2], Miguel Muzzio[7], Guillaume Bouchoux[8], Carole Desseaux[8], Michael Canney[8], Alexandre Carpentier[9], Bin Zhang [5], Jason M. Miska [1,2], Maciej S. Lesniak [1,2], Craig M. Horbinski [1,2], Rimas V. Lukas[2,10], Roger Stupp [1,2,5,10], Catalina Lee-Chang [1,2,11] ✉ & Adam M. Sonabend [1,2,11] ✉

[1]Department of Neurological Surgery, Feinberg School of Medicine, Northwestern University, Chicago, IL, USA. [2]Northwestern Medicine Malnati Brain Tumor Institute of the Lurie Comprehensive Cancer Center, Feinberg School of Medicine, Northwestern University, Chicago, IL, USA. [3]PECEM, Facultad de Medicina, Universidad Nacional Autónoma de Mexico, Mexico City, Mexico. [4]Deparment of Radiation Oncology, Feinberg School of Medicine, Northwestern University, Chicago, IL, USA. [5]Department of Medicine, Division of Hematology and Oncology, Feinberg School of Medicine, Northwestern University, Chicago, IL, USA. [6]Deparment of Pathology, Feinberg School of Medicine, Northwestern University, Chicago, IL, USA. [7]Life Sciences Group, IIT Research Institute, Chicago, IL, USA. [8]Carthera, Lyon, France. [9]Sorbonne Université, Inserm, CNRS, UMR S 1127, AP-HP, Hôpitaux Universitaires La Pitié Salpêtrière—Charles Foix, Service de Neurochirurgie, Paris, France. [10]Department of Neurology, Feinberg School of Medicine, Northwestern University, Chicago, IL, USA. [11]These authors jointly supervised this work: Catalina Lee-Chang, Adam M. Sonabend. ✉e-mail: catalina.leechang@northwestern.edu; adam.sonabend@northwestern.edu

