## [Peer Review File · Nature Communications]

Ultrasound-mediated delivery of doxorubicin to the brain results in immune modulation and improved responses to PD-1 blockade in gliomasREVIEWER COMMENTS

Reviewer #1 (Remarks to the Author): with expertise in brain tumors, cancer immunology

The manuscript by Arrieta et al directly addresses the recognized main limitation to drugs to treat brain tumors, namely the strict limitations imposed by the BBB. The authors directly address this challenge by using a novel method to open the BBB. They test two drugs, DOX, and anti-PD-1, in both rodents AND in human patients. In both they achieve an increase in the amount of drug that enters the brain tumors and the surrounding brain. And in both they achieve indices of increased drug activity, and in mice, they obtain very clinically significant increases in animal survival.

The manuscript is well written, given the complex nature of the topic addressed, and measured in its conclusions.

Questions which the authors may wish to address:

1. Dox is only increased by 2x compared to no BBB opening in patients. How do these drug levels compare to the amount of drug reaching the liver; i.e., after the BBB opening, is the amount of DOX reaching the brain comparable to that reaching peripheral organs -that don't have an organ-blood barrier-, or does the BBB opening need to be further increased?
2. In Figure 1, why is there no p value for 'c'?; please include.
3. What is the total number of patients studied?, two, 3, or four? Please clarify for each figure.
4. For figure 2b,c the t test needs to be two tailed, since the number of cells could presumably go up or down.
5. In Figure 2f,g the bar graphs apparently have significance symbols at the top of some bars, but their meaning is not given in the legend.
6. Figure 3 is very dense and interesting. The authors are to be commended for an elegant comparison of patients and rodents.
7. The results shown in Figure 4 are very impressive, especially Figure 4b. However, it is unclear to which comparisons the p values refer to. Please clarify.
8. GL261 cells are known to be quite immunogenic. If the authors have repeated this experiment with other rodent tumor cells that are known to be less immunogenic and carry mutations shared with human glioblastoma tumors, they should include such data following Figure 4.

9. In Figure 5, the authors change to CT2A cells to study increases in aPD-1 across the BBB; please explain.

10. Figure 5h is very impressive, even if the increase obtained is only doubled. It would be interesting if the authors could demonstrate that even a doubling in the concentration of aPD-1 is effectively clinically relevant.

11. Figure 6 is also very impressive. Please improve the indications of p values, as it is difficult to identify what comparisons they correspond to. It would be interesting if, in the survival curves shown in Figure 6, the authors could indicate, for each curve the values of DOX and aPD-1.

12. The KM in the lower part of Figure 7b is unclear. The colors do not allow to identify exactly which curve is which. Also, a difference shown has a p value of 0.0452. This is a very marginal significance. Could it be that under certain conditions CD8 T cells are actually not needed?

ExtFig 1. the significances shown in b are not given a numerical value.

13. Please discuss the meaning of your data vis-a-vis the failure of checkpoint inhibitors to make any difference in patient survival.

14. The effects of the lack of CD8 T cells appears to be only marginally significant. Please discuss whether CD8 T cells are really necessary in the models used or not.

In summary, an important and elegant manuscript that proposes new ways to treat brain tumors. This is an original and powerful contribution to the treatment of patients suffering from glioblastoma. The manuscript certainly merits the further consideration towards publication.

Reviewer #2 (Remarks to the Author): with expertise in brain tumors, cancer immunology

Arrieta et al report on a translational study combining focused ultrasound with doxorubicin and anti-PD1 treatment in experimental glioblastoma models. The combination of DOX and anti-PD-1 was synergistic in the GL261 and in the CTA2A model. Treatment response was dependent on cytotoxic T cells and resulted in preclinical long-term survivorship. To increase the translational aspect of this study, the authors included tissue analyses from 4 patients with recurrent glioblastoma that had previously received an implantable ultrasound device receiving Pembrolizumab. Some aspects, especially the observation of increased HLA molecule

expression post-treatment was found also in human tissues. Whereas the clinical value of this pilot cohort is limited and there is no control cohort available, it nicely supports some findings of the preclinical models. Overall this is an interesting study inducing clinically-warranted localized inflammation in the highly immunosuppressive tumor microenvironment of glioblastoma.

I have the following questions/comments:

1. Were control animals sham-treated / anesthetized? Please clarify.
2. Taking the higher abundance of CD4+ IFN-g+ TNF-a+ T cells in long term survivors into consideration - does the therapeutic effect also depend on CD4 T cells?
3. Figure 5h: In 2 human tissues there is a 1.5-fold concertation change following LIPU/MB treatment – what is the biological meaning of this – please discuss critically in the discussion section
4. The availability of pre- and post-treatment tumor samples should enable a more sophisticated spatial heterogeneity analysis following LIPU/MB treatment. Is the effect of LIPU/MB highly localized? Are the T cells / microglia in the perivascular areas? Are there TLS-like lesions? One exemplary IF image is not sufficient here. Spatial human analyses could easily be corroborated with tissue analyses from the experimental glioblastoma models.

Minor points:

1. Line 158; According to the predefined level of significance, an increased cell density of SOX2+ HLA-ABC+ is not observed (P=0.0557)
2. Line 241 : According to the predefined level of significance, an increased number of CD163+ cells expressing HLA-ABC in on-treatment GBM is not observed (P=0.0712).

Reviewer #3 (Remarks to the Author): with expertise in brain tumors, ultrasound guided therapy

In this article Arrieta et al. characterized the immunomodulatory effects of low intensity pulsed ultrasound (LIPU) + microbubbles (MBs), which is an established technology to disrupt the BBB and improve drug delivery in brain tumors, in combination with Liposomal Doxorubicin (LipoDox) and then studied the potential of this combination to further improve the efficacy of immune checkpoint blockade (PD-1) in gliomas. The authors showed that LIPU-MBs + LipoDox can induce an IFN-g phenotype in GBM-infiltrating CD11b+ myeloid cells and elicit antitumor immunity in murine glioma tumors. These responses were further augmented by the addition of the immune checkpoint inhibitor aPD-1. Crucially the authors showed that the presence of tumor-infiltrating T cells is required for improved survival. To indicate the clinical relevance of these data the authors showed that the combined treatment can upregulate antigen-presenting molecules in GBM cells and promote an IFN-g+ phenotype in GBM infiltrating T cells in resected tumors from GBM patients. While in isolation the proposed findings are not particularly novel, the application of the combined strategy (LIPU-MBs + LipoDox + aPD-1) to GBM led to very encouraging findings that if they can be replicated in humans can have a very high impact to the treatment and management of GBM. As LIPU-MBs in combination with LipoDox or aPD-1 is currently under clinical investigation by several teams, the findings of this paper can potentially have an immediate clinical impact.

Despite the encouraging findings and rigorous preclinical investigations, the presented clinical data lack controls and most of the data used for statistical analysis are from two patients. The patients have also been treated with paclitaxel in the past and its impact on the observed findings is unknown. Together these limitations challenge the robustness of the observed clinical responses and potentially confound the observed immunological responses.

Past preclinical and clinical investigations that have explored the potential of FUS to improve the delivery of Doxorubicin or antibody-based immunotherapy in brain tumors are either not cited or not critically reviewed. For example, it is not clear how the exposure settings as well as improvement in delivery and survival achieved in the current investigations

compares with past investigations? Are the findings explicit to the specific device and experimental conditions or they support past investigations and therefore have broader value.

There are several questions about the data shown in Figure 1. Most notably the delineation of the peritumoral brain regions that were sonicated versus those that were not sonicated is poor. Acoustic simulations and preferably pre and post treatment DCE MRI data are required to clearly demonstrate that the data are indeed from sonicated or unsonicated regions as authors claim.

In the human studies it is not clear when the Dox infusion started with respect to FUS and how this protocol has been selected and subsequently replicated in mice.

Data assessing the safety of the proposed treatment strategy, especially in healthy brains are needed.

I wonder if the error in the measurements shown in Fig 1d (2.033-fold increase) is 0.001?

The subplots in Fig. 3 not cited correctly in the main text.

In the methods there is a description of experiments with carboplatin (line 1007), but I could not find any data with this drug.

Reviewer #4 (Remarks to the Author): with expertise in brain tumors, immunology

In this report, investigators use ultrasound combined with microbubble administration to open the blood-brain barrier and allow increased access to the brain of delivered compounds, here liposomal doxorubicin and anti-PD1. They report data in patients suffering from GBM and in the GL261 and CT-2A mouse models.

The LIPU/MB technique has been published already and is being tested in clinical trials. The combination of a cytotoxic drug, which activates innate immune responses, with an immune checkpoint inhibitor is exciting and the aim of the study is to increase penetrance of both

compounds in the brain. A strength of the study is the availability of pre- and post-treatment tumor samples (in all 4 patients), with, in 2 of them, availability of peritumor areas to assess compound influx in sonicated vs. non-sonicated areas.

Overall, this study carefully investigates the potential of the approach, which is of high interest and timely. However, there are concerns that have to be addressed before this work can be considered for publication.

Major comments:

1. Figure 1: why is a violin plot used in panel c whereas all other panels use bar plots? Data should be presented as bar plots in panels c, d, f and g or the reason for doing differently explained. In addition, how many mice were used in the experiment, and which points correspond to each time point (45 minutes, 24 and 48h)? Finally, in panels c, d, f, g: would it be possible to identify to which of the 2 patients/mice the dots originate, in order to see variation within sonicated vs. non-sonicated brain areas in given individuals?
2. Figure 2: is absence of SOX2 expression in one patient indicative of absence of tumor in that sample? Would it be possible to use another tumor marker (such as GFAP) instead of SOX2 to identify tumor cells, in order potentially to be able to analyze all 4 patients? In addition, it is not correct to say that there is an increase in SOX2+ HLA-ABC+ cells if the p value is above 0.05 (defined as such in the methods). This should be changed and the title and concluding sentence of the paragraph should be tempered. Finally, are HLA class I and II upregulated in non-tumor cells as well?
3. In addition to data presented in Figure 2f and g, would the observed levels of increase in MHC class I upregulation translate into efficient peptide presentation and tumor cell recognition by T cells? Is that testable (have T cell epitopes in GBM6/GBM63 been described or are other models available to test that)?
4. The gating of the macrophage population in Figure 3 is unusual. Whereas the CD45++ CD11b++ population has been described as macrophages, the classification of the CD11b intermediate population as macrophages, with the gate expanding far towards lymphocytes is not typical (Gabrusiewicz et al., JCI Insights 2016, Brandenburg et al. IJMS 2020, Khan et al., JCI 2023 among others). Do authors have confirmation that this population is macrophages? Would the results be different if the gating was restricted to the CD45++ CD11b++ population?

5. How was the setting of the positive gate for IFN-g production as shown in Figure 3d determined? It seems to cut the population in two without a clear negative and positive population being visible. An isotype control should be used or the gate be set at the right of the population in the control condition. Data in panels b, c and d should be reanalyzed once the gating has been modified. The same applies to Figure 4e-g. Similarly, isotypes should be used for assessment of TNF-a and IL-1b secretion as well as of H2-Kb and PD-L1 expression (extended data fig 2 and 3).
6. In the experiment shown in Figure 3 (a-e), were the number or percentages of macrophages and microglial cells modified upon treatment? (This is shown for lymphoid cells but not for myeloid cells.)
7. Figure 3 f: how do authors explain increase in MHC in microglial cells with 5 mg/kg of DOX in absence of LIPU/MB whereas no IFN-g is detected in that condition (panel b)? What are the other possible mechanisms in play here?
8. The observation that type II IFN is secreted by myeloid cells, including microglia, is not common. Measure of type I IFN would be informative in that regard, also as authors did not observe induction of inflammatory cytokines such as IL-1b and TNF-a.
9. Related to extended data Figure 4, could authors show an exemplary staining of secretion of cytokines by macrophages? There is in addition no description of these analyses (related to Figure 3 and extended data Figures 2 and 4 panels a-c) in the methods (no mention either of measure of TNF-a, IL-1b or GZMb production in the methods).
10. How do authors explain that DOX has no effect on peripheral immune cells? In addition, the high proportion of macrophages producing IFN-g in all conditions, even the control ones is unexpected. How can this be explained?
11. In Figure 3 panel l and m, authors show increase of IFN-g+ TMEM119+ cells in human GBM samples after treatment. However, they do not mention the results regarding IFN-g+ CD163+ cells. Could this be added, also as it is further studied in panels n and o?
12. Please specify the number of mice used in each treatment group in Figure 4 (the numbers are difficult to reconcile) as well as in other experiments where it is not specified.
13. At several occasions (figure 2b, Extended data, Figure 4e, figure 4c, Figure 6d), authors say that there is an increase in a measured parameter although the p value is above 0.05. Strictly speaking, this is not correct and authors should modify the way they report it (also in the abstract regarding HLA I in patient samples). The observations are interesting and the p

value likely linked to the limited number of patients or mice analyzed.

14. An interesting experiment to confirm the role of myeloid cells in the effect of the LIPU/MB/DOX/anti-PD1 treatment
15. How do the authors explain the fact that DOX does not induce IFN-g secretion by CD4 or CD8 T cells in absence of tumor (extended data Fig 3)?
16. How can the differences in survival observed in the cd8+/+ mice treated with CT-2A in figure 6d (80% survival with LIPU/MB+DOX+aPD-1) vs. figure 7b (40% survival) be explained?
17. DOX has been shown in past studies to induce IFN-g-secreting T cells, which is not observed here (Figure 3). Is it due to a difference in the models used?

Minor comments:

1. Methods: I assume that blood drawing was performed in patients at the time of biopsy/resection, but this could be specified in the methods.
2. Figure 1c, d, f, g: would it be possible to identify to which of the 2 patients/mice the dots originate, in order to see variation within sonicated vs. non-sonicated brain areas in given individuals?
3. Figure 2: do authors have an explanation why high doses of DOX do not increase and even decrease class I and class II expression in some instances?
4. Legend to extended data Fig 2: please correct: Microglia were gated based on CD45- and CD11b- (should be CD11b+). In addition, they are usually CD45dim, although the brightness of the staining might make them CD45- here.
5. Legend to extended data Fig 3: please correct, panel a is not present as described.
6. The caption for panel e is missing in Figure 3.
7. Figure 3, legends to panels m, n and o have been exchanged.

REVIEWER COMMENTS

Reviewer #1 (Remarks to the Author): with expertise in brain tumors, cancer immunology

The manuscript by Arrieta et al directly addresses the recognized main limitation to drugs to treat brain tumors, namely the strict limitations imposed by the BBB. The authors directly address this challenge by using a novel method to open the BBB. They test two drugs, DOX, and anti-PD-1, in both rodents AND in human patients. In both they achieve an increase in the amount of drug that enters the brain tumors and the surrounding brain. And in both they achieve indices of increased drug activity, and in mice, they obtain very clinically significant increases in animal survival. The manuscript is well written, given the complex nature of the topic addressed, and measured in its conclusions.

Questions which the authors may wish to address:

- 1. Dox is only increased by 2x compared to no BBB opening in patients. How do these drug levels compare to the amount of drug reaching the liver; i.e., after the BBB opening, is the amount of DOX reaching the brain comparable to that reaching peripheral organs -that don't have an organ-blood barrier-, or does the BBB opening need to be further increased?**

In our study, we primarily focused on the brain tissue and did not collect other organs for pharmacokinetic analysis. However, previous preclinical studies, such as the one conducted by Gaillard et al., have characterized the biodistribution of liposomal DOX¹. In their study using a 5 mg/kg dose of radiolabeled liposomal DOX in mice, a significant disparity was observed between the brain (< 0.1 %ID/gram tissue) and other organs (approximately 1-10 %ID/gram tissue). In our research, we noted a four-fold increase in DOX concentration in the brain following LIPU/MB treatment. Extrapolating from these preclinical findings, it's plausible that our delivery strategy could achieve DOX concentrations in the brain that are more similar to those in other organs, such as the muscle, liver, and spleen.

It's important to acknowledge that direct comparisons between these preclinical findings and our study are limited due to differences in experimental designs, including assessment methods (radio-labeling versus direct measurement) and timing between administration and assessment (21 hours versus 48 hours).

Regarding clinical data, there is a notable gap in literature directly comparing DOX concentrations in the brain and peripheral organs in humans. This is partly due to the challenges associated with measuring drug concentrations in these different tissues in clinical settings.

- 2. In Figure 1, why is there no p value for 'c'?; please include.**

Thank you for your suggestion. We have included the P value for this figure and have moved it to Extended Figure 1 of the revised manuscript.

- 3. What is the total number of patients studied?, two, 3, or four? Please clarify for each figure.**

For the study on concentrations of DOX in peritumoral brain regions in GBM patients we analyzed 2 patients, (a total of 12 peritumoral brain samples: 8 sonicated and 4 non-sonicated). For the immune response characterization by multiplex immunofluorescence analysis, we have included 4 patients (with pre and on-treatment tumor samples). We have made sure to specify the number of patient samples in both the figure legends and the manuscript for each figure and analysis to enhance clarity.

- 4. For figure 2b,c the t test needs to be two tailed, since the number of cells could presumably go up or down.**

Thank you for your insightful comment regarding the statistical approach for Figure 2b,c. Initially, we employed a one-tailed t-test in our analysis based on prior scientific evidence suggesting that DOX increases the expression of MHC class I^{2,3}. Upon applying a 2-tail t-test, Figure 2b p=0.053, and Figure 2c p=0.069). These p values are the reflection of the small number of patients, and of the fact that the data has been greatly reduced as for this calculation, we averaged all cells for every patient. To better analyze these results with the full dataset of observations, and account for both within-patient (time-dependent) and between-patient variability, we conducted a more comprehensive analysis using a mixed-effects model. Our dataset presents inherent biological variability among patients. By including 'patient' as a random effect in the mixed-effects model, we can adjust for these individual differences, providing a more accurate reflection of the treatment's impact. In addition, the mixed-effects model offers a robust framework for analyzing complex data structures, such as ours. It evaluates both fixed effects (like DOX treatment) and random effects (patient-specific variations), making it a superior choice over a standard t-test for our study. We used this statistical framework for all the multiplex immunofluorescence analyses comparing pre-treatment and on-treatment GBM samples in the manuscript. Additionally, by further optimizing the staining with SOX2, in the revised manuscript we were able to include all 4 patients for the analysis of SOX2+ HLA ABC+ cells and SOX2+ HLA-DR+ cells.

With this in mind, our mixed-effects model analysis indicated a statistically significant increase in the cell density of SOX2+ HLA ABC+ cells ($P = 0.0079$, **Fig. 1a of this document**) and SOX2+ HLA-DR+ cells ($P = 0.024$, **Fig. 1b of this document**) in on-treatment GBM samples compared to pre-treatment GBM samples. These findings support the immunogenic impact of DOX on GBM cells. These updates were made to Figure 2 of the revised manuscript.

Figure 1. DOX upregulates antigen-presenting molecules in human GBM cells. a, b, Dot plots showing the cell density of SOX2+ HLA-ABC+ (a) and SOX2+ HLA-DR+ (b) cells in pre-treatment GBM samples and on-treatment GBM samples. n = 4 paired GBM samples. A mixed effects model was constructed considering DOX+aPD-1 treatment as a fixed effect and patients as a random effect influencing the cell density of SOX2+ HLA-ABC+ and SOX2+ HLA-DR+. P values were obtained by chi-squared tests of the likelihood ratio tests of the full model with the fixed effect against the model without the fixed effect.

5. In Figure 2f,g the bar graphs apparently have significance symbols at the top of some bars, but their meaning is not given in the legend.

Thank you for pointing out the oversight in Figure 2f,g of the revised manuscript. We have now clarified their meaning and included an explanation in the figure legend.

6. Figure 3 is very dense and interesting. The authors are to be commended for an elegant comparison of patients and rodents.

We appreciate your positive remarks on this figure and the parallel analysis across species.

7. The results shown in Figure 4 are very impressive, especially Figure 4b. However, it is unclear to which comparisons the p values refer to. Please clarify.

To address this, we have now included brackets in Figure 4b and the rest of the figures of the revised manuscript involving survival analysis to clearly denote the group comparisons for which the p values are applicable.

8. GL261 cells are known to be quite immunogenic. If the authors have repeated this experiment with other rodent tumor cells that are known to be less immunogenic and carry mutations shared with human glioblastoma tumors, they should include such data following Figure 4.

In response to the comment, we expanded our investigation to include the CT2A murine glioma cell line, known for its *Pten* loss, a mutation relevant to human GBM and associated with resistance to PD-1 blockade in GBM patients^{4,5}. Our treatment approach, consistent with that applied to the GL261 model, did not yield a significant survival advantage in CT2A models. Specifically, while GL261 models showed an induction of approximately 80% of long-term survivors with DOX delivered via LIPU/MB, the CT2A model did not exhibit a similar survival benefit (**Fig. 2 of this document**). This indicated that while LIPU/MB significantly enhances DOX efficacy in the GL261 model, in the CT2A model it may require a combination with immunotherapy to achieve efficacy as we show in Fig. 6 of the revised manuscript.

Figure 2. Effect of DOX delivery to the brain by LIPU/MB-based BBB opening on survival of CT2A-bearing mice. Kaplan-Meier curve representing the survival outcomes of CT2A-bearing mice treated with liposomal DOX with and without LIPU/MB. P value derived from Log-rank test.

9. In Figure 5, the authors change to CT2A cells to study increases in aPD-1 across the BBB; please explain.

The CT2A glioma model is particularly relevant for our study as it is known for its resistance to immunotherapy. This resistance makes CT2A cells stringent system for assessing whether the increased penetration of the immune checkpoint blockade antibody can enhance its efficacy.

10. Figure 5h is very impressive, even if the increase obtained is only doubled. It would be interesting if the authors could demonstrate that even a doubling in the concentration of aPD-1 is effectively clinically relevant.

We thank the reviewer for acknowledging the potential impact of the results depicted in Figure 5h of the revised manuscript. Unfortunately, the scope of our manuscript and the clinical dataset we have will fall short of demonstrating the clinical efficacy related to this increase in penetration of aPD-1. Yet, we kindly ask the reviewer to consider the following:

- 1) We could measure the antibody in the human brain 2 days after sonication and delivery, similar analysis in mice within hours of sonication showed 4-6 fold increases. In other studies, we have shown a 4-6 fold increase for multiple drugs, such as carboplatin, and most importantly albumin-bound paclitaxel (which is a large compound as it contains a protein), in the human brain following sonication⁶. Moreover, other studies have also shown that LIPU/MB can increase the concentration of radio-labeled antibodies in the human brain⁷.
- 2) The observed doubling of aPD-1 concentration, while seemingly modest, is likely to increase its efficacy considering the challenging nature of drug delivery across the blood-brain barrier (BBB). Pharmacokinetic human studies showed that marginal increases in the concentration of therapeutic antibodies in the CSF are effective for blocking PD-1 in T cells⁸.

11. Figure 6 is also very impressive. Please improve the indications of p values, as it is difficult to identify what comparisons they correspond to. It would be interesting if, in the survival curves shown in Figure 6, the authors could indicate, for each curve the values of DOX and aPD-1.

In response, we have included brackets in Figure 6 of the revised manuscript to more clearly denote the specific group comparisons for which the p values are presented. We hope that this aids in removing any ambiguity regarding the statistical analyses. In addition, we have included the p values for the comparisons between DOX and aPD-1 curves.

12. The KM in the lower part of Figure 7b is unclear. The colors do not allow to identify exactly which curve is which. Also, a difference shown has a p value of 0.0452. This is a very marginal significance. Could it be that under certain conditions CD8 T cells are actually not needed?

We thank the reviewer for their insightful observations on Figure 7b. In response to the formatting concern with the colors, we opted to remove groups not treated with LIPU/MB. This decision was made to simplify the visual presentation and to direct attention to the most relevant comparisons for assessing the role of CD8⁺ T cells in our combinatorial treatment strategy.

Additionally, we like to clarify our mislabeling of the original p value for this figure, as it represented the overall comparison of the four initial groups included in the analysis. However, a specific comparison between the *Cd8^{+/+}* DOX+aPD-1+LIPU/MB group and the *Cd8^{-/-}* DOX+aPD-1+LIPU/MB group using the log-rank test led to a p-value of 0.0104. Yet, to confirm the reproducibility of the original result, we repeated the survival experiment, including a cohort of mice devoid of CD8⁺ T cells alongside the original wild-type mice. The new result derived from this survival study provides a more conclusive insight into the significance of CD8⁺ T cells in the efficacy of the DOX plus aPD-1 treatment delivered with LIPU/MB. This new survival study yielded a more compelling statistical significance ($P < 0.0001$), which is presented in **Fig. 3 of this document**.

Figure 3. LIPU/MB-mediated delivery of liposomal DOX and aPD-1 influences and requires tumor-infiltrating CD8 T cells. Kaplan Meier curve showing the survival of CT2A-bearing mice treated with the combination of liposomal DOX and aPD-1 delivered with LIPU/MB in wild type and *Cd8^{-/-}* genetic backgrounds. *P* value was derived from the log-rank test.

ExtFig 1. the significances shown in b are not given a numerical value.

In response, we have included the numerical *p* values on top of the bar plots for this figure (Fig 4 of this document, and Extended Data Fig 2 of the revised manuscript).

Figure 4. DOX upregulates antigen-presenting molecules in GBM. Bar plots representing the expression of H2-K^b assessed as MFI values in GL261 and CT2A. *P* values were obtained by one-way ANOVA with post hoc Dunnett’s multiple comparisons test.

13. Please discuss the meaning of your data vis-a-vis the failure of checkpoint inhibitors to make any difference in patient survival.

In response, we have added the following paragraph to the manuscript discussion section:

“Phase III clinical trials have consistently demonstrated the limited efficacy of anti-PD-1 immune checkpoint inhibitors in treating unselected GBM populations⁹⁻¹¹. However, there is evidence reporting sustained benefits in a subset of GBM patients^{5,12,13}. Our study and several other lines of evidence suggest that the lack of efficacy observed in these trials can be partially attributed to poor penetration of antibodies to the brain, the relative weakness of aPD-1 as monotherapy in the context of an immunosuppressive microenvironment in GBM, and biological differences across tumors that render some cases more susceptible to others^{12,14,15}. In this study, we focused on the modulation of the immunosuppressive microenvironment through the use of DOX, and the delivery challenge posed by the BBB, through the use of LIPU/MB. Indeed, despite a recent clinical study reporting sufficient concentration of pembrolizumab in the CSF to block PD-1 on endogenous T cells and CAR T cells⁸, it remains unclear whether these

antibodies effectively penetrate brain tissue. We measured the concentration of aPD-1 within tumors and peritumoral tissues in GBM patients. Given previous findings of immune checkpoint blockade inducing a clonal remodeling of peripheral T cells¹⁶, our strategy to employ LIPU/MB for brain-directed aPD-1 delivery was designed to maintain T cell activity once they infiltrate the brain by preventing interactions between PD-1 and their ligands PD-L1/2 expressed by both tumor and myeloid cells.”

14. The effects of the lack of CD8 T cells appears to be only marginally significant. Please discuss whether CD8 T cells are really necessary in the models used or not.

We appreciate the reviewer's interest in the significance of CD8⁺ T cells in our models. As mentioned in our response to a previous comment, we conducted additional experiments. This new survival study yielded a more robust statistical significance ($P < 0.0001$; **Fig. 3 of this document**), suggesting a substantial role of CD8⁺ T cells in the efficacy of our combinatorial treatment involving DOX plus aPD-1 delivered with LIPU/MB. The increased statistical power from the initial and the new survival studies underscores the importance of CD8⁺ T cells in the treatment response.

The presence of CD8⁺ T cells is widely recognized as critical in mediating anti-tumor immunity, particularly in the context of immunotherapies like PD-1 blockade. CD8⁺ T cells are known for their ability to directly kill tumor cells and for their role in creating a broader immune response. Our model aligns with this understanding, as the reduced efficacy observed in the absence of CD8⁺ T cells indicates their functional importance in our treatment strategy. Furthermore, the upregulation of MHC molecules by tumor cells following DOX treatment aligns with the ability of T cells to recognize tumor cells for killing.

In summary, an important and elegant manuscript that proposes new ways to treat brain tumors. This is an original and powerful contribution to the treatment of patients suffering from glioblastoma. The manuscript certainly merits the further consideration towards publication.

Thank you for your feedback and recognition of the significance of our work. Your positive endorsement and suggestions are greatly appreciated.

Reviewer #2 (Remarks to the Author): with expertise in brain tumors, cancer immunology

Arrieta et al report on a translational study combining focused ultrasound with doxorubicin and anti-PD1 treatment in experimental glioblastoma models. The combination of DOX and anti-PD-1 was synergistic in the GL261 and in the CT2A model. Treatment response was dependent on cytotoxic T cells and resulted in preclinical long-term survivorship. To increase the translational aspect of this study, the authors included tissue analyses from 4 patients with recurrent glioblastoma that had previously received an implantable ultrasound device receiving Pembrolizumab. Some aspects, especially the observation of increased HLA molecule expression post-treatment was found also in human tissues. Whereas the clinical value of this pilot cohort is limited and there is no control cohort available, it nicely supports some findings of the preclinical models. Overall this is an interesting study inducing clinically-warranted localized inflammation in the highly immunosuppressive tumor microenvironment of glioblastoma.

I have the following questions/comments:

1. Were control animals sham-treated / anesthetized? Please clarify.

We appreciate your comment and question. For the groups treated with liposomal doxorubicin (Doxil), we did not include a comparison group receiving only the liposomal vehicle due to the unavailability of empty liposomes from the commercial Doxil formulation. Nonetheless, we employed the same procedural conditions as the treatment groups, including anesthesia and surgeries.

In the case of our immunotherapy studies, control animals received an isotype antibody control to match for any non-specific immune modulations. We believe this control is stringent and have detailed this approach in the methods section.

2. Taking the higher abundance of CD4⁺ IFN- γ ⁺ TNF- α ⁺ T cells in long term survivors into consideration - does the therapeutic effect also depend on CD4 T cells?

Thank you for your insightful question regarding the role of CD4⁺ T cells in the therapeutic efficacy observed in our study.

The increased abundance of CD4⁺ IFN- γ ⁺ TNF- α ⁺ T cells following our combinatorial therapy of liposomal DOX, aPD-1, and LIPU/MB does suggest a contributory role of these cells in mediating anti-tumor immunity.

To dissect the contributions of different immune cells, we conducted a survival experiment using mice with intact immunity and mice with different immune deficiencies: *Cd8*^{-/-} mice, lacking CD8⁺ T cells, and *Rag1*^{-/-} mice, lacking both T cells and B cells (**Fig. 5 of this document**, Fig. 7d of the revised manuscript). Our findings indicate that *Rag1*^{-/-} mice had reduced survival compared to *Cd8*^{-/-} mice when treated with the combinatorial therapy. This suggests that while CD8⁺ T cells are important, the presence and function of CD4⁺ T cells and B cells also play a significant role in the observed therapeutic effect. The synergistic action of B cells, potentially in antibody production and antigen presentation, in conjunction with CD4⁺ T cells, may enhance the cytotoxic activity of CD8⁺ T cells.

We have added a discussion of the potential role of CD4⁺ T cells and B cells in the manuscript to reflect this consideration.

Figure 5. CD4⁺ T cells and B cells contribute to the efficacy of LIPU/MB in CT2A-bearing mice. Kaplan-Meier curve representing the survival experiment and outcomes of CT2A-bearing mice in the context of wild type, *Cd8*^{-/-} and *Rag1*^{-/-} backgrounds treated with liposomal DOX and aPD-1 with LIPU/MB. P value derived from Log-rank test.

3. Figure 5h: In 2 human tissues there is a 1.5-fold concertation change following LIPU/MB treatment – what is the biological meaning of this – please discuss critically in the discussion section

Thank you for your valuable observation regarding the increase in DOX and pembrolizumab concentrations. We have updated our discussion to include an expanded analysis of these findings:

“Our study underpins the potential benefits of combining DOX with immune checkpoint inhibitors in the treatment of GBM. An approximately 2-fold increase in the concentration of both DOX and pembrolizumab was observed, which corresponds to the concentration range that facilitates MHC molecule upregulation and immune modulation in both tumor and myeloid cells in controlled experiments that were performed *in vitro*. Although the limited number of GBM patients prevents us from investigating whether the drug concentration increases translate into clinical efficacy, such an increase is nonetheless suggestive of enhanced drug delivery and potential therapeutic advantage. The efficacy of DOX, when administered with LIPU/MB, extends beyond simple drug delivery; it notably modulates the TME. It induces an IFN- γ phenotype in T cells and in GBM-associated microglia and macrophages and enhances the expression of antigen-presenting molecules, including HLA-ABC and HLA-DR. Such DOX-mediated upregulation potentially improves the presentation of tumor antigens to T cells, thereby amplifying the effectiveness of T cell-based immunotherapies, including PD-1 blockade.”

4. The availability of pre- and post-treatment tumor samples should enable a more sophisticated spatial heterogeneity analysis following LIPU/MB treatment. Is the effect of LIPU/MB highly localized?

We thank the reviewer for their insightful inquiry regarding the spatial aspects of the LIPU/MB effects. We would like to clarify that the ultrasound technology employed in our study differs from other sonication methods such as focused ultrasound (FUS). The implantable ultrasound device used in our research is designed to disrupt the BBB over a broader volume, allowing for extensive delivery of therapeutic agents, as opposed to a highly localized target area with FUS.

Consequently, the effects of LIPU/MB we report are not localized to a small focal point but are distributed across a larger region within the brain. This characteristic of the implantable ultrasound device is crucial for treatments aimed at diffusely infiltrative tumors such as GBM, where widespread BBB opening may be more beneficial.

We also want to clarify that our phenotypic analysis was conducted on available tumor regions without specific annotations for sonicated versus non-sonicated areas. We prioritized the limited numbers of brain biopsy samples for pharmacokinetic measurements to quantitatively determine the effect of LIPU/MB on drug delivery into the brain. However, we acknowledge the potential insights that could be gained from a spatial heterogeneity analysis and recognize it as a valuable direction for future research.

Are the T cells / microglia in the perivascular areas? Are there TLS-like lesions?

One exemplary IF image is not sufficient here. Spatial human analyses could easily be corroborated with tissue analyses from the experimental glioblastoma models.

In response to this question, we implemented a multiplex immunofluorescence panel to assess the spatial distribution of T cells, B cells, and microglia in our paired tumor samples. We included CD31 as an endothelial cell marker to identify perivascular areas. This allowed us to quantify the average number of T cells (CD3+), B cells (CD20+), macrophages (CD163+), and microglia (TMEM119+) within a 15 μm radius relative to CD31+ endothelial cells (**Fig. 6a of this document**). These analyses were performed on pre-treatment and on-treatment tumor samples from three GBM patients. We utilized all the available tumor tissue from the fourth patient for our analysis, leaving no remaining samples for further testing. The results, as illustrated in **Fig. 6b of this document**, did not reveal significant differences in immune cell clustering in perivascular areas between these two time points.

Figure 6. Immune cell clustering around endothelial cells in GBM samples treated with DOX plus aPD-1 delivered with LIPU/MB. Dot plots representing the average number of immune cells (CD3⁺, CD20⁺, CD163⁺, and TMEM119⁺) towards CD31⁺ cells in a radius of 15 μm . P values were obtained by paired t -tests.

Minor points:

1. Line 158; According to the predefined level of significance, an increased cell density of SOX2+ HLA-ABC+ is not observed ($P=0.0557$)

Thank you for your observation. The initial analysis with a one-tailed t -test yielded a p -value of 0.0557, slightly above the significance threshold. However, a subsequent re-analysis using a mixed-effects model, better suited for our data's paired and longitudinal structure, revealed a statistically significant effect of DOX on SOX2+ HLA-ABC+ cell density ($P = 0.0079$; **Fig. 1a of this document**). This model's capability to account for patient-specific variations provides a more robust and accurate interpretation of our results.

2. Line 241 : According to the predefined level of significance, an increased number of CD163+ cells expressing HLA-ABC in on-treatment GBM is not observed ($P=0.0712$).

Thank you for your attention to the statistical details of CD163+ HLA-ABC+ cell numbers in our study. Upon reanalysis with a mixed-effects model, acknowledging the inherent variability in our data, we observed a p -value of 0.066 (**Fig. 7 of this document**), suggesting a trend towards increased expression post-treatment with DOX. Nonetheless, we agree with the reviewer and have changed the claim of this result in our manuscript to reflect the non-statistical significance of this result.

Figure 7. Effect of liposomal DOX in CD163⁺ cells in human GBM samples. Dot plot showing the cell density of CD163⁺ HLA-ABC⁺ cells in pre-treatment GBM samples and on-treatment GBM samples. n = 4 paired GBM samples. A mixed effects model was constructed considering DOX+aPD-1 treatment as a fixed effect and patients as a random effect influencing the cell density of CD163⁺ HLA-ABC⁺. P values were obtained by a chi-squared test of the likelihood ratio test of the full model with the fixed effect against the model without the fixed effect.

Reviewer #3 (Remarks to the Author): with expertise in brain tumors, ultrasound guided therapy

In this article Arrieta et al. characterized the immunomodulatory effects of low intensity pulsed ultrasound (LIPU) + microbubbles (MBs), which is an established technology to disrupt the BBB and improve drug delivery in brain tumors, in combination with Liposomal Doxorubicin (LipoDox) and then studied the potential of this combination to further improve the efficacy of immune checkpoint blockade (PD-1) in gliomas. The authors showed that LIPU-MBs + LipoDox can induce an IFN-g phenotype in GBM-infiltrating CD11b⁺ myeloid cells and elicit antitumor immunity in murine glioma tumors. These responses were further augmented by the addition of the immune checkpoint inhibitor aPD-1. Crucially the authors showed that the presence of tumor-infiltrating T cells is required for improved survival. To indicate the clinical relevance of these data the authors showed that the combined treatment can upregulate antigen-presenting molecules in GBM cells and promote an IFN-g⁺ phenotype in GBM infiltrating T cells in resected tumors from GBM patients. While in isolation the proposed findings are not particularly novel, the application of the combined strategy (LIPU-MBs + LipoDox + aPD-1) to GBM led to very encouraging findings that if they can be replicated in humans can have a very high impact to the treatment and management of GBM. As LIPU-MBs in combination with LipoDox or aPD-1 is currently under clinical investigation by several teams, the findings of this paper can potentially have an immediate clinical impact.

Despite the encouraging findings and rigorous preclinical investigations, the presented clinical data lack controls and most of the data used for statistical analysis are from two patients.

We thank the reviewer for their comment. In response to the concerns raised, we have included a control cohort of eight paired GBM samples at initial recurrence and second recurrence to mirror and provide a baseline comparison for our experimental cohort that received DOX between recurrences (**Fig. 8a of this document**). We performed multiplex immunofluorescence evaluating the same markers that we tested in the tumor samples treated with DOX.

Our control cohort analysis revealed no significant differences in the cell densities of SOX2⁺ HLA-ABC⁺, SOX2⁺ HLA-DR⁺ (**Fig. 8b of this document**), TMEM119⁺ IFN- γ ⁺, CD163⁺ IFN- γ ⁺ (**Fig. 8c of this document**), TMEM119⁺ HLA-ABC⁺ and CD163⁺ HLA-ABC⁺ (**Fig. 8d of this document**) between the first and second recurrences. This suggests that the observed changes in the experimental cohort are likely due to the intervention rather than the disease's natural progression or treatment-related effects. The

inclusion of the control cohort in our analysis underscores the potential immunomodulatory impact of DOX when administered between recurrences.

We have discussed the implications of these control cohort findings within our manuscript and believe that they lend further credence to the potential clinical efficacy of DOX in modulating the tumor and immune microenvironment in GBM.

Figure 8. Immunophenotyping analysis of a control cohort of GBM patient samples at two stages of disease progression. a, Schematic representation of the patient cohort and the surgical timeline. Eight GBM patients underwent surgery at initial recurrence and second recurrence. **b,** Quantitative analysis of SOX2⁺ HLA-ABC⁺ and SOX2⁺ HLA-DR⁺ cell densities in GBM samples from the first and second recurrences. **c,** Quantitative analysis of TMEM119⁺ IFN- γ ⁺ and CD163⁺ IFN- γ ⁺ cell densities in GBM samples from the first and second recurrences. **d,** Analysis of TMEM119⁺ HLA-ABC⁺ and CD163⁺ HLA-ABC⁺ cell densities in GBM samples from first and second recurrences. Each point represents the cell density from an individual patient sample, with lines connecting samples from the same patient across different recurrences. A mixed effects model was constructed considering DOX+aPD-1 treatment as a fixed effect and patients as a random effect influencing the cell density of the evaluated phenotypes. *P* values were obtained by chi-squared tests of likelihood ratio tests of the full model including the fixed effect in question against the model without the fixed effect.

The patients have also been treated with paclitaxel in the past and its impact on the observed findings is unknown. Together these limitations challenge the robustness of the observed clinical responses and potentially confound the observed immunological responses.

We appreciate the reviewer's insightful comments and recognize the importance of considering previous treatments in our analysis. Our study elucidated the novel IFN- γ phenotype induction in human brain-infiltrating myeloid cells following DOX exposure, a finding that has not been previously documented in the context of GBM. This observation is significant as it underscores DOX's potential to synergize with and enhance the efficacy of immunotherapeutic strategies such as anti-PD-1 antibodies in the treatment of GBM.

We acknowledge that prior chemotherapy with paclitaxel may influence the immune landscape of GBM and have added this to the discussion of our manuscript as a potential limitation to the generalizability of the findings of our human analysis. Yet, please consider that our observations of DOX's effects in both human and murine models provide evidence that these immunomodulatory effects can be attributed to DOX specifically. This is supported by the mechanistic insights gathered from our study, which align with the

known pharmacodynamic actions of DOX, distinct from those of paclitaxel. Moreover, the parallel responses observed in our animal models, which were not subject to previous chemotherapeutic regimens, further supports that the immunological effects noted are related to DOX treatment.

Past preclinical and clinical investigations that have explored the potential of FUS to improve the delivery of Doxorubicin or antibody-based immunotherapy in brain tumors are either not cited or not critically reviewed. For example, it is not clear how the exposure settings as well as improvement in delivery and survival achieved in the current investigations compares with past investigations? Are the findings explicit to the specific device and experimental conditions or they support past investigations and therefore have broader value.

In addressing the reviewer's insightful comment on the comparison of our study's ultrasound sonication parameters with those in prior research, we have reviewed the existing literature to contextualize and enhance the significance of our findings. To this end, we have enriched the discussion section of our manuscript with a new paragraph that critically compares our methodology and results with previous investigations. The revised paragraph is as follows:

“Like our study, other previous preclinical reports relied on FUS and DOX for brain tumor treatment^{17,18}. Similarly, previous clinical studies have employed ultrasound-mediated disruption of the BBB to facilitate the delivery of chemotherapeutic agents like DOX and antibodies in the treatment of brain tumors^{7,19,20}. However, these studies, including the sole clinical investigation that evaluated ultrasound-based BBB disruption for delivering DOX in a GBM patient¹⁹, primarily utilized MR-guided transcranial focused ultrasound (FUS). The technology we employed in this study relies on an implantable ultrasound device that in contrast to MR-guided transcranial FUS, the sound waves do not require penetrating across the skull. Therefore, ultrasound waves at 1 MHz can be used in this implantable approach, allowing it to target a large volume at once. This differs from transcranial FUS where a much lower frequency (typically 220 kHz) is used to pass the bone more efficiently, and a small focal volume is scanned across the targeted region²¹.”

There are several questions about the data shown in Figure 1. Most notably the delineation of the peritumoral brain regions that were sonicated versus those that were not sonicated is poor. Acoustic simulations and preferably pre and post treatment DCE MRI data are required to clearly demonstrate that the data are indeed from sonicated or unsonicated regions as authors claim.

In response, we like to clarify that a peri-tumoral brain sample being sonicated versus non-sonicated was meticulously determined by the senior author and neurosurgeon for these cases, based on whether the peri-tumoral brain was immediately deep to an ultrasound emitter (sonicated) or in a region of the brain that was not covered by the ultrasound device (non-sonicated). This is easily seen in surgery, and once the device is explanted, it can also be confirmed using the stereotaxic navigator, as the ultrasound position appears in the MRI used for navigation. Thus, to determine this with high certainty, the neurosurgeon relied on intraoperative navigation once the ultrasound device was removed, as the MRI used for this navigation clearly shows each of the ultrasound emitters. We and others have described the ultrasound beam/field of sonication as a circular prism that is the diameter of the emitter, and approximately 7 cm deep^{6,22}. In response to the reviewer's request, we now show below the typical field of sonication determined by contrast MRI obtained shortly after LIPU/MB as we previously reported (**Fig. 9a of this document**⁶), and also on the figure below, as requested, we show the acoustic simulations for the ultrasound field of the cases we used for the drug concentration analysis presented on the manuscript.

Due to tumor recurrence outside of the sonication field, the original cranial window used for the ultrasound implant was expanded for further resection during the surgery where we obtained non-sonicated samples, exposing the peri-tumoral brain that was not sonicated (we present the snapshots of the stereotaxic

navigation for the sites where we obtained these biopsies on **Fig. 9b of this document, and in Fig. 1 of the manuscript**). Whereas below we also present DCE MRI images to further demarcate and validate the sonicated regions of the brain following LIPU/MB (**Fig. 9a of this document**), it is important to note that the MRI following sonications were acquired as part of a clinical trial months before patients having disease recurrence, and undergoing LIPU/MB with DOX and aPD-1, when the tumor size and location was different from the initial post-sonication MRI. We do not have a post-sonication MRI obtained right after LIPU/MB for delivery of DOX and aPD-1 prior to surgery during this therapy (when we obtained the biopsies), as this was not done as part of a clinical trial, but as part of an expanded-access protocol.

Thus, in summary, we determined whether the peri-tumoral brain specimen was recently sonicated or not, based on whether it was located immediately deep to the ultrasound device or not, given the cylindrical field of sonication from the Sonocloud-9 ultrasound device. Additional research-related procedures like intraoperative sonication or post-sonication MRI just before resection were not done and were not possible given that these fall outside of the scope of the expanded access protocol. In spite of this limitation, our approach revealed a significant approx. 2-fold increase in the concentration of DOX and aPD-1 in the biopsies of sonicated brain obtained in two patients, compared to the non-sonicated brain samples from the same patients. We have now mentioned the limitations of this approach in the discussion section.

Figure 9. Visualization of sonicated and non-sonicated peritumoral brain regions in GBM patients. a, Representative examples of BBB opening in one patient. T1-weighted MRI scans with contrast sequences show gadolinium leaking into the peritumoral brain after sonication (post-LIPU-MB images), with pre-sonication images (preoperative, postoperative, and pre-LIPU-MB) shown for comparison. Brain enhancement (seen as hyperintensity) on post-LIPU-MB images that is not seen on pre-LIPU-MB images represents BBB opening elicited by SC9, with the permeation of gadolinium. Post-LIPU-MB scans show patients who had gadolinium injection before paclitaxel infusion, within approximately 2 min of LIPU-MB. LIPU-MB=low-intensity pulsed ultrasound with concomitant administration of intravenous microbubbles. SC9=SonoCloud-9. **b,** DCE MRI scans for a GBM patient. These scans demarcate the sonicated (top) and non-sonicated regions (bottom left), corroborating the simulated acoustic pressure fields and providing visual evidence of the biopsied areas. A magnification of the MRI scan is shown to visualize the

position of the ultrasound emitter represented in yellow in relation to the biopsy site. 3D modeling of the simulated acoustic pressure fields projected from the SonoCloud-9 US device emitters (bottom right), illustrating the expected zones of BBB disruption.

In the human studies it is not clear when the Dox infusion started with respect to FUS and how this protocol has been selected and subsequently replicated in mice.

For the human studies, DOX was administered immediately after sonication, over 30 minutes in all cases. We did this to maximize the amount of drug that would penetrate across the BBB, as there is rapid restoration of BBB integrity (Sonabend, et al. Lancet Oncology)⁶. Anti-PD-1 immunotherapy was administered prior to sonication, as its half-life is long, and plasma concentrations are constant during the 1 hour after sonication due to the long plasma half-life. In mice this cannot be completely replicated, as there are differences in doses and administration techniques. In this case, drugs were administered intravenously as a bolus before sonication. We have further clarified this in the manuscript in the results and the methods section.

Data assessing the safety of the proposed treatment strategy, especially in healthy brains are needed.

In response to the need for safety assessment of our treatment strategy, we extended our toxicity study to include histopathological examination of the brain tissue alongside weight monitoring in healthy mice. The histological analysis focused on detecting potential morphological changes in healthy brain regions from mice in two treatment groups: DOX plus LIPU/MB and DOX + anti-PD-1 delivered with LIPU/MB (**Fig. 10a of this document**).

Our histopathological findings showed no observable tissue damage or morphological alterations in the brains of treated mice compared to controls, suggesting the absence of direct neuronal toxicity due to the treatment (**Fig. 10a of this document**). Complementing these observations, the body weight of the mice remained stable across all treatment groups throughout the observation period (**Fig. 10b of this document**), which aligns with the absence of systemic toxicity.

Figure 10. Assessment of systemic toxicity in mice following DOX plus aPD-1 delivered with LIPU/MB. a, H&E micrographs showing low (left) and high (right) magnification images of mouse brains from a control group that did not receive treatment, a DOX + LIPU/MB group, and from the group treated with DOX + aPD-1 + LIPU/MB. Images are representative of 3 mice per group. **b,** Line graph showing longitudinal monitoring of mouse body weight post-treatment with a therapeutic combination of DOX, aPD-1, and LIPU/MB. The control group (black line) and the treated group (blue line) are compared over 40 days. Body weight is a general indicator of health and potential systemic toxicity in murine models. The data suggests no significant weight loss across all groups, indicating that the treatment was well-tolerated without overt signs of systemic toxicity.

I wonder if the error in the measurements shown in Fig 1d (2.033-fold increase) is 0.001?

Thank you for your inquiry regarding the statistical error presented in Figure 1d. Upon reviewing the descriptive statistics of our analysis, we would like to clarify, for the non-sonicated peritumoral brain measurements, the standard deviation (SD) is 0.2902, and the standard error of the mean (SEM) is 0.1451. These statistics for the sonicated peritumoral brain are an SD of 0.7498 and a SEM of 0.2651. These values represent the variability and the precision of our dataset respectively, and neither equates to an error of 0.001.

The subplots in Fig. 3 not cited correctly in the main text.

Thank you for highlighting this issue. We have revised the main text to correctly cite the subplots in Figure 3 of the revised manuscript.

In the methods there is a description of experiments with carboplatin (line 1007), but I could not find any data with this drug.

Thank you for pointing out the discrepancy. The mention of carboplatin in the methods section was an oversight, as this drug was not part of the current study's analysis. We have corrected this error in the manuscript.

Reviewer #4 (Remarks to the Author): with expertise in brain tumors, immunology

In this report, investigators use ultrasound combined with microbubble administration to open the blood-brain barrier and allow increased access to the brain of delivered compounds, here liposomal doxorubicin and anti-PD1. They report data in patients suffering from GBM and in the GL261 and CT-2A mouse models.

The LIPU/MB technique has been published already and is being tested in clinical trials. The combination of a cytotoxic drug, which activates innate immune responses, with an immune checkpoint inhibitor is exciting and the aim of the study is to increase penetrance of both compounds in the brain. A strength of the study is the availability of pre- and post-treatment tumor samples (in all 4 patients), with, in 2 of them, availability of peritumor areas to assess compound influx in sonicated vs. non-sonicated areas.

Overall, this study carefully investigates the potential of the approach, which is of high interest and timely. However, there are concerns that have to be addressed before this work can be considered for publication.

Major comments:

1. Figure 1: why is a violin plot used in panel c whereas all other panels use bar plots? Data should be presented as bar plots in panels c, d, f and g or the reason for doing differently explained.

Thank you for your observation. We recognize the importance of consistency in data presentation and the value of bar plots in providing a standardized view of data across different panels. To address this, we have changed our figure with a bar plot showing individual data points in the Extended Data Figure 1 (**Fig. 11 of this manuscript**). This allows for a comprehensive view of the data, catering to the detailed distribution information provided by the bar plots.

Figure 11. LIPU/MB increases DOX concentrations in the human brain. Bar plot showing the concentrations of DOX in sonicated and non-sonicated peritumoral brain regions obtained during surgery after 48 hours of DOX infusion.

In addition, how many mice were used in the experiment, and which points correspond to each time point (45 minutes, 24 and 48h)?

In this specific experiment, we focused on a single time point for brain harvesting and DOX quantification. This time point was set at 48 hours following LIPU/MB treatment to model the scenario we had in the human samples. We have now explicitly indicated this in the figure caption and within the figure itself. Regarding the number of mice used in the experiment, 5 mice per group were utilized.

Finally, in panels c, d, f, g: would it be possible to identify to which of the 2 patients/mice the dots originate, in order to see variation within sonicated vs. non-sonicated brain areas in given individuals?

Thank you for this suggestion. In response to your comment, we have revised panels c and d of the main figures to enhance the clarity and informativeness of our data presentation. We generated a new version of these figures where the data points are now color-coded to indicate their origin from the two different patients (Fig. 11 and 12).

Figure 12. LIPU/MB increases DOX concentrations in the human brain. Bar plot representing the fold change in DOX concentration in non-sonicated (n = 4 brain samples) and sonicated peritumoral brain tissues (n = 8 brain samples) from 2 GBM patients.

2. Figure 2: is absence of SOX2 expression in one patient indicative of absence of tumor in that sample? Would it be possible to use another tumor marker (such as GFAP) instead of SOX2 to identify tumor cells, in order potentially to be able to analyze all 4 patients? In addition, it is not correct to say that there is an increase in SOX2+ HLA-ABC+ cells if the p value is above 0.05 (defined as such in the methods). This should be changed and the title and concluding sentence of the paragraph should be tempered. Finally, are HLA class I and II upregulated in non-tumor cells as well?

Thank you for your observations regarding the use of SOX2 as a tumor cell marker in Figure 2. In response to your comments, we have revised our staining protocol by adjusting the dilution of the SOX2 antibody to 1:1000, which allowed for enhanced detection of SOX2 expression. With this optimized staining, we re-stained all tumor samples and have now included data from the fourth GBM patient that was previously detected at low levels. This has strengthened our analysis by enabling the inclusion of all patient samples in our study.

Our revised immunophenotyping analysis using a mixed-effects model, which accounts for both within-patient and between-patient variability, has revealed a statistically significant increase in the density of SOX2+ HLA-ABC+ cells ($P = 0.0079$, **Fig. 1a of this document**) and SOX2+ HLA-DR+ cells ($P = 0.024$, **Fig. 1b of this document**) in on-treatment GBM samples compared to pre-treatment GBM samples.

Furthermore, to address the potential upregulation of HLA class I and II in non-tumor cells, we expanded our analysis to include TMEM119 and CD163 as markers for myeloid cells. Our analysis shows increased cell density of TMEM119+ HLA-ABC+ cells in on-treatment GBM samples ($P = 0.03229$) and a trend towards increased CD163+ HLA-ABC+ cell density without reaching statistical significance ($P = 0.066$; **Fig. 13a of this document**). We also evaluated HLA-DR+ in TMEM119+ cells ($P = 0.071$) and CD163+ ($P = 0.16$). However, these analyses did not reach statistical significance (**Fig. 13b**), thus the results suggest that these myeloid cells/microglia up-regulate HLA-ABC rather as opposed to an increase in the cell number for these phenotypes.

We have carefully revised the language in our manuscript to accurately reflect the statistical outcomes of our analyses. We now state that there is a 'significant increase' only when the p-value is below 0.05, and we describe any results with p-values above this threshold as 'trends' rather than definitive increases.

Figure 13. Differential expression of antigen-presenting molecules in microglial and macrophage populations in GBM patient samples before and after DOX treatment. **a**, Dot plot showing the cell density of TMEM119⁺ HLA-ABC⁺ and CD163⁺ HLA-ABC⁺ cells in pre-treatment GBM samples and on-treatment GBM samples. **b**, Dot plot showing the cell density of TMEM119⁺ HLA-DR⁺ and CD163⁺ HLA-DR⁺ cells in pre-treatment GBM samples and on-treatment GBM samples. *n* = 4 paired GBM samples. A mixed effects model was constructed considering DOX treatment as a fixed effect and patients as a random effect influencing the cell density of the evaluated phenotypes. *P* values were obtained by likelihood ratio tests of the full model with the effect in question against the model without the effect in question.

3. In addition to data presented in Figure 2f and g, would the observed levels of increase in MHC class I upregulation translate into efficient peptide presentation and tumor cell recognition by T cells? Is that testable (have T cell epitopes in GBM6/GBM63 been described or are other models available to test that)?

We appreciate the reviewer's inquiry into the functional consequences of MHC class I upregulation on peptide presentation and subsequent T cell recognition in our model. In response, we conducted a T cell proliferation assay using CD8⁺ T cells from OT-1 mice (**Fig. 14a of this document**), which are known to recognize the OVA peptide presented by MHC class I. Upon co-culture with GL261-OVA cells treated with DOX, we observed an increase in T cell proliferation. This is demonstrated in the flow cytometry results, where enhanced proliferation is indicated by the dilution of the BV421 dye, signifying cell division (**Fig. 14b of this document**). The Proliferation Index further supports this, showing a statistically significant increase in the presence of DOX at various concentrations (**Fig. 14c of this document**).

These results suggest that the upregulation of MHC class I molecules on the tumor cells by DOX treatment does indeed facilitate more efficient antigen presentation to T cells, leading to their activation and proliferation.

Figure 14: Assessment of CD8⁺ T cell proliferation in response to DOX-treated glioma cells. **a**, Schematic representation of the experimental setup. CD8⁺ T cells, isolated from OT-1 mice, are co-cultured with GL261-OVA glioma cells treated with varying concentrations of DOX to assess T cell activation and proliferation. **b**, Flow cytometry scatter plot displaying CD8⁺ T cell proliferation after co-culture with GL261-OVA cells, untreated (red dots) or treated with 0.3 μM DOX (blue dots). Proliferation was measured by the dilution of the proliferation dye BV421. **c**, Bar graph showing the CD8⁺ T cell Proliferation Index for untreated and DOX-treated GL261-OVA cells at different concentrations (0, 0.1, 0.3, and 0.6 μM). The Proliferation Index is calculated based on the fluorescence intensity decay of BV421, indicating the division of T cells. Error bars represent standard

deviation, and P-values indicate the statistical significance of increased proliferation compared to the control group derived from one-way ANOVA with post hoc Dunnett's multiple comparisons test.

4. The gating of the macrophage population in Figure 3 is unusual. Whereas the CD45⁺⁺ CD11b⁺⁺ population has been described as macrophages, the classification of the CD11b intermediate population as macrophages, with the gate expanding far towards lymphocytes is not typical (Gabrusiewicz et al., JCI Insights 2016, Brandenburg et al. IJMS 2020, Khan et al., JCI 2023 among others). Do authors have confirmation that this population is macrophages? Would the results be different if the gating was restricted to the CD45⁺⁺ CD11b⁺⁺ population?

We appreciate the reviewer's constructive critique of our original gating strategy for macrophage populations in Figure 3. We have revised our approach and re-gated to make for a tighter population, distinguishing CD45⁺⁺ CD11b⁺⁺ cells as macrophages.

With the implementation of our refined gating strategy (Fig. 15 of this document), we have also re-analyzed the phenotype of this macrophage population and presented the updated results in the manuscript. The new gating strategy has been integrated into the Extended Data Fig. 3 of our manuscript.

Figure 15. Gating strategy used to analyze GBM-infiltrating immune cells in mice treated with liposomal DOX.

Concerning the question of whether this population represents macrophages, we leveraged findings from a single-cell RNA-seq study that elucidated the myeloid cell landscape of GL261 gliomas²³. In this murine model, microglia were characterized by *Tmem119* and *P2ry12* expression (Fig. 16a of this document). On the other hand, monocyte-derived macrophages were characterized by *Lgals3* (Gal-3) and a higher expression of *Ptpnc1* (CD45) compared to microglia (Fig. 16b of this document). By flow cytometry, these authors ascertained that CD45⁺ CD11b^{hi} cells aligned with monocyte-derived macrophages, a classification further supported by Gal-3⁺ protein expression (Fig. 16c of this document). Similarly, CD45^{lo} CD11b⁺ cells were identified as microglia, confirmed by the expression of TMEM119 (Fig. 16c of this document). This robust body of evidence supports the validity of our revised gating strategy, affirming that CD45⁺ CD11b⁺ cells are indeed representative of monocyte-derived macrophages.

Fig. 16. Transcriptomic characterization of main myeloid subpopulations. a, b, Feature plots depicting genes highly expressed in microglia (a) and monocyte-derived macrophages (b). **c** Flow cytometric analysis of the distribution of Tmem119 and Gal-3 protein markers within CD11b⁺ cells and projection of Tmem119⁺ and Gal-3⁺ cells onto CD45/CD11b graphs, dot plots demonstrate percentages of Tmem119⁺ and Gal-3⁺ cells within CD45^{hi} and CD45^{lo} groups (n= 8, 4 males, and 4 females, two-sided Mann-Whitney U test, mean ± SD, **** < 0.001, Tmem119 Pv = 0.0002, Gal-3 Pv = 0.0002). Figure legend adapted from Ochocka et al., 2021²³.

5. How was the setting of the positive gate for IFN-g production as shown in Figure 3d determined? It seems to cut the population in two without a clear negative and positive population being visible. An isotype control should be used or the gate be set at the right of the population in the control condition. Data in panels b, c and d should be reanalyzed once the gating has been modified. The same applies to Figure 4e-g. Similarly, isotypes should be used for assessment of TNF-a and IL-1b secretion as well as of H2-Kb and PD-L1 expression (extended data fig 2 and 3).

We thank the reviewer for their insightful suggestions regarding the gating strategy for cytokine production. As suggested, we have implemented a re-gating process using reference control conditions to set the positive gates. Therefore, we have re-evaluated and adjusted our gating strategy for IFN- γ (Fig. 17a of this document), TNF- α (Fig. 17b of this document), and confirmed the gating for IL-1 β , which remained unchanged (Fig. 17c of this document). We modified the panel in Figure 3 of the manuscript to show representative scatter plots for each treatment condition (Fig. 17d of this document) Importantly, this re-analysis did not change the main finding that 5 mg/kg of DOX delivered with LIPU/MB into the brain induces the production of IFN- γ by microglia (Fig. 17e of this document). Furthermore, following the re-gating process for macrophages addressed earlier (Fig. 15 of this document), we have a more precise identification of positive populations for IFN- γ that refines and strengthens the finding that 5 mg/kg of DOX delivered with LIPU/MB has the same effect of inducing the production of IFN- γ by macrophages (Fig. 17f of this document).

Our re-analysis, with the revised gating, confirms the results observed in our initial submission for TNF- α and IL-1 β with no significant alterations of these molecules following treatment with DOX and LIPU/MB in both microglia and macrophages (Fig. 17g and 17h of this document). Notwithstanding the foregoing, as a complementary analysis that takes into account all myeloid cells, we provide the MFI for the analyzed cytokines for both microglia and macrophages. In this way, we can also confirm the increased production of IFN- γ following 5 mg/kg of DOX delivered with LIPU/MB (Fig. 17i of this document). On the other hand, we did not find differences in TNF- α and IL-1 β MFI across conditions (Fig. 17j and 17k of this document).

Figure 17. Revised gating strategy and cytokine expression analysis. **a, b, c,** Representative scatter plots from each treatment group showing the gating strategy derived from control conditions to determine a positive gate for IFN- γ (**a**), TNF- α (**b**) and IL-1 β (**c**). **e, f,** Bar plots showing the percentage of cells IFN- γ ⁺ from groups treated with different doses of liposomal DOX (1, 2, and 5 mg/kg) with or without LIPU/MB in microglia (**e**) and macrophages (**f**). **g, h,** Bar plots showing the percentage of cells TNF- α ⁺ (**g**) and IL-1 β ⁺ (**h**) from groups treated with different doses of liposomal DOX (1, 2, and 5 mg/kg) with or without LIPU/MB in

microglia and macrophages. **i, j, k**, Bar plots showing the MFI of cells IFN- γ^+ (**i**) TNF- α^+ (**j**) and IL-1 β^+ (**k**) from groups treated with different doses of liposomal DOX (1, 2, and 5 mg/kg) with or without LIPU/MB in microglia (top) and macrophages (bottom). *P* values were derived from one way-ANOVA with post hoc Tukey's multiple comparisons test. Data are presented as mean \pm SEM.

Considering the continuous expression of H2-K^b and PD-L1, we employed the MFI as a quantitative measure (**Fig. 18a of this document**), allowing us to corroborate the originally reported expression levels within microglia and macrophage populations (**Fig. 18b of this document**).

We have incorporated these updated results as well as the scatter plots showing how we set the positive populations for the analyzed cytokines to the Extended Data Fig. 3 of the manuscript.

Figure 18. Assessment of H2-K^b and PD-L1 expression in microglia and macrophages. **a**, Bar plots showing the MFI of H2-K^b in microglia (left) and macrophages (right). **b**, Bar plots showing the MFI of PD-L1 in microglia (left) and macrophages (right). Mouse groups included a control group with and without LIPU/MB and those treated with different doses of liposomal DOX (1, 2, and 5 mg/kg) with or without LIPU/MB. *P* values were derived from one way-ANOVA with post hoc Tukey's multiple comparisons test. Data are presented as mean \pm SEM.

6. In the experiment shown in Figure 3 (a-e), were the number or percentages of macrophages and microglial cells modified upon treatment? (This is shown for lymphoid cells but not for myeloid cells.)

In response to the reviewer's inquiry about Figure 3 (a-e), following the revised gating based on CD45⁺ and CD11b⁺ markers to define macrophages (**Fig. 19 of this document**), we quantified the percentages of both microglia and macrophages from the total live cells. With this, we did not find statistically significant differences in the percentages of these cells upon treatment with different doses of DOX delivered with and without LIPU/MB.

Figure 19. Percentages of microglia and macrophages in the brain of GBM-bearing mice following treatment with DOX delivered with LIPU/MB. a, b. Bar plots showing the percentages of microglia (a) and macrophages (b) from groups treated with different doses of liposomal DOX (1, 2, and 5 mg/kg) with or without LIPU/MB. *P* values were derived from one way-ANOVA with post hoc Tukey's multiple comparisons test. Data are presented as mean \pm SEM.

7. Figure 3 f: how do authors explain increase in MHC in microglial cells with 5 mg/kg of DOX in the absence of LIPU/MB whereas no IFN- γ is detected in that condition (panel b)? What are the other possible mechanisms in play here?

For this analysis, it is important to consider that the murine models have relative lack of blood-brain barrier in the tumor core, where DOX concentration might be higher than in the brain. The increase in MHC class I expression in microglial cells following treatment with 5 mg/kg of DOX, even in the absence of LIPU/MB, suggests that DOX at this concentration can affect antigen presentation independently of its enhanced delivery into the brain by LIPU/MB. Previous studies have shown that exposure of the brain parenchyma to even low levels of DOX may induce microglial inflammation in the brain²⁴. Indeed, the upregulation of MHC class I by DOX may be due to a direct pharmacological effect of the drug on the cells, such as through the induction of cellular stress responses, which are known to enhance MHC expression^{2,25}.

On the other hand, the detection of IFN- γ production only in the presence of LIPU/MB suggests that a higher intra-tumoral concentration of DOX, achieved through ultrasound-mediated BBB opening, is necessary for inducing a robust IFN- γ response. This could be due to the requirement for higher intracellular drug concentrations to trigger the pathways leading to IFN- γ production, which are different from those required for MHC class I upregulation.²⁶ It is also possible that the induction of IFN- γ requires an interaction between DOX plus the mechanical/acoustic stimulation achieved LIPU/MB.

We have added a section in our discussion that critically evaluates these findings and explores the potential mechanisms at play. This includes a consideration of direct drug effects, differential permeability, immune cell crosstalk, and the role of the tumor microenvironment in modulating the immune response.

8. The observation that type II IFN is secreted by myeloid cells, including microglia, is not common. Measure of type I IFN would be informative in that regard, also as authors did not observe induction of inflammatory cytokines such as IL-1b and TNF-a.

In response, we evaluated the expression of type I interferons (IFN- α and IFN- β) and IFN- γ by treating our microglia cell line with DOX for 5 hours, cultured with drug-free media and performed gene expression analysis (**Fig. 20a of this document**). Our RT-qPCR data, as depicted in the provided bar plots, indicates a significant induction of IFN- γ in response to DOX treatment (**Fig. 20b of this document**) which is consistent with the flow cytometry data in glioma-associated microglia and HMC3 cells.

Additionally, we assessed the expression levels of type I interferons, IFN- α and IFN- β . The results exhibit variability in expression for *IFNA1*, with certain DOX concentrations prompting a modest increase, though not reaching statistical significance (**Fig. 20c of this document**). Interestingly, we found that DOX treatment upregulated *IFNB1* in our microglial cell line (**Fig. 20d of this document**). This new finding complements our initial observations regarding IFN type II secretion by myeloid cells.

As for the inflammatory cytokines such as IL-1 β and TNF- α , our study did not observe their induction under the experimental conditions tested. This could be due to various factors, including the specific microenvironment of the brain, the timing of sample collection, or the particular phenotype of myeloid cells engaged in our model.

Our findings lend support to emerging evidence that myeloid cells possess a diverse repertoire of immune functions, including the secretion of IFN- γ . Such responses may be influenced by epigenetic remodeling and the transcriptional activation of specific genes, such as those encoding IFN- γ and its associated transcription factors, *Eomes* and *Tbx21* (T-bet)²⁷, which can lead to increased production of IFN- γ in both T cells and glioma-associated myeloid cells.

We want to mention that we planned to perform an *in vivo* immunophenotyping analysis of mice treated with DOX to determine the production of type I IFN. However, there are no reliable antibodies against these cytokines that are commercially available. Thus, we relied on qPCR to determine whether type I and II are produced by microglia.

In conclusion, our data suggest that DOX can initiate transcriptional activity of genes related to type I and II interferons, demonstrating a broader scope of DOX's immunomodulatory effects in microglia.

Figure 20. Upregulation of type II IFN in HMC3 cells by DOX treatment. **a**, Schematic of the RT-qPCR experiment performed to assess the effect of different DOX concentrations on the expression of *IFNG*, *IFNA1*, and *IFNB1* in HMC3 cells. **b**, **c**, **d**, Bar plots representing the expression of *IFNG* (**b**), *IFNA1* (**c**) and *IFNB1* (**d**) assessed as fold change in expression in HMC3 cells. *P* values were obtained by one-way ANOVA with post hoc Dunnett's multiple comparisons test.

9. Related to extended data Figure 4, could authors show an exemplary staining of secretion of cytokines by macrophages?

In response to your request for an illustrative example of cytokine secretion by macrophages derived from the *in vivo* immunophenotyping of GL261-bearing mice treated with different doses of DOX, we present flow cytometry plots for IFN- γ (**Fig. 21a of this document**), TNF- α (**Fig. 21b of this document**), and IL-1 β (**Fig. 21c of this document**). This figure provides a visual representation of positive macrophage populations and control conditions for the cytokines examined. The new gating strategy employed ensures that the analysis is restricted to the positive macrophage population. We trust that this addition provides a clear visual confirmation of the cytokine secretion dynamics that underlie our findings.

Figure 21. Representative flow cytometry plots of cytokine staining in macrophages. a, b, c, Scatter plots showing representative positive macrophage populations in the top for IFN- γ (a), TNF- α (b), and IL-1 β (c) as detected by specific antibody staining. Control conditions are shown below of the positive macrophage populations to delineate the baseline fluorescence and to highlight the positive staining for each cytokine of interest. These scatter plots are representative of GL261-bearing mice treated with different doses of liposomal DOX (1, 2, and 5 mg/kg) with or without LIPU/MB in macrophages.

There is in addition no description of these analyses (related to Figure 3 and extended data Figures 2 and 4 panels a-c) in the methods (no mention either of measure of TNF- α , IL-1 β or GZMb production in the methods).

We thank the reviewer for pointing out the omission in the methods section of our manuscript. To address this, we have now included a detailed description of the flow cytometry analysis for cytokine production in the methods section for clarity and reproducibility:

“Immune cell populations were first identified using forward and side scatter characteristics to exclude doublets, followed by viability dye exclusion for dead cells using the eBioscience Fixable Viability Dye eFluor780 (Thermo Fisher). After gating on the CD45⁺ and CD11b⁺ population to identify myeloid cells, and on CD45⁺ and CD11b⁻ for lymphocytes, we further characterized these populations by CD8⁺ and CD4⁺ markers for T cells. For the detection of cytokine production, cells were re-stimulated eBioscience Cell Stimulation Cocktail (plus protein transport inhibitors) to inhibit cytokine secretion, thus allowing for their accumulation within the cells. Following the re-stimulation, cells were fixed, permeabilized, and stained with antibodies against IFN- γ , GZMb, TNF- α , and IL-1 β . We used fluorochrome-conjugated antibodies specific to these cytokines. Macrophages and microglia were similarly analyzed for cytokine production and for the expression of H2-K^b and PD-L1 molecules.”

10. How do authors explain that DOX has no effect on peripheral immune cells? In addition, the high proportion of macrophages producing IFN-g in all conditions, even the control ones is unexpected. How can this be explained?

We agree with the reviewer that this is an interesting, and somehow unexpected finding. There might be several reasons for this: 1) Perhaps some of the effects of DOX on modulation of immune cell

phenotype is more pronounced for microglia than for bone-marrow derived myeloid cells (as we show on several of our analyses). 2) The phenotype of tumor-infiltrating microglia/macrophages in gliomas is distinct from that of the periphery, and the fact that in the brain. 3) DOX concentration remains elevated for at least 2 days, whereas DOX is cleared from the circulation faster.

Regarding the high proportion of macrophages producing IFN- γ , with the new gating strategy implemented for IFN- γ (Fig. 17a of this document), we found an overall lower percentage of IFN- γ ⁺ macrophages across all conditions except for the group treated with 5 mg/kg of DOX delivered with LIPU/MB (Fig. 17f of this document).

Figure 22. Percentages of monocytes producing IFN- γ in the blood of GBM-bearing mice following treatment with DOX delivered with LIPU/MB. a, b, Bar plots showing the percentages of monocytes IFN- γ ⁺ from groups treated with different doses of liposomal DOX (1, 2, and 5 mg/kg) with or without LIPU/MB. *P* values were derived from one-way ANOVA with post hoc Tukey's multiple comparisons test. Data are presented as mean \pm SEM.

11. In Figure 3 panel l and m, authors show increase of IFN-g+ TMEM119+ cells in human GBM samples after treatment. However, they do not mention the results regarding IFN-g+ CD163+ cells. Could this be added, also as it is further studied in panels n and o?

We have included the analysis of IFN- γ ⁺ CD163⁺ cells showing a non-significant trend towards increased cell density post-treatment (*P* = 0.081; Fig. 23 of this document). These results, alongside the increase in IFN- γ ⁺ TMEM119⁺ cells, have been incorporated into the revised manuscript to provide a comprehensive overview of the immune response in treated GBM samples.

Figure 23. Evaluation of an IFN- γ ⁺ phenotype in CD163⁺ myeloid cells following LIPU/MB-mediated delivery of liposomal DOX. Dot plot showing the cell density of CD163⁺ IFN- γ ⁺ cells in pre-treatment GBM samples and on-treatment GBM samples. *n* = 4 paired GBM samples. A mixed effects model was constructed considering DOX+aPD-I treatment as a fixed effect and patients

as a random effect influencing the cell density of CD163⁺ IFN- γ ⁺ cells. *P* values were obtained by likelihood ratio tests of the full model with the effect in question against the model without the effect in question.

12. Please specify the number of mice used in each treatment group in Figure 4 (the numbers are difficult to reconcile) as well as in other experiments where it is not specified.

In Figure 4 of the revised manuscript, we have included the number of mice in each treatment group in the figure legend. For other survival experiments where the number of mice was not previously specified, we have now included these details in the revised manuscript. For instance, we have included this information in the experiment detailed in Figures 5e, 6b, 6c, 6d, 6f, 6g, 8b, and 8d. We have revised the manuscript to specify the number of mice used in each group for all experiments to ensure transparency and reproducibility.

13. At several occasions (figure 2b, Extended data, Figure 4e, figure 4c, Figure 6d), authors say that there is an increase in a measured parameter although the p value is above 0.05. Strictly speaking, this is not correct and authors should modify the way they report it (also in the abstract regarding HLA I in patient samples). The observations are interesting and the p value likely linked to the limited number of patients or mice analyzed.

Thank you for your comment. We have reviewed the figures mentioned (Figure 2b, Extended Data, Figure 4e, Figure 4c, Figure 6d) and have made appropriate revisions to ensure that our language accurately reflects the statistical evidence. Where p-values are above the threshold of 0.05, we have adjusted the manuscript to indicate that there is a "trend" of an increase rather than stating there is an increase. This change has been applied consistently throughout the text. We acknowledge these trends may be due to our study's limited sample size, particularly for our patient cohort, and have adjusted our language accordingly.

14. An interesting experiment to confirm the role of myeloid cells in the effect of the LIPU/MB/DOX/anti-PD1 treatment

Thank you for suggesting the exploration of the role of myeloid cells in the efficacy of our LIPU/MB/DOX/anti-PD1 treatment. In response, we conducted a series of experiments using a CSF1 inhibitor (PLX3397) to deplete microglia and bone-marrow-derived macrophages, aiming to elucidate their role in our treatment strategy.

Initially, we confirmed the effective depletion of brain myeloid cells with PLX3397 (**Fig. 24a of this document**). Next, we combined PLX3397 with our therapeutic strategy involving DOX and aPD-1 delivered with LIPU/MB (**Fig. 24b of this document**). We observed a significant decrease in survival in the group with depleted GBM-infiltrating myeloid cells compared to the control diet group ($P = 0.0422$, log-rank test; **Fig. 24c of this document**). This result highlights the crucial role of myeloid cells in the treatment's efficacy.

Further emphasizing this point, in our experiment with long-term survivors who were rechallenged with CT2A, those treated with PLX3397 showed a significantly reduced survival following tumor rechallenge compared to mice on the control diet ($P = 0.0169$, log-rank test; **Fig. 24d of this document**). This suggests that the presence of myeloid cells is vital not only for the initial therapeutic efficacy but also for sustained resistance against tumor recurrence.

These experiments collectively underscore the significant influence of GBM-infiltrating myeloid cells in the therapeutic effectiveness of DOX plus aPD-1 delivered with LIPU/MB. The depletion of these cells notably compromises the treatment outcome.

Figure 24. Myeloid cells contribute to the efficacy of LIPU/MB in GBM-bearing mice. a, Representative flow cytometry plots (left) and bar plot (right) showing the depletion of brain myeloid cells. **b,** Therapeutic scheme employed following myeloid cell depletion using PLX3397 in GBM-bearing mice with the combination of liposomal DOX and aPD-1 delivered with LIPU/MB using CT2A cells. **c,** Kaplan-Meier curve showing the survival of CT2A-bearing mice treated with either control diet or PLX3397 followed by liposomal DOX, aPD-1, and LIPU/MB. **d,** Kaplan-Meier curve showing the survival of newly injected mice with CT2A and long-term survivors treated with either control diet or PLX3387 with the latter group treated with DOX, aPD-1, and LIPU/MB. *P* values in **c** and **d** were derived from the log-rank test.

15. How do the authors explain the fact that DOX does not induce IFN-g secretion by CD4 or CD8 T cells in absence of tumor (extended data Fig 3)?

We value the reviewer's observation on the absence of IFN- γ secretion by T cells in response to DOX treatment in our experimental setup. It's important to note that the mice groups mentioned have tumors. Our findings suggest that the immune response, particularly the secretion of IFN- γ by T cells, is linked to the presence of the brain tumor microenvironment where innate immune activation precedes and potentially primes later adaptive responses. The early time points post-treatment in our study likely capture these initial immune dynamics rather than the full maturation of T cell responses, as evidenced by the robust Th1 phenotype observed in long-term survivor mice treated with DOX and LIPU/MB.

16. How can the differences in survival observed in the cd8^{+/+} mice treated with CT-2A in figure 6d (80% survival with LIPU/MB+DOX+aPD-1) vs. figure 7b (40% survival) be explained?

Whereas these experiments show similar findings, the actual % of surviving mice is different. Considering the expected noise of *in vivo* experiments, and the relatively low number of mice per group, we do not think one can conclude that 80% in one experiment is meaningfully different than 40% in the different experiments. This underlies the importance of having control groups in each experiment. Yet, to further explore this, we have repeated this survival experiment and found the result to be robust and

consistent, as in the context of *Cd8^{-/-}* mice, the survival benefit of the therapy is diminished (**Fig. 3 of this document**).

As mentioned in our response to a previous comment, we conducted additional experiments to investigate the role of CD8⁺ T cells. We repeated the survival experiment, including a cohort of mice devoid of CD8⁺ T cells alongside the original wild-type mice (**Fig. 3 of this document**). This approach helped us to gain a broader understanding of the treatment's efficacy across different conditions and cohorts. The combined results, yielding a statistically significant improvement in survival ($P < 0.0001$) underline the importance of CD8⁺ T cells in our treatment strategy. However, it is crucial to note that individual experiments, as represented in different figures, may show variability in outcomes due to the aforementioned experimental nuances.

17. DOX has been shown in past studies to induce IFN-g-secreting T cells, which is not observed here (Figure 3). Is it due to a difference in the models used?

We think that the discrepancy noted in our findings can indeed be attributed to the timing of the immune response in our experimental brain tumor models. In our models, we focused on the early stages of the immune response in the brain following DOX treatment. As such, the robust phenotype of IFN- γ -secreting T cells might not have been pronounced at the time points we analyzed. In the context of enhanced delivery of DOX to the brain enabling an unseen local immune response, we found that activation of the innate immunity represented as the induction of an IFN- γ phenotype by microglia and bone-marrow-derived macrophages precedes the peak of the adaptive immune response, especially in terms of T cell activation. In our long-term survivor cohort of mice treated with DOX delivered with LIPU/MB, we indeed observed a more pronounced adaptive T cell response represented by the induction of a Th1 phenotype (IFN- γ and TNF- α) suggesting that these responses are more evident at later stages post-treatment.

Minor comments:

1. Methods: I assume that blood drawing was performed in patients at the time of biopsy/resection, but this could be specified in the methods.

Thank you for pointing this out. We have specified the following in the methods section:

“Blood samples were collected concurrently with peritumoral brain tissue during the patients’ surgical resection.”

2. Figure 1c, d, f, g: would it be possible to identify to which of the 2 patients/mice the dots originate, in order to see variation within sonicated vs. non-sonicated brain areas in given individuals?

We appreciate the suggestion to further delineate the data derived from different individuals in our study. As per your recommendation, we have updated Fig. 1d and Extended Data Fig. 1 with color coding to clearly distinguish the data derived from the two patients, enabling a direct visual comparison of sonicated and non-sonicated brain areas within individual subjects. This modification has been applied to the main figures in our manuscript to aid in the assessment of intra-individual variation and the localized effects of our treatment strategy.

We would like to also clarify that for Fig. 1f and Fig. 1g of the manuscript, every dot represents one independent mouse. We did not obtain sonicated and non-sonicated brain samples from the same mouse brain for the quantification of DOX.

3. Figure 2: do authors have an explanation why high doses of DOX do not increase and even decrease class I and class II expression in some instances?

Thank you for the insightful question regarding the differential effects of DOX on MHC class I and II expression at higher doses. The nuanced response of MHC molecule expression to DOX is indeed intriguing and is thought to be mediated by the drug's epigenetic influence. It has been established that DOX can cause histone eviction and chromatin remodeling in a dose-dependent manner, leading to transcriptionally active open chromatin states at certain genomic loci^{28,29}. This histone eviction, and consequent modulation of gene expression, is complex and appears to have a biphasic nature where low and high concentrations of DOX can differentially impact the binding of DNA-regulatory proteins to the chromatin³⁰. Specifically, at lower concentrations, DOX may enhance the chromatin association of certain DNA-binding proteins, while at higher concentrations, it could disrupt these associations. This may explain the observed decrease in MHC class I and II expression at higher DOX doses, as the drug could be altering the chromatin landscape around gene promoters in a manner that is not conducive to the transcription of these particular genes. Hence, while our data provide preliminary insights, we advocate for further research to dissect these intricate epigenetic mechanisms related to different DOX concentrations and their implications for MHC expression and immune response modulation in GBM treatments.

4. Legend to extended data Fig 2: please correct: Microglia were gated based on CD45- and CD11b- (should be CD11b+). In addition, they are usually CD45dim, although the brightness of the staining might make them CD45- here.

We thank the reviewer for the observation regarding the legend to Extended Data Figure 2. We have amended the legend to accurately reflect the gating strategy for microglia. The corrected legend now reads: "Microglia were gated based on CD45^{dim} and CD11b⁺."

5. Legend to extended data Fig 3: please correct, panel a is not present as described.

We appreciate your attention to detail regarding the legend for this Extended Data Figure. We have rectified this error in the figure legend to accurately reflect the content of this Extended Data Figure.

6. The caption for panel e is missing in Figure 3.

Thank for noting the oversight. We have included the caption for panel e in Figure 3 of the revised manuscript.

7. Figure 3, legends to panels m, n and o have been exchanged.

We have modified the order of the panels in this figure and corrected the figure legends.

References

1. Gaillard PJ, Appeldoorn CC, Dorland R, et al. Pharmacokinetics, brain delivery, and efficacy in brain tumor-bearing mice of glutathione pegylated liposomal doxorubicin (2B3-101). *PLoS One*. 2014;9(1):e82331. doi:10.1371/journal.pone.0082331

2. Alagkiozidis I, Facciabene A, Carpenito C, et al. Increased immunogenicity of surviving tumor cells enables cooperation between liposomal doxorubicin and IL-18. *J Transl Med*. Dec 10 2009;7:104. doi:10.1186/1479-5876-7-104
3. Takayama T, Shimizu T, Lila ASA, et al. Adjuvant Antitumor Immunity Contributes to the Overall Antitumor Effect of Pegylated Liposomal Doxorubicin (Doxil). *Pharmaceutics*. Oct 19 2020;12(10)doi:10.3390/pharmaceutics12100990
4. Marsh J, Mukherjee P, Seyfried TN. Akt-dependent proapoptotic effects of dietary restriction on late-stage management of a phosphatase and tensin homologue/tuberous sclerosis complex 2-deficient mouse astrocytoma. *Clin Cancer Res*. Dec 01 2008;14(23):7751-62. doi:10.1158/1078-0432.CCR-08-0213
5. Zhao J, Chen AX, Gartrell RD, et al. Immune and genomic correlates of response to anti-PD-1 immunotherapy in glioblastoma. *Nat Med*. Mar 2019;25(3):462-469. doi:10.1038/s41591-019-0349-y
6. Sonabend AM, Gould A, Amidei C, et al. Repeated blood-brain barrier opening with an implantable ultrasound device for delivery of albumin-bound paclitaxel in patients with recurrent glioblastoma: a phase 1 trial. *Lancet Oncol*. May 2023;24(5):509-522. doi:10.1016/S1470-2045(23)00112-2
7. Meng Y, Reilly RM, Pezo RC, et al. MR-guided focused ultrasound enhances delivery of trastuzumab to Her2-positive brain metastases. *Sci Transl Med*. Oct 13 2021;13(615):eabj4011. doi:10.1126/scitranslmed.abj4011
8. Portnow J, Wang D, Blanchard MS, et al. Systemic Anti-PD-1 Immunotherapy Results in PD-1 Blockade on T Cells in the Cerebrospinal Fluid. *JAMA Oncol*. Oct 2020;doi:10.1001/jamaoncol.2020.4508
9. Reardon DA, Brandes AA, Omuro A, et al. Effect of Nivolumab vs Bevacizumab in Patients With Recurrent Glioblastoma: The CheckMate 143 Phase 3 Randomized Clinical Trial. *JAMA Oncol*. May 2020;doi:10.1001/jamaoncol.2020.1024
10. Lim M, Weller M, Idbaih A, et al. Phase 3 Trial of Chemoradiotherapy With Temozolomide Plus Nivolumab or Placebo for Newly Diagnosed Glioblastoma With Methylated MGMT Promoter. *Neuro Oncol*. May 2 2022;doi:10.1093/neuonc/noac116
11. Omuro A, Brandes AA, Carpentier AF, et al. Radiotherapy Combined With Nivolumab or Temozolomide for Newly Diagnosed Glioblastoma With Unmethylated MGMT Promoter: An International Randomized Phase 3 Trial. *Neuro Oncol*. Apr 14 2022;doi:10.1093/neuonc/noac099
12. Arrieta VA, Chen AX, Kane JR, et al. ERK1/2 phosphorylation predicts survival following anti-PD-1 immunotherapy in recurrent glioblastoma. *Nat Cancer*. 2021;doi:10.1038/s43018-021-00260-2
13. Johanns TM, Miller CA, Dorward IG, et al. Immunogenomics of Hypermutated Glioblastoma: A Patient with Germline POLE Deficiency Treated with Checkpoint Blockade Immunotherapy. *Cancer Discov*. 11 2016;6(11):1230-1236. doi:10.1158/2159-8290.CD-16-0575
14. Arrieta VA, Duerinck J, Burdett KB, et al. ERK1/2 Phosphorylation Predicts Survival in Recurrent Glioblastoma Following Intracerebral and Adjuvant PD-1/CTLA-4 Immunotherapy: A REMARK-Guided Analysis. *Clin Cancer Res*. Nov 08 2023;doi:10.1158/1078-0432.CCR-23-1889
15. Arrieta VA, Dmello C, McGrail DJ, et al. Immune checkpoint blockade in glioblastoma: from tumor heterogeneity to personalized treatment. *J Clin Invest*. Jan 17 2023;133(2)doi:10.1172/JCI163447

16. Yost KE, Satpathy AT, Wells DK, et al. Clonal replacement of tumor-specific T cells following PD-1 blockade. *Nat Med.* 08 2019;25(8):1251-1259. doi:10.1038/s41591-019-0522-3
17. Kim C, Guo Y, Velalopoulou A, et al. Closed-loop trans-skull ultrasound hyperthermia leads to improved drug delivery from thermosensitive drugs and promotes changes in vascular transport dynamics in brain tumors. *Theranostics.* 2021;11(15):7276-7293. doi:10.7150/thno.54630
18. Kovacs Z, Werner B, Rassi A, Sass JO, Martin-Fiori E, Bernasconi M. Prolonged survival upon ultrasound-enhanced doxorubicin delivery in two syngenic glioblastoma mouse models. *J Control Release.* Aug 2014;187:74-82. doi:10.1016/j.jconrel.2014.05.033
19. Mainprize T, Lipsman N, Huang Y, et al. Blood-Brain Barrier Opening in Primary Brain Tumors with Non-invasive MR-Guided Focused Ultrasound: A Clinical Safety and Feasibility Study. *Sci Rep.* Jan 23 2019;9(1):321. doi:10.1038/s41598-018-36340-0
20. Rezai AR, D'Haese PF, Finomore V, et al. Ultrasound Blood-Brain Barrier Opening and Aducanumab in Alzheimer's Disease. *N Engl J Med.* Jan 04 2024;390(1):55-62. doi:10.1056/NEJMoa2308719
21. Lipsman N, Meng Y, Bethune AJ, et al. Blood-brain barrier opening in Alzheimer's disease using MR-guided focused ultrasound. *Nat Commun.* Jul 25 2018;9(1):2336. doi:10.1038/s41467-018-04529-6
22. Carpentier A, Stupp R, Sonabend A, et al. Repeated blood–brain barrier opening with a nine-emitter implantable ultrasound device in combination with carboplatin in recurrent glioblastoma: a phase I/II clinical trial. *Nature Communications.* In Press;
23. Ochocka N, Segit P, Walentynowicz KA, et al. Single-cell RNA sequencing reveals functional heterogeneity of glioma-associated brain macrophages. *Nat Commun.* 02 19 2021;12(1):1151. doi:10.1038/s41467-021-21407-w
24. Allen BD, Apodaca LA, Syage AR, et al. Attenuation of neuroinflammation reverses Adriamycin-induced cognitive impairments. *Acta Neuropathol Commun.* Nov 21 2019;7(1):186. doi:10.1186/s40478-019-0838-8
25. Buttiglieri S, Galetto A, Forno S, De Andrea M, Matera L. Influence of drug-induced apoptotic death on processing and presentation of tumor antigens by dendritic cells. *Int J Cancer.* Sep 10 2003;106(4):516-520. doi:10.1002/ijc.11243
26. Rock KL, Reits E, Neefjes J. Present Yourself! By MHC Class I and MHC Class II Molecules. *Trends Immunol.* Nov 2016;37(11):724-737. doi:10.1016/j.it.2016.08.010
27. Ma Y, Aymeric L, Locher C, et al. Contribution of IL-17-producing gamma delta T cells to the efficacy of anticancer chemotherapy. *J Exp Med.* Mar 2011;208(3):491-503. doi:10.1084/jem.20100269
28. Pang B, Qiao X, Janssen L, et al. Drug-induced histone eviction from open chromatin contributes to the chemotherapeutic effects of doxorubicin. *Nat Commun.* 2013;4:1908. doi:10.1038/ncomms2921
29. Yang F, Kemp CJ, Henikoff S. Doxorubicin enhances nucleosome turnover around promoters. *Curr Biol.* May 06 2013;23(9):782-7. doi:10.1016/j.cub.2013.03.043
30. Bosire R, Fadel L, Mocsár G, et al. Doxorubicin impacts chromatin binding of HMGB1, Histone H1 and retinoic acid receptor. *Sci Rep.* May 16 2022;12(1):8087. doi:10.1038/s41598-022-11994-z

REVIEWERS' COMMENTS

Reviewer #1 (Remarks to the Author):

The authors have adequately addressed all my queries raised during the initial review. The manuscript is now acceptable for further processing by the journal.

Reviewer #2 (Remarks to the Author):

The manuscript has been substantially revised and the authors addressed all my comments. I congratulate the authors on this important work.

Reviewer #3 (Remarks to the Author):

The authors addressed my critiques.

Here are a few suggestions that I hope will further improve the quality of the manuscript.

Line 140: Simply mention “2-fold” increase instead of “2.033-fold”, as the standard deviation (SD) is much larger than 0.01. Also align the significant digits reported in the paper according to the minimum standard deviation.

Line 72: The authors mention “This technology works by using a skull implantable device that sends ultrasound waves that induce the vibration of MB to open the BBB” please revise include non-implantable devices.

Lines 467-477: How do current findings with Dox or PD-1 compare with past investigations in terms of PK/PD and survival? Apart from covering larger brain area, does the frequency used offers any advantage? Note that current MRgFUS systems can cover larger area – see for example Anastasiadis et al., PNAS 2022. Finally, how do the selected exposure settings compare with past investigations (e.g. Pressure / Mechanical Index)?

Reviewer #4 (Remarks to the Author):

The revised version of Arrieta et al. addressed all comments and queries thoroughly with extensive new data notably on the characterization of the effect of DOX and aPD1 post LIPU on the resident myeloid cells and lymphocytes.

The manuscript is informative for the field and brings insights on the impact of BBB opening for drug penetrance. The upvalue of their approach is the BBB disruption on a broader volume as compared to other previously tested technologies, which provide opportunity to test a variety of non penetrating drugs and their impact on the TME.

REVIEWERS' COMMENTS

Reviewer #1 (Remarks to the Author):

The authors have adequately addressed all my queries raised during the initial review. The manuscript is now acceptable for further processing by the journal.

Thank you for your feedback and for confirming that our revisions have addressed your queries. We appreciate your approval and look forward to the next steps in the publication process.

Reviewer #2 (Remarks to the Author):

The manuscript has been substantially revised and the authors addressed all my comments. I congratulate the authors on this important work.

Thank you for your positive feedback and congratulations. We are grateful for your support and acknowledgment of our revisions.

Reviewer #3 (Remarks to the Author):

The authors addressed my critiques.

Here are a few suggestions that I hope will further improve the quality of the manuscript.

Line 140: Simply mention “2-fold” increase instead of “2.033-fold”, as the standard deviation (SD) is much larger than 0.01. Also align the significant digits reported in the paper according to the minimum standard deviation.

Line 72: The authors mention “This technology works by using a skull implantable device that sends ultrasound waves that induce the vibration of MB to open the BBB” please revise include non-implantable devices.

Lines 467-477: How do current findings with Dox or PD-1 compare with past investigations in terms of PK/PD and survival? Apart from covering larger brain area, does the frequency used offers any advantage? Note that current MRgFUS systems can cover larger area – see for example Anastasiadis et al., PNAS 2022. Finally, how do the selected exposure settings compare with past investigations (e.g. Pressure / Mechanical Index)?

Thank you for your comments and suggestions. Here are our responses to each point:

Line 140: We have revised the manuscript to mention a "2-fold" increase instead of "2.033-fold" to reflect the larger standard deviation associated with our measurements. We have also reviewed and aligned the significant digits throughout the paper to ensure consistency with the smallest standard deviation reported.

Line 72: We have updated this section to include information about non-implantable devices that use similar technologies to open the blood-brain barrier (BBB). This inclusion provides a broader perspective on the technology and its applications:

“This technology works by using a skull-implantable device or MRI-guided transcranial focused ultrasound that sends ultrasound waves that induce the vibration of MB to open the BBB”.

Lines 467-477: The main difference between past investigations and our findings is that we are able to cover broader regions of the brain with the ultrasound device that we employ. Regarding the frequency used, this depends on the therapeutic scheme of the drug employed with the ultrasound device. In our study, we used specific settings tailored for skull-implantable devices, which generally require different pressure and Mechanical Index (MI) levels due to their proximity to the target tissue and their operational mechanics compared to MR-guided ultrasound devices. The settings of our implantable ultrasound device are optimized for localized treatment with minimal invasiveness. We have provided a comparison in the discussion highlighting these differences.

Reviewer #4 (Remarks to the Author):

The revised version of Arrieta et al. addressed all comments and queries thoroughly with extensive new data notably on the characterization of the effect of DOX and aPD1 post LIPU on the resident myeloid cells and lymphocytes.

The manuscript is informative for the field and brings insights on the impact of BBB opening for drug penetrance. The upvalue of their approach is the BBB disruption on a broader volume as compared to other previously tested technologies, which provide opportunity to test a variety of non penetrating drugs and their impact on the TME.

Thank you for your positive feedback on our manuscript revisions. We are pleased that the additional data contributed to the understanding of BBB disruption's impact on drug delivery and the tumor microenvironment.